# A unified computational model for cortical post-synaptic plasticity

**Tuomo Mäki-Marttunen[1]\*, Nicolangelo Iannella[2], Andrew G Edwards[1], Gaute T Einevoll[3,4], Kim T Blackwell[5]**

[1]Simula Research Laboratory, Oslo, Norway; [2]Department of Biosciences, University of Oslo, Oslo, Norway; [3]Faculty of Science and Technology, Norwegian University of Life Sciences, Oslo, Norway; [4]Department of Physics, University of Oslo, Oslo, Norway; [5]The Krasnow Institute for Advanced Study, George Mason University, Fairfax, United States

**Abstract** Signalling pathways leading to post-synaptic plasticity have been examined in many types of experimental studies, but a unified picture on how multiple biochemical pathways collectively shape neocortical plasticity is missing. We built a biochemically detailed model of post-synaptic plasticity describing CaMKII, PKA, and PKC pathways and their contribution to synaptic potentiation or depression. We developed a statistical AMPA-receptor-tetramer model, which permits the estimation of the AMPA-receptor-mediated maximal synaptic conductance based on numbers of GluR1s and GluR2s predicted by the biochemical signalling model. We show that our model reproduces neuromodulator-gated spike-timing-dependent plasticity as observed in the visual cortex and can be fit to data from many cortical areas, uncovering the biochemical contributions of the pathways pinpointed by the underlying experimental studies. Our model explains the dependence of different forms of plasticity on the availability of different proteins and can be used for the study of mental disorder-associated impairments of cortical plasticity.

**\*For correspondence:**
tuomo@simula.no

**Competing interests:** The authors declare that no competing interests exist.

## Introduction

Synaptic plasticity in the neocortex has been under intense research since the first observations of neocortical long-term potentiation (LTP) (*Komatsu et al., 1981*; *Lee, 1982*). Although most often studied in brain slices, synaptic plasticity in the neocortex is a key phenomenon underlying vital mammalian brain processes ranging from formation and storage of memories to attentional selection (*Roelfsema and Holtmaat, 2018*). These processes are impaired in heritable mental illnesses such as schizophrenia and fragile X syndrome, as well as neurodegenerative diseases such as Alzheimer's disease, all of which have been associated with deficits in cortical plasticity (*Kantrowitz et al., 2017*; *Martin and Huntsman, 2012*; *Koch et al., 2014*). Improved understanding of neocortical synaptic plasticity all the way from molecular to circuit level is therefore needed to further our understanding of these yet incurable diseases.

Similar to hippocampal synaptic plasticity (*Larkman and Jack, 1995*), synaptic plasticity in the neocortex is highly variable — the outcomes of any plasticity-inducing protocol depends on the cortical area, neuron type as well as details of the stimulation protocol (*Castro-Alamancos et al., 1995*; *Froc and Racine, 2005*; *Sjöström et al., 2008*; *Feldman, 2009*). Computational models provide a tool for efficient hypothesis testing of mechanisms of neocortical plasticity, which helps to overcome the challenges posed by excessive variability. The foundations of our mechanistic understanding of neocortical synaptic plasticity lie upon the phenomenological Bienenstock-Cooper-Munro (BCM) theory, which predicts that small synaptic activity (later attributed to small $Ca^{2+}$ transients [*Bear et al., 1987*; *Lisman, 1989*]) cause long-term depression (LTD) whereas large synaptic activity (large $Ca^{2+}$ transients) give rise to LTP (*Bienenstock et al., 1982*). Simple BCM-based models and the closely

related models of spike-timing-dependent plasticity (STDP) have been widely used to explain the emergence of input-specific cell assemblies mediating, e.g., orientation selectivity (*Shouval et al., 1997*) or memory traces (*Klampfl and Maass, 2013*) in the cortex. These models, however, typically fail to provide a mechanistic understanding of the biochemistry within the synapse — namely, they do not reveal how various molecules downstream of $Ca^{2+}$ regulate the induction and maintenance of plasticity occurring in neuronal circuits, their composite neurons and synapses of the cortex. Moreover, current models often ignore the joint contributions of neuromodulators, which are critical for inducing some forms of cortical synaptic plasticity (*Meunier et al., 2017*; *Brzosko et al., 2019*). These shortcomings impede testing biochemical mechanisms of heritable mental illnesses associated with impaired cortical plasticity.

In this work, we aim at filling this gap of knowledge by introducing a biochemically detailed, mass-action law-based model of neocortical post-synaptic plasticity that can be used to study the induction of plasticity in different genetic conditions and neuromodulatory states, and under various stimulation protocols. Despite the lack of biochemically detailed models of synaptic plasticity in the neocortex, models of intracellular signalling have been used to study LTP and LTD in the hippocampus (*Bhalla and Iyengar, 1999*; *Jędrzejewska-Szmek et al., 2017*), cerebellum (*Gallimore et al., 2018*), and striatum (*Blackwell et al., 2019*). These models permit systematic studies on how patterns of $Ca^{2+}$ inputs to the post-synaptic spine, either alone or in combination with neuromodulatory actions, activate different signalling pathways leading to post-synaptic plasticity in the form of, e.g., AMPA-receptor (AMPAR) phosphorylation and membrane insertion. We integrate quantitative descriptions of the intracellular signalling pathways underlying synaptic plasticity in the neocortex into a unified model that is capable of describing both stimulation protocol-dependent plasticity, as well as neocortically observed neuromodulator-gated forms of STDP. We show that our model can be tuned by alterations of protein expression to reproduce not only BCM-like forms of plasticity but also experimental observations on neocortical plasticity from various cortical areas. Our results help to quantify and explain the differences in molecular constituents of different forms of neocortical LTP and LTD, and the different, data-fitted versions of our model can be directly used to examine the effects of chemical inhibitors and genetic manipulations of signalling proteins on synaptic plasticity in different cortical cells.

## Results

### Model construction

We reviewed the literature of molecular signalling pathways that needed for neocortical LTP/LTD, in particular in the post-synaptic spine of pyramidal cells (*Table 1A*). Three main pathways were highlighted in the experimental studies, namely, the protein kinase A (PKA), protein kinase C (PKC), and $Ca^{2+}$/calmodulin-dependent kinase II (CaMKII) pathways. To construct a computational model of cortical post-synaptic plasticity that describes these pathways, we adopted mass-action law-based descriptions of these pathways from biochemically detailed models of post-synaptic LTP/LTD in other brain areas, namely, hippocampus, basal ganglia and cerebellum (*Table 1B*). We prioritised the model components from hippocampal models due to the relatively small ontological differences between hippocampus and neocortex (*Kirsch and Chechik, 2016*). We focused on the effects of these pathways on AMPARs due to the better description of intracellular regulation of AMPAR dynamics in comparison to that of NMDA and kainate receptors or voltage-gated ion channels. In short, we based our model on that of *Jędrzejewska-Szmek et al., 2017*, which describes the PKA- and CaMKII-dependent phosphorylation of AMPAR subunit 1 (GluR1), and added the metabotropic glutamate receptor (mGluR) and muscarinic acetylcholine M1 receptor-mediated activation of PKC from *Kim et al., 2013* and *Blackwell et al., 2019*, respectively. Other types of receptors that interact with these pathways, such as serotonin (5HT) and dopamine receptors (*He et al., 2015*), have been shown to underlie certain types of neocortical plasticity. Dopamine D1/D5 receptors as well as serotonin 5HT4, 5HT6 and 5HT7 receptors are coupled to Gs proteins whereas 5HT2 receptors are Gq-coupled. The effects of these neurotransmitters would therefore be similar to those of norepinephrine and acetylcholine in our model (depending on the receptor composition in the post-synaptic neuron), and thus they are omitted in the present work. We then adopted the reactions describing PKC-dependent phosphorylation and endocytosis of AMPAR subunit 2 (GluR2) and

**Table 1.** Pathways contributing to cortical synaptic plasticity.

(A) Experimental evidence on the requirement of various molecular species for specific types of synaptic regulation in different cortical areas. (B) Model components needed for describing the modes of plasticity listed in (A). References are made to previous computational models describing these pathways. The types of phosphorylation of AMPAR subunit that mediate the plasticity are printed in bold.

**(A)**

| Pathway components | Type of neurons | Type of regulation | Pre-/post-synaptic | References |
|---|---|---|---|---|
| CaMKII | Cingulate cortex | Esophageal acid-induced sensitisation | post-syn. | *Banerjee et al., 2013* |
| CaMKII | Prefrontal cortex, pyramidal neurons | 5-HT1-induced modulation of AMPA currents | post-syn. | *Cai et al., 2002* |
| $\beta$-adr. receptors, PKA | Visual cortex, layer 4 pyramidal cells | Potentiation of AMPA currents | post-syn. | *Seol et al., 2007* |
| M1 receptors, PKC | Visual cortex, layer 4 pyramidal cells | Depression of AMPA currents | post-syn. | *Seol et al., 2007* |
| D1–PKA | Prefrontal cortex, pyramidal neurons | Potentiation of AMPA currents | post-syn. | *Sun et al., 2005* |
| $\beta$-adr. receptors | Frontal cortex | Potentiation of field EPSPs | n/a | *Sáez-Briones et al., 2015* |
| PKC | Cultured cortical neurons | Internalisation of AMPARs | post-syn. | *Chung et al., 2000* |
| ERK | Visual cortex | Potentiation of field EPSPs | n/a | *Di Cristo et al., 2001* |

**(B)**

| Molecular pathway | Cell type and references |
|---|---|
| $Ca^{2+} \rightarrow CaM \rightarrow CaMKII$ | Hippocampal CA1 neuron *Bhalla and Iyengar, 1999*; *Jędrzejewska-Szmek et al., 2017*, generic *Hayer and Bhalla, 2005*, |
| | cerebellar Purkinje cells *Gallimore et al., 2018*, striatal spiny projection neuron *Blackwell et al., 2019* |
| $CaMKII \rightarrow GluR1\ S831p$ | Hippocampal CA1 neuron *Jędrzejewska-Szmek et al., 2017* |
| $\beta$-adrenergic receptors $\rightarrow$ cAMP | Hippocampal CA1 neuron *Jędrzejewska-Szmek et al., 2017* |
| $cAMP \rightarrow PKA$ | Hippocampal CA1 neuron *Bhalla and Iyengar, 1999*; *Jędrzejewska-Szmek et al., 2017*, cerebellar Purkinje |
| | cells *Gallimore et al., 2018* |
| $PKA \rightarrow GluR1\ S845p$ | Hippocampal CA1 neuron *Jędrzejewska-Szmek et al., 2017* |
| M1 receptors $\rightarrow$ PLC | Cerebellar Purkinje cells *Gallimore et al., 2018* |
| $PLC \rightarrow PKC$ | Hippocampal CA1 neuron *Bhalla and Iyengar, 1999*, striatal spiny projection neuron |
| | *Kim et al., 2013*; *Blackwell et al., 2019* |
| | cerebellar Purkinje cells *Kotaleski et al., 2002*; *Gallimore et al., 2018* |
| $PKC \rightarrow GluR2\ S880p$ | Cerebellar Purkinje cells *Gallimore et al., 2018* |

reinsertion to the membrane from *Gallimore et al., 2018*, which allowed the representation of post-synaptic depression with our model. The pathways included in the model are illustrated in *Figure 1*. A description of the model calibration is given in Materials and methods, section '*Construction and calibration of the biochemically detailed model of post-synaptic plasticity in the cortex*', and the full set of model reactions and initial concentrations is provided in Tables 3 and 4, respectively.

## $Ca^{2+}$ activates multiple pathways that regulate the post-synaptic plasticity in cortical PCs

All pathways of *Table 1B* are $Ca^{2+}$-dependent, but due to the variability in binding rates and quantities of different $Ca^{2+}$-binding molecules, some pathways become more easily activated than others. This permits LTP or LTD to be induced in a way that is sensitive to the amount of $Ca^{2+}$ inputs and may serve as a basis for BCM-type rules of plasticity.

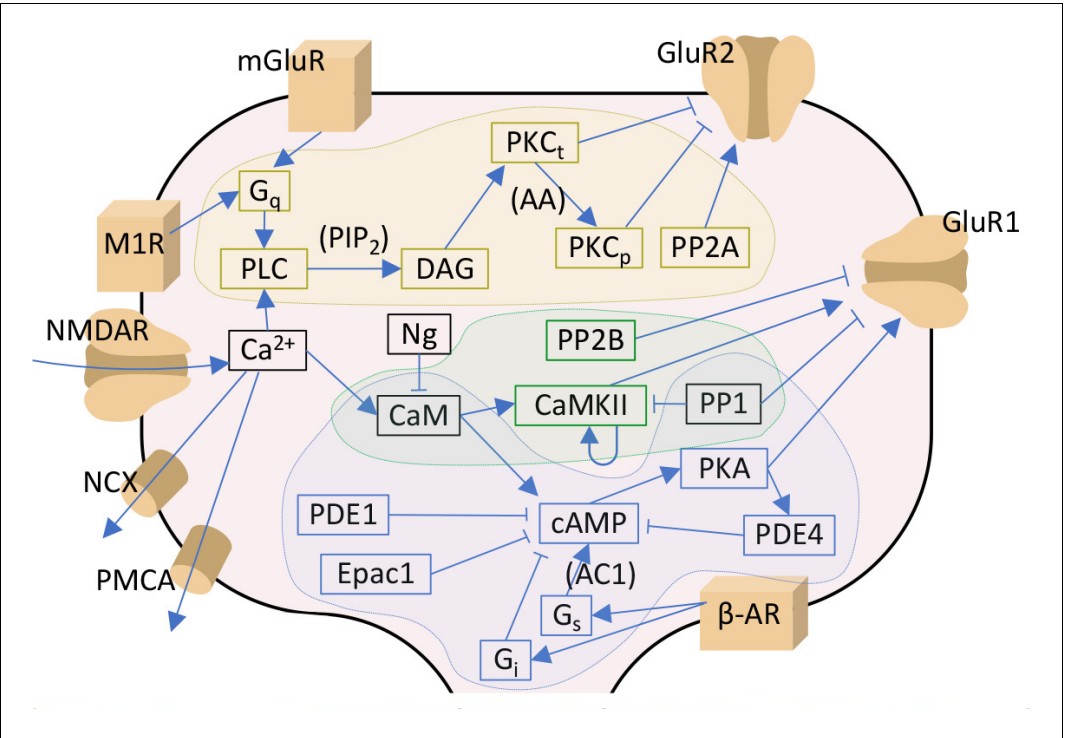

**Figure 1.** Signalling pathways included in the model. The PKA-pathway-related proteins and signalling molecules are highlighted by blue, PKC-pathway molecules by yellow, and CaMKII-pathway molecules by green colours. Reactions associated with a molecular species in parenthesis indicate a dependency on the denoted species — for details, see *Table 3*. Acronyms: β-AR – β-adrenergic receptor; AC1 and AC8 – adenylyl cyclase type 1 or 8; CaM – calmodulin; CaMKII – calmodulin-dependent protein kinase II; cAMP – cyclic adenosine monophosphate; DAG – diacylglycerol; Epac1 – exchange factor directly activated by cAMP 1; Gi, Gq and Gs – G-protein type I, Q, or S; GluR1 and GluR2 – AMPAR subunit 1 or 2; mGluR – metabotropic glutamate receptor; M1R – cholinergic receptor M1; NCX – Na$^+$-Ca$^{2+}$ exchanger; Ng – neurogranin; NMDAR – NMDA receptor; PDE1 and PDE4 – phosphodiesterase type 1 or 4; PIP$_2$ – phosphatidylinositol 4;5-bisphosphate; PKA – protein kinase A; PKCt and PKCp – transiently or persistently active protein kinase C; PLC – phospholipase C; PMCA – plasma membrane Ca$^{2+}$ ATPase; PP1 – protein phosphatase 1; PP2A – protein phosphatase 2A; PP2B – protein phosphatase 2B (calcineurin). In this work, the NMDARs are considered only in section '*Paired pre- and post-synaptic stimulation induces PKA- and PKC-dependent spike-timing-dependent plasticity (STDP) in GluR1-GluR2-balanced synapses*': in the rest of the work, Ca$^{2+}$ is directly injected as a square-pulse current into the spine.

To examine the sensitivities of LTP- and LTD-inducing pathways to Ca$^{2+}$, we simulated the injection of a prolonged square-pulse Ca$^{2+}$ input of varying magnitude (illustrated in *Figure 2A*) into the post-synaptic spine and quantified the degree of activation of each of the Ca$^{2+}$-binding molecules and the downstream signalling cascades. The simulations were carried out in the presence of mGluRs and β-adrenergic and cholinergic neuromodulation, which were modelled as prolonged square-pulse inputs as well.

The injected Ca$^{2+}$ quickly bound to Ca$^{2+}$ buffers (immobile buffer and calbindin, *Figure 2B*) and pumps (PMCA and NCX, *Figure 2C*) as well as to the proteins of the PKC pathway (phospholipase A2 (PLA2) and C (PLC), *Figure 2D*): a 95% saturation was reached in 1–2 s (*Figure 2B–D*). In contrast, the activation of calmodulin (CaM) was slower (*Figure 2E*): a 95% saturation was reached in 32–53 s, depending on the magnitude of the Ca$^{2+}$ input. Consistent with experimental literature, a vast majority of Ca$^{2+}$ was quickly bound and only a small fraction remained free in the cytosol (*Figure 2F*).

To further illustrate the differences between the activation patterns of these pathways, we quantified the degrees of Ca$^{2+}$ binding of these molecules in a steady state (5 min after the onset of Ca$^{2+}$) and the overall activation/deactivation of downstream molecules as a function of the magnitude of the Ca$^{2+}$ injection. Both PKC pathway-mediating proteins PLC, diacylglycerol lipase (DGL), and

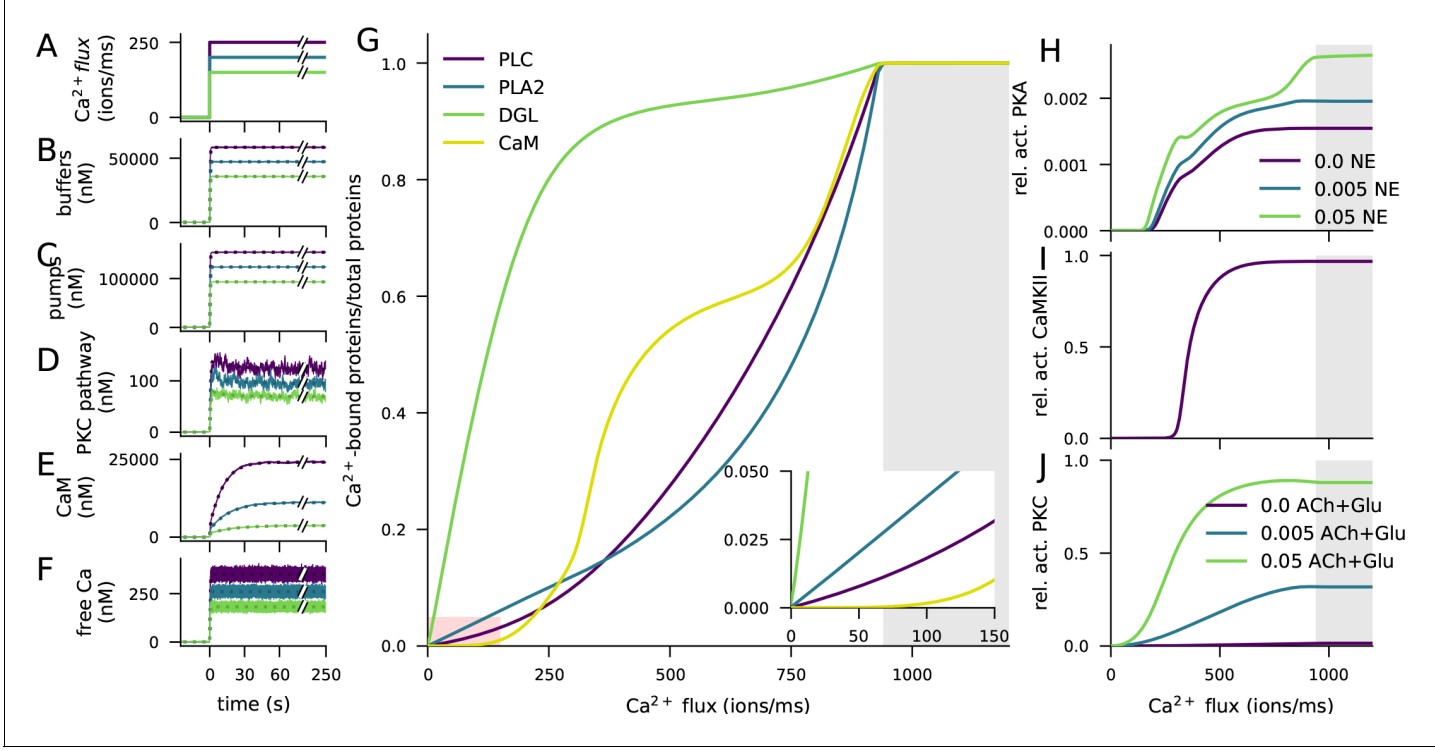

**Figure 2.** Ca$^{2+}$ activates CaMKII, PKA, and PKC pathways. (**A**) Illustration of the stimulus protocols with Ca$^{2+}$ flux amplitudes 150 (green), 200 (cyan), and 250 (purple) particles/ms. (**B–F**) Time courses of Ca$^{2+}$ (in nM) bound to buffers (**B**), pumps (**C**), PKC-pathway proteins (**D**), or CaM (**E**), and the concentration of free Ca$^{2+}$ ions (**F**), according to NeuroRD (solid; averaged across eight samples) or NEURON (dashed) simulations. Colours indicate the Ca$^{2+}$ flux used (see **A**). (**B**) Number of Ca$^{2+}$ ions bound to Ca$^{2+}$ buffers, that is immobile buffer and calbindin. (**C**) Number of Ca$^{2+}$ ions bound to Ca$^{2+}$ pumps and exchangers, that is PMCA and NCX. (**D**) Number of Ca$^{2+}$ ions bound to PKC-pathway proteins PLC and PLA2. (**E**) Number of Ca$^{2+}$ ions bound to CaM, in all its forms. (**F**) Cytosolic Ca$^{2+}$ concentration (mM) (**G**) Degrees of activation of different Ca$^{2+}$-binding proteins in a steady state (5 min after onset of Ca$^{2+}$ input) as a function of the magnitude of Ca$^{2+}$ flux. The x-axis shows the amplitude of the Ca$^{2+}$ input (see panel **A**), and the y-axis shows the ratio of the underlying species in a Ca$^{2+}$-bound form over the total number of the proteins. For CaM, only the CaM molecules bound by four Ca$^{2+}$ ions are considered activated — in PLC, PLA2, and DGL, binding of only one Ca$^{2+}$ ion is needed for activation. Here, the measured quantity of active PLC includes both Gq-bound and non-Gq-bound CaPLC. Inset: zoomed-in view on the red area. (**H**) Ratio of the steady-state concentration of PKA catalytic subunit over the theoretical maximum where all PKA molecules were dissociated into residuals and catalytic subunits. Colour of the curve indicates the amplitude of the β-adrenergic ligand flux (particles/ms). (**I**) Fraction of phosphorylated CaMKII subunits. (**J**) Fraction of (transiently or persistently) activated PKC. Colour of the curve indicates the amplitude of the cholinergic and glutamatergic ligand flux (particles/ms). The grey area in panels (**G–J**) represents Ca$^{2+}$ inputs that cause cytosolic Ca$^{2+}$ concentration to reach extremely high levels (>1 mM) that are likely to lead to apoptosis.

PLA2, along with the PKA and CaMKII pathway-related protein CaM became completely activated if large enough Ca$^{2+}$ flux is given, but their degrees of activity varied across the magnitude of the injected Ca$^{2+}$ flux (*Figure 2G*). DGL was most completely activated throughout the Ca$^{2+}$ amplitude, owing to the large equilibrium constant of its Ca$^{2+}$ binding. At extremely large Ca$^{2+}$ fluxes, CaM was more completely bound by Ca$^{2+}$ than PLC and PLA2 (*Figure 2G*), but at lower Ca$^{2+}$ amplitudes, CaM remained largely unbound (*Figure 2G* inset). This is reflected in the activation patterns of the catalytic subunit of PKA (*Figure 2H*) and CaMKII (*Figure 2I*), both of which are dependent on the activation of CaM and thus had a small response at low Ca$^{2+}$ amplitudes. PKC, by contrast, became activated at relatively small Ca$^{2+}$ amplitudes (*Figure 2J*). Of these three pathways, the PKC pathway was dependent on the cholinergic ligands or the activation of the mGluRs (*Figure 2J*), and the PKA pathway was dependent on the availability of β-adrenergic ligands (*Figure 2H*). Taken together, these results highlight the need for large Ca$^{2+}$ flux to the post-synaptic spine for the activation of the CaMKII pathway, relatively large Ca$^{2+}$ flux for the activation of the PKA pathway, and relatively small Ca$^{2+}$ flux for the activation of the PKC pathway.

## High-frequency stimulation (HFS) causes LTP and low-frequency stimulation (LFS) causes LTD in GluR1-GluR2-balanced synapses

The $Ca^{2+}$ flux entering the post-synaptic spine is extremely large during and after synaptic transmission and low otherwise, which causes the signalling pathways to be activated and deactivated in a more dynamic way than described in the previous section ('*$Ca^{2+}$ activates multiple pathways that regulate the post-synaptic plasticity in cortical PCs*'). The activation of these pathways and their dependence on the stimulus protocol are difficult to study experimentally due to methodological constraints (e.g., side effects of fluorescence indicators, lack of signal calibration, and poor temporal or spatial resolution), but biochemically detailed models, such as the one considered in this work, can provide insights into the transient molecular mechanisms behind LTP and LTD. Our model is particularly well suited to study the mechanisms behind CaMKII-, PKA- and PKC-mediated phosphorylation of AMPAR subunits, which are important mediators of long-term plasticity (*Wang et al., 2005*). Phosphorylation of GluR1 subunits at S845 increases the insertion rate of the AMPAR into the membrane, thus leading to post-synaptic LTP (*Diering et al., 2016*). Conversely, phosphorylation of GluR2 subunits at S880 increases the rate of receptor endocytosis from the membrane, and has thus been observed to lead to post-synaptic LTD (*Xia et al., 2000*). However, it is not the number of the membrane-expressed AMPAR subunits alone that determine the strength of the synapse, but different compositions of the subunits have different single-channel conductances, and phosphorylation at S831 of the GluR1 subunit also affects the conductance of the channel (*Oh and Derkach, 2005*).

To simulate the reaction dynamics in the post-synaptic spine under realistic input patterns, we applied the 4xHFS and LFS protocols. Each input contained transient (3 ms) influxes of $Ca^{2+}$ (1900 particles/ms) into the cytosol and glutamate (20 particles/ms), acetylcholine (20 particles/ms) and β-adrenergic ligand (10 particles/ms) into the extracellular subspace near the spine membrane. We used a balanced ratio (1:1) of GluR1 and GluR2 subunits. We recorded the time courses of the concentrations of all CaMKII-, PKA-, and PKC-pathway molecules contributing to LTP or LTD to monitor their activity during and following the stimulation protocols. We also recorded the numbers of membrane-inserted GluR1 and GluR2 and their state of phosphorylation and used *Equation 5* for determining the maximal synaptic conductance.

In the 4xHFS protocol, which typically causes LTP in plasticity experiments, our model predicts a large increase in total synaptic conductance (*Figure 3A*) due to a radical increase in membrane-inserted GluR1 subunits (*Figure 3B*) and a decrease in GluR2 subunits (*Figure 3C*). These changes in membrane-expression of AMPAR subunits were dependent on activations of many signalling proteins in the CaMKII (*Figure 3D–H*), PKA (*Figure 3I–M*), and PKC (*Figure 3N–R*) pathways. First, the $Ca^{2+}$ entry (*Figure 3D*) caused a rapid increase in half-activated calmodulin (bound by two $Ca^{2+}$ ions; *Figure 3E*), leading to a longer-lasting increase in active calmodulin (*Figure 3F*). Calmodulin activation led to an increase in the concentration of phosphorylated CaMKII (*Figure 3G*), which phosphorylated the GluR1-type receptors at S831 (*Figure 3H*). The β-adrenergic input (*Figure 3I*), in turn, bound to the β-adrenergic receptors and activated the Gs proteins (*Figure 3J*), which bound to the adenylyl cyclase AC1 to produce cyclic adenosine monophosphate (cAMP, *Figure 3K*). cAMP bound to PKA to release the catalytic subunits of PKA (*Figure 3L*), which led to a phosphorylation of the GluR1-type receptors at S845 (*Figure 3M*) and thus to increased membrane expression of GluR1 subunits and total synaptic conductance (*Figure 3A–B*). Due to the simultaneous activation of the CaMKII pathway, a significant proportion of double phosphorylated GluR1-type receptors was observed (*Figure 3H,M*). As for the PLC–PKC pathway, glutamate (*Figure 3N*, blue) bound to mGluRs and acetylcholine (*Figure 3N*, green) bound to muscarinic receptors (M1), and the activation of these receptors contributed to the activation of Gq proteins (*Figure 3O*). The activated Gq proteins bound with PLC and metabolised phosphatidylinositol 4,5-bisphosphate (Pip2) into diacylglycerol (DAG, *Figure 3P*), which activated PKC (*Figure 3Q*). This led to the phosphorylation of GluR2-type receptors at S880 (*Figure 3R*), which caused the decrease in membrane expression of GluR2 observed in *Figure 3C*.

The differences between NEURON and NeuroRD simulation results in *Figure 3M* were due to the stochasticity in NeuroRD simulator — both smaller and larger GluR1 phosphorylation levels compared to NEURON simulation results (*Figure 3M*, dashed) were obtained when NeuroRD simulations were run with different random number seeds (not shown).

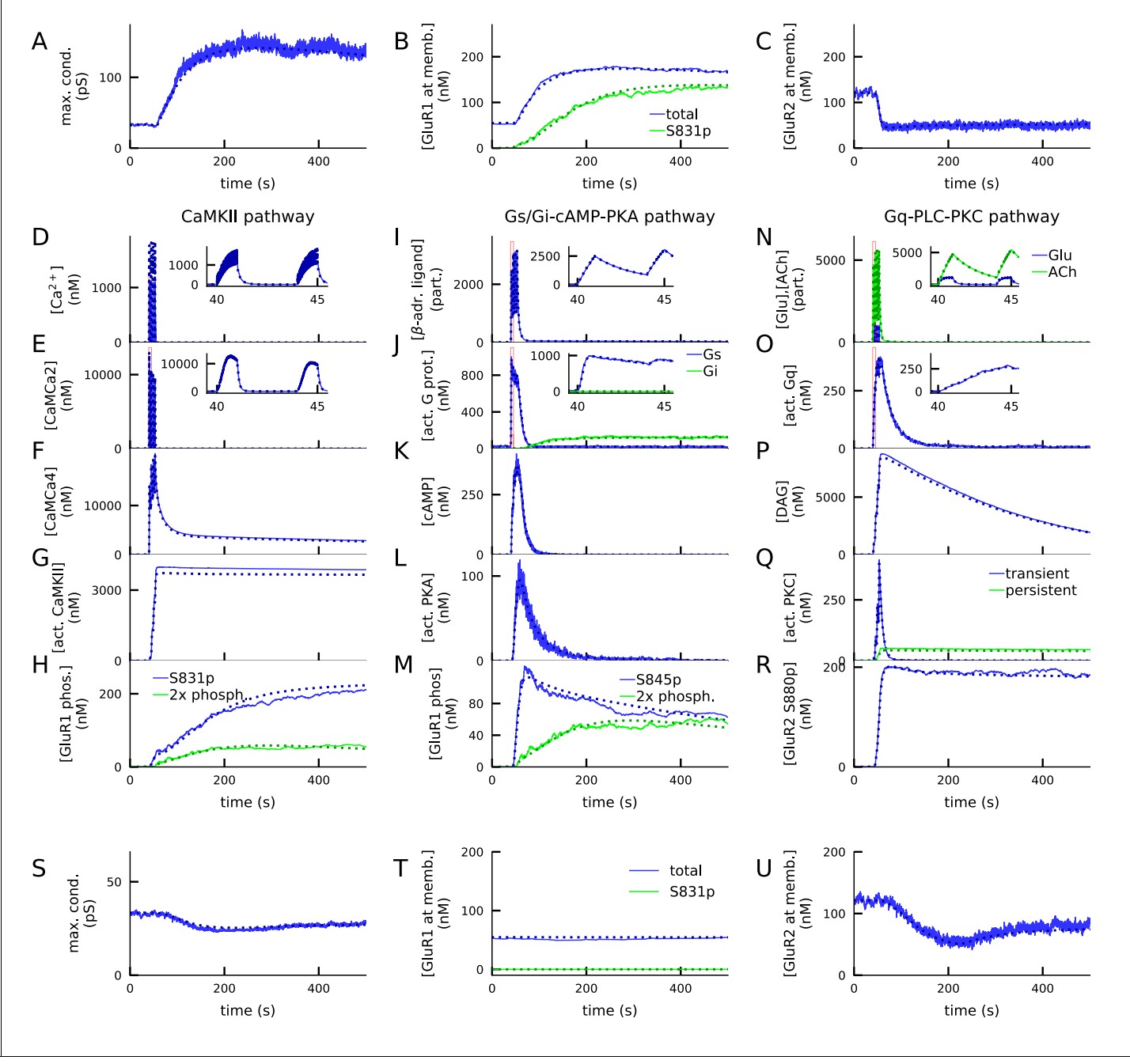

**Figure 3.** 4xHFS activates CaMKII, PKA, and PKC pathways and leads to LTP (A–R), while LFS activates the PKC pathway and leads to LTD (S–U). (A) Total synaptic conductance in response to 4xHFS, determined by the numbers of membrane-inserted GluR1s and GluR2s — see *Equation 5*. The stimulation starts at 40 s and lasts until 53 s. (B–C) Concentration of membrane-inserted GluR1s (B) and GluR2s (C) in response to 4xHFS. (D–H) Concentration of different species in the CaMKII pathway, namely, intracellular unbound $Ca^{2+}$ (D), CaM bound with two $Ca^{2+}$ ions (E), CaM bound with four $Ca^{2+}$ ions (active CaM; F), phosphorylated CaMKII, bound or unbound by CaMCa4 (G), and S831-phosphorylated and double-phosphorylated GluR1 subunits (H) in response to 4xHFS. (I–M) Concentration of different species in the cAMP-PKA pathway, namely, β-adrenergic ligand in all its forms (I), activated (GTP-bound but not bound to ATP) Gs and Gi proteins (J), intracellular cAMP (K), catalytic subunit of PKA (L), and S845-phosphorylated and double-phosphorylated GluR1 subunits (M) in response to 4xHFS. (N–R) Concentration of different species in the PLC-PKC pathway, namely, glutamate and acetylcholine in all their forms (N), activated (GTP-bound but not bound to DAG) Gq proteins (O), intracellular DAG (P), activated PKC (Q), and S880-phosphorylated GluR2 subunits (R) in response to 4xHFS. S: Total synaptic conductance in response to LFS. (T–U) Concentration of membrane-inserted GluR1s (T) and GluR2s (U) in response to LFS, which starts at 40 s and lasts until 220 s. The solid lines represent stochastic (NeuroRD) simulation results, while the dashed lines represent data from deterministic (NEURON RxD) simulations. β-adrenergic ligands, glutamate,

*Figure 3 continued on next page*

*Figure 3 continued*

and acetylcholine are measured in numbers of particles as they reside both at the membrane (when bound to receptors) and at the extracellular subspace near the spine membrane (when unbound); other species measured in concentration.

The online version of this article includes the following figure supplement(s) for figure 3:

**Figure supplement 1.** Both GluR1 and GluR2 are needed for bidirectional plasticity.

**Figure supplement 2.** An alternative dimers-of-like-dimers rule of tetramer formation reproduces the HFS-induced LTP, LFS-induced LTD, and STDP predictions obtained with the default tetramer formation rule.

**Figure supplement 3.** The biochemical signalling network model, given the NMDAR-conducted $Ca^{2+}$ inputs from the multicompartmental neuron model of layer 2/3 pyramidal cell under 1.3 mM extracellular $[Mg^{2+}]$, predicts LTP for 6xHFSt and LTD for LFS-1Hz.

**Figure supplement 4.** The biochemical signalling network model robustly predicts LTP for HFS and LTD for LTP with altered durations of neuromodulatory inputs.

In the LFS protocol, which typically causes LTD in the experiments, our model predicts a prominent (20%) decrease in total synaptic conductance (*Figure 3S*) due to a decrease in GluR2 subunits. In this protocol, the $Ca^{2+}$ inputs are insufficiently large to activate CaM, and the Gs proteins remain deactivated as well (data not shown). In consequence, CaMKII and PKA pathways remain deactivated, and the effect of the LFS protocol on GluR1 phosphorylation and membrane insertion is small (*Figure 3T*). By contrast, the PKC pathway is almost as strongly activated as in the 4xHFS protocol (data not shown), leading to prominent S880 phosphorylation of GluR2 (data not shown) and removal of GluR2 from the membrane (*Figure 3U*).

The expression of both LFS-induced LTD and 4xHFS-induced LTP of these types is dependent on the presence of both GluR1 and GluR2 subunits: GluR1-deficient synapses failed to show 4xHFS-induced LTP (*Figure 3—figure supplement 1A*) and GluR2-deficient synapses failed to show LFS-induced LTD (*Figure 3—figure supplement 1B*). To show that our results were not an artefact of the tetramer formation rule (*Equation 1–5*), we applied an alternative tetramer formation rule where GluR1 and GluR2 subunits randomly dimerised and the dimers paired with like dimers (which disallows the emergence of heterotetramers with 1:3 or 3:1 proportion of GluR1:GluR2 subunits; *Gan et al., 2015*). We reproduced the LFS-induced LTD and 4xHFS-induced LTP using this dimer-of-like-dimers rule with a modified (35:65) balance of GluR1 vs. GluR2 subunits (*Figure 3—figure supplement 2A*).

In the above analyses, we used brief square-pulse fluxes of $Ca^{2+}$ to the synapse model, which is a simple representation of inputs during synaptic plasticity induction protocols. Alternatively, $Ca^{2+}$ current entering the post-synaptic spines can be estimated by using multicompartmental biophysically detailed neuron models. We simulated a model of layer 2/3 pyramidal cell, stimulated with synaptic inputs from a 6xHFSt or LFS-1Hz protocol (see Materials and methods, section '*Modelling the $Ca^{2+}$ inputs and neuromodulatory inputs*'), to determine the $Ca^{2+}$ inputs entering the post-synaptic spine through NMDA receptors (NMDARs). In accordance with *Figure 3* and experimental data from somatosensory cortex (*Heusler et al., 2000*), our model predicted that 6xHFSt induced LTP whereas LFS-1Hz induced LTD (*Figure 3—figure supplement 3*). Here, the 6xHFSt protocol was used instead of 4xHFS to model the same protocol as in *Heusler et al., 2000*; our model would also predict an LTP for 4xHFS (data not shown). The HFS-induced LTP and LFS-induced LTD of *Figure 3* could also be reproduced with alternative durations of neuromodulator inputs, including 10 min bath applications (*Figure 3—figure supplement 4*). These results indicate that our model can reproduce HFS-induced LTP and LFS-induced LTD also when using realistic NMDAR-conducted $Ca^{2+}$ transients and that these forms of plasticity are robust to the temporal profile of the neuromodulatory inputs.

The activations of the above pathways are dependent on the magnitude and dynamics of the inputs to the model, namely, $Ca^{2+}$, β-adrenergic and cholinergic ligands, and glutamate. All pathways leading to GluR1 and GluR2 phosphorylation and the consequent exocytosis and endocytosis are $Ca^{2+}$-dependent: blocking $Ca^{2+}$ entry completely abolished 4xHFS-induced LTP (*Figure 4A*) that followed GluR1 insertion (*Figure 4B*) and GluR2 endocytosis (*Figure 4C*). Blocking β-adrenergic ligands abolished the 4xHFS-induced LTP (*Figure 4A*) by suppressing the membrane-insertion of GluR1 (*Figure 4B*), but had no effect on GluR2 endocytosis (*Figure 4C*). Likewise, blocking β-adrenergic ligands had no effect on LFS-induced LTD (not shown). In contrast, LFS-induced LTD (*Figure 4E*) that followed GluR2 endocytosis (*Figure 4G*) was reduced by blockade of mGluR activation while the number of GluR1 subunits at the membrane remained unaffected (*Figure 4F*). This

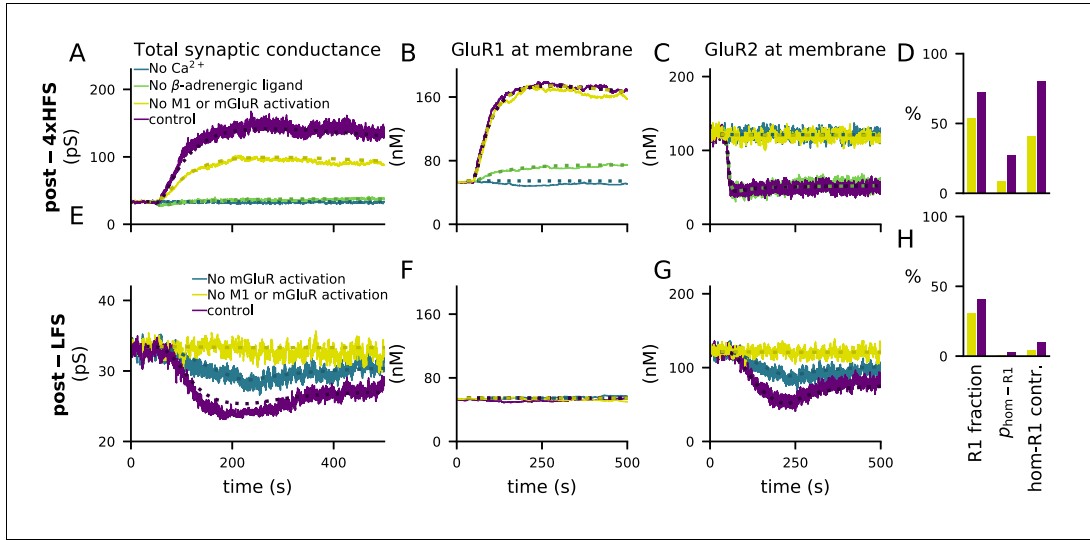

**Figure 4.** 4xHFS-induced LTP is dependent on β-adrenergic ligands and LFS-induced LTD is dependent on activation of mGluRs or cholinergic receptors. (**A–D**) 4xHFS-induced LTP in the control case (dark purple), without $Ca^{2+}$ inputs (blue), without β-adrenergic ligands (green), and under blockade of PKC pathway-activation (mGluRs or cholinergic receptors; yellow). (**E–H**) LFS-induced LTD in the control case (dark purple), under the blockade of mGluR activation (blue), and under blockade of both mGluRs or cholinergic receptors (yellow). (**A, E**) Total synaptic conductance. (**B, F**) Membrane expression of GluR1. (**C, G**) Membrane expression of GluR2. (**D, H**) The fraction of membrane-inserted GluR1 over all membrane-inserted GluR subunits (left), the probability of an AMPAR tetramer being homomeric GluR1 (middle), and the relative contribution of homomeric GluR1 subunits to the total conductance (i.e., summed conductance of homomeric GluR1 tetramers divided by the summed conductance of all tetramers; right). The bars represent the values at the end of the 4xHFS (**D**) or LFS (**H**) simulation with (dark purple) and without (yellow) PLC-activating ligands.

reduction was strengthened by simultaneous blockade of cholinergic inputs (*Figure 4E–G*, yellow traces). Counterintuitively, blocking mGluR and M1-receptor activation also reduced the amplitude of the 4xHFS-induced LTP (*Figure 4A*) by disabling GluR2 endocytosis (*Figure 4C*) while it had no effect on GluR1 insertion (*Figure 4B*). The reason for this is that in the PKC pathway-blocked case there is a smaller post-4xHFS membrane-bound GluR1 ratio (fraction of GluR1 subunits over all GluR subunits at the membrane) than in the control case, and thus the probability of AMPARs being homomeric GluR1 tetramers (which had a very large conductance compared to other tetramers; *Equation 5*) is much smaller in the former case than in control (*Figure 4D*). Although qualitatively similar difference can be observed in post-LFS membrane-bound GluR1 ratios between PKC pathway-blocked case and control, the probability of homomeric GluR1 tetramers and their contribution to the synaptic conductance are very small in both cases (*Figure 4H*) and thus the LFS-induced LTD is not affected.

Taken together, our results show that cortical synapses expressing both GluR1 and GluR2 subunits can express a frequency-dependent form of post-synaptic plasticity (LTP for high-frequency inputs, LTD for low-frequency inputs) that is gated by neuromodulators affecting the PKA and PKC pathways. Our findings also lend support to that GluR2 endocytosis may lead to either potentiation (*Figure 4A*) or depression (*Figure 4E*), depending on the prevalence of the GluR1 subunits at the membrane.

## Paired pre- and post-synaptic stimulation induces PKA- and PKC-dependent spike-timing-dependent plasticity (STDP) in GluR1-GluR2-balanced synapses

Cortical synapses typically exhibit a type of synaptic plasticity, namely STDP, that is dependent on both the pre- and post-synaptic activity. According to a classical model, the differences in the outcome of STDP for different pairing intervals of pre- and post-synaptic stimulus are explained by different amount of $Ca^{2+}$ entering the post-synaptic spine, which is affected by both the pre-

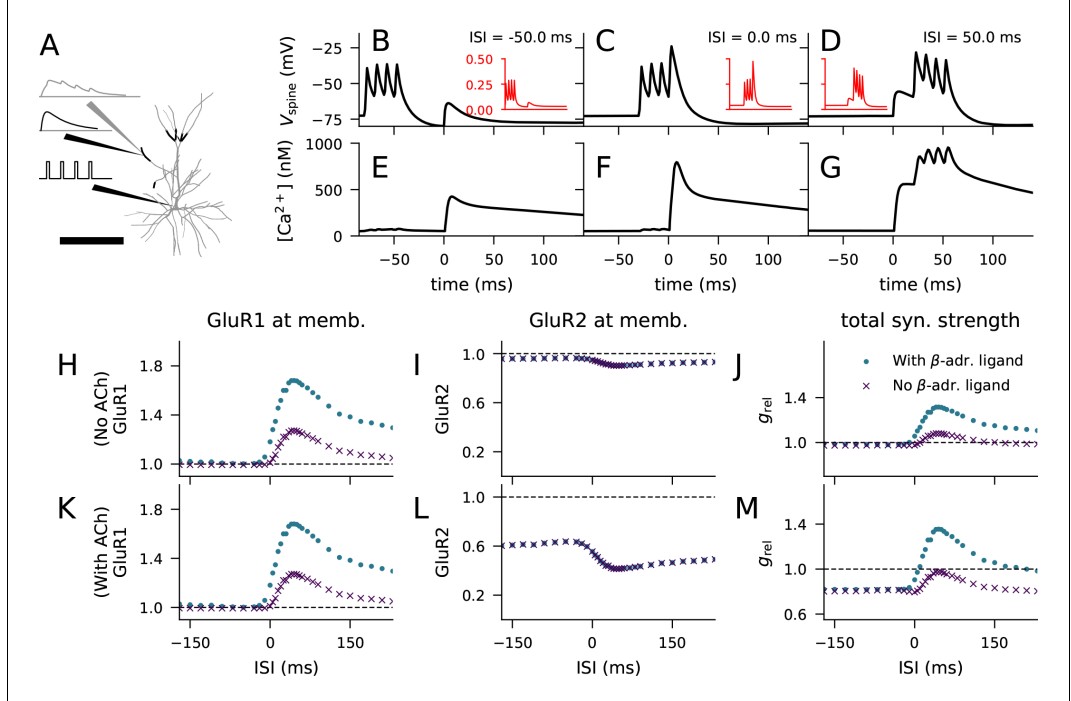

**Figure 5.** Layer 2/3 pyramidal cell plasticity in response to STDP protocol depends on neuromodulatory state and pairing interval. (A) Layer 2/3 pyramidal cell morphology (grey, thin), locations of synaptic input highlighted (black, thick). Inset: Illustration of the inputs (black) and the recorded synaptic intracellular $Ca^{2+}$ (grey). Scale bar 200 µm. (B–D) Membrane potential at the dendritic spine when the pre-synaptic stimulation onset is 50 ms after (B), at the same time as (C), or 50 ms prior to (D) the onset of the last somatic stimulus. Inset (red): $Mg^{2+}$-gate variable as a function of time, ranging from −80 ms to 140 ms in a similar manner as the data in the main panel. (E–G) Concentration of free $Ca^{2+}$ in the dendritic spine according to the biochemical spine model when the pre-synaptic stimulation onset is 50 ms after (B), at the same time as (C), or 50 ms prior to (D) the onset of the last somatic stimulus. (H–J) No LTD was induced by the stimulation protocol (1 Hz paired with post-synaptic stimulation for 2 min) in the absence of M1-receptor activation, but pairing-interval-dependent LTP was induced in presence of β-adrenergic inputs. (K–M) Pairing-interval-dependent LTD was induced when the synaptic input was coupled with cholinergic inputs, and STDP was induced when both cholinergic and β-adrenergic inputs were present. (H, K) Relative concentration of GluR1 at the membrane 16 min after the stimulation onset (normalised by concentration of membrane-inserted GluR1 at rest). (I, L) Relative concentration of GluR2 at the membrane 16 min after the stimulation onset (normalised by concentration of membrane-inserted GluR2 at rest). (J, M) Relative synaptic conductance (*Equation 5*) 16 min after the stimulation onset (normalised by synaptic conductance at rest).

The online version of this article includes the following figure supplement(s) for figure 5:

**Figure supplement 1.** $Ca^{2+}$ fluxes predicted by the multicompartmental layer 2/3 pyramidal cell model depend on the inter-stimulus interval (ISI).

synaptically released glutamate and the elevation of post-synaptic membrane potential. Biophysically detailed neuron modelling offers a powerful tool for determining the size of these $Ca^{2+}$ inputs as a function of the pairing interval.

We considered the LTP/LTD response to paired stimulation protocol using a multicompartmental model of a layer 2/3 pyramidal cell (*Figure 5A*; *Markram et al., 2015*). We placed a synaptic spine with volume 0.5 µm³ at a random location on the apical dendrite, 250–300 µm from the soma (*Figure 5A*, thick, black branches), and stimulated the head of the spine with glutamatergic synaptic currents (*Hay and Segev, 2015*; *Markram et al., 2015*; *Figure 5A*, black traces, top). In parallel, we stimulated the soma with a burst of four short (2 ms) supra-threshold square-pulse currents (*Figure 5A*, black traces, bottom). Given that approximately 10% of the NMDAR-mediated currents and none of the AMPAR-mediated currents are conducted by $Ca^{2+}$ flux, we could determine the number of $Ca^{2+}$ ions entering the spine at each time instant following the onset of the synaptic input (*Figure 5A*, grey traces). This experiment was repeated using different inter-stimulus intervals (ISI) between the synaptic and somatic stimuli and averaged across $N_{samp} = 200$ trials. The membrane potential dynamics at the post-synaptic spine depended on the ISI (*Figure 5B–D*), largest effects response being obtained with near-coincident stimuli (*Figure 5C*). The higher the membrane

potential at the spine, the higher the value of the variable describing the $Mg^{2+}$-gate opening in the NMDA receptor (**Figure 5B–D**, insets) (**Hay and Segev, 2015**; **Markram et al., 2015**). Thus, the $Ca^{2+}$ flux time course varied across the pairing ISIs (**Figure 5—figure supplement 1**). These $Ca^{2+}$ flux time series were imported into our biochemical model ($Ca^{2+}$ transients in the spine model showed in **Figure 5E–G**), which allowed us to predict the magnitude of GluR subunit phosphorylation and membrane insertion for each pairing interval. When added as bath application, the β-adrenergic and cholinergic ligands were simulated by prolonged injections of 50 particles/s for 10 min, starting 8 min before the STDP protocol — these neuromodulators alone (without the electric stimulation-mediated $Ca^{2+}$ inputs) did not cause synaptic plasticity. Throughout the experiments, the activation of mGluRs was blocked.

We first confirmed that the membrane expression of the glutamate receptors was not strongly affected by paired synaptic and somatic stimulation in the absence of β-adrenergic (which activates the PKA pathway) and cholinergic (which activates the PKC pathway) neuromodulation. Our model predicted that there is little change in the membrane expression of GluR1 and GluR2 type receptor subunits in this stimulation protocol (**Figure 5H–I**, purple). Consequently, our model reproduced the observation (**Seol et al., 2007**) that this stimulation protocol led to little change in predicted synaptic conductance (**Figure 5J**, purple).

We next considered the paired synaptic-somatic stimulation in the presence of β-adrenergic ligand. Our model predicted a prominent (up to 70%) increase in GluR1 membrane expression with little effect on GluR2 membrane expression (**Figure 5H–I**, blue). The predicted increase in GluR1 membrane expression (**Figure 5H**) and the consequent increase in synaptic conductance (**Figure 5J**, blue) were most prominent when the ISI was around 20–80 ms, and modest for large ISIs. These predictions are consistent with the experiments where an ISI-dependent potentiation of the EPSCs in the presence of β-adrenergic receptor agonists and absence of cholinergic agonists was observed (**Seol et al., 2007**).

When β-adrenergic neurotransmission was blocked but M1 receptors were activated by cholinergic ligands, the model predicted a prominent (up to 60%) decrease in GluR2 membrane expression, with little effect on GluR1 membrane expression (**Figure 5K–L**, purple). Our model of synaptic conductance (**Equation 5**) predicted a decrease in total conductance in a GluR1-GluR2-balanced synapse for this condition (**Figure 5M**, purple), which is in line with the experimental data (**Seol et al., 2007**). The depression takes place throughout the tested ISIs, but the effect was smallest for ISIs very close to zero due to the counteracting effects of GluR1 membrane-insertion (**Figure 5K**, purple). Finally, when both β-adrenergic and cholinergic neurotransmission were active, our model predicted an increased GluR1 membrane expression and decreased GluR2 membrane expression, both of which were ISI dependent (**Figure 5K–L**, blue). In these simulations, the predicted synaptic conductance was increased for small and moderate pre-post intervals and decreased otherwise (**Figure 5M**, blue), which is qualitatively similar to experimental data (**Seol et al., 2007**). These results are dependent on the availability of both GluR1 and GluR2 subunits at the post-synaptic spine: in simulations where GluR1 or GluR2 subunits were absent, only LTD (**Figure 3—figure supplement 1C**) or LTP (**Figure 3—figure supplement 1D**), respectively, was induced by the STDP protocol. In a similar manner as the HFS- and LFS-induced plasticity in **Figure 3—figure supplement 2A**, we could reproduce the STDP using the dimer-of-like-dimers tetramer formation rule with a GluR1 fraction of 35% (**Figure 3—figure supplement 2B**). Taken together, our model with balanced numbers of GluR1 and GluR2 subunits reproduces the neuromodulator-gated STDP observed in layer 2/3 pyramidal cells of the visual cortex.

The combination of our biochemically detailed model with the biophysically detailed model of layer 2/3 pyramidal cell model provides a compelling means of hypothesis testing for cortical STDP in this cell type. We analyzed how the shape of the STDP curve of **Figure 5M** is affected by the number of spikes in each post-synaptic burst stimulus. Our simulations suggest that decreasing the number of spikes per burst decreases the amplitude of both LTP and LTD in the STDP protocol and, in particular, brings the LTD for large post-pre ISIs close to zero (**Figure 6A**). These alterations are mediated by changes in both the level of membrane-insertion of GluR1 and endocytosis of GluR2 subunits (**Figure 6A**, insets). For small and moderate pre-post ISIs, the effects of decreasing the number of post-synaptic stimuli on the STDP curve are expected: both GluR1 insertion and GluR2 endocytosis are of smaller amplitude, and hence the dampened LTP amplitude (**Figure 6A**). By contrast, for post-pre ISIs and large pre-post ISIs, decreasing the number of post-synaptic stimuli results

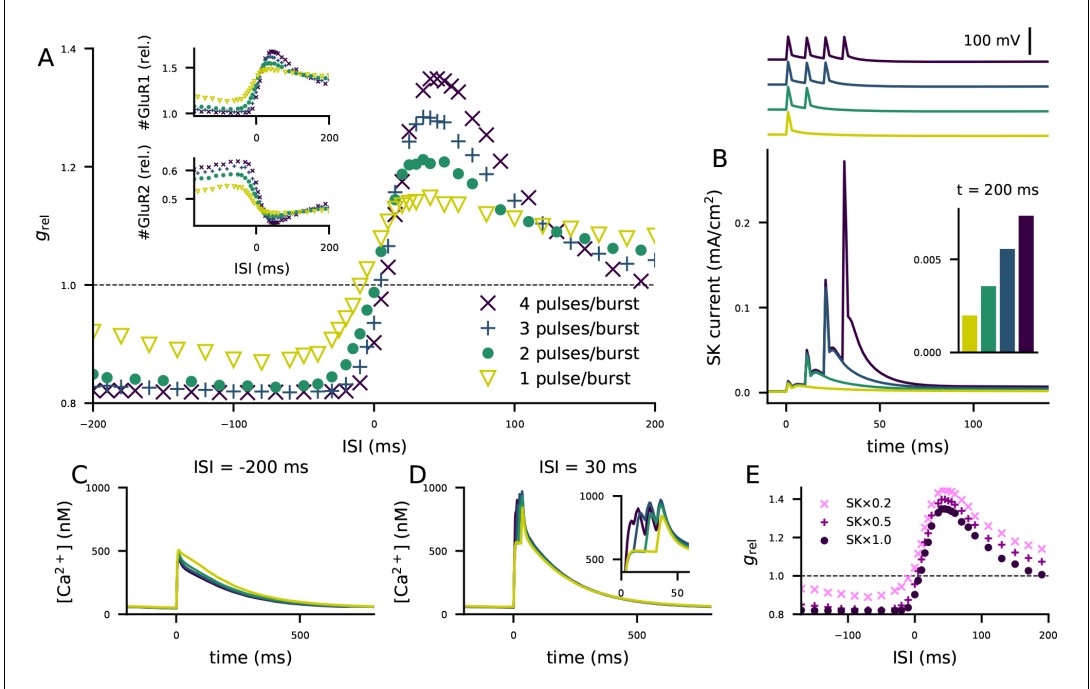

**Figure 6.** The STDP curve of layer 2/3 pyramidal cells is affected by the number of post-synaptic stimulus pulses associated with the pre-synaptic input. (A) The STDP curves of *Figure 5M* when the number of spikes per post-synaptic burst was 1 (yellow), 2 (green), 3 (blue), or 4 (as in *Figure 5*; dark purple). Inset: relative concentrations of membrane-inserted GluR1 (top) or GluR2 (bottom) subunits — see *Figure 5K–L* for reference. (B) Top: somatic membrane potential time course (aligned according to the onset of the first stimulus) for different numbers of post-synaptic stimulus pulses. Bottom: somatic SK current-density time course in the four conditions. Inset: the SK current densities 200 ms after the onset of the first post-synaptic stimulus. (C–D) $Ca^{2+}$ flux to the dendritic spine when the pre-synaptic stimulation onset is 200 ms after (C) or 30 ms before (D) the onset of the last post-synaptic stimulus. (E) The STDP curves of *Figure 5M* when the number of spikes per post-synaptic burst was four but the somatic SK conductance parameter was either normal (dark purple), 50% smaller (magenta), or 80% smaller (pink).

The online version of this article includes the following figure supplement(s) for figure 6:

**Figure supplement 1.** The post-STDP synaptic conductance is weakly correlated with the peak of the $Ca^{2+}$ input but strongly correlated with the mean $Ca^{2+}$ input during the inter-stimulus interval.

in larger amplitude of GluR1 insertion and GluR2 endocytosis, which yields a dampened LTD for post-pre ISIs and strengthened LTP for large pre-post ISIs (*Figure 6A*). These counter-intuitive effects can be explained by the accumulation of small-conductance K+ (SK) conductance in the post-synaptic neuron: the larger the number of post-synaptic pulses in the pairing burst, the larger the SK currents (*Figure 6B*). The SK current decays slowly (matching the $Ca^{2+}$ concentration decay), and remnants of the SK currents can be observed as long as 200 ms after the post-synaptic stimulus (*Figure 6B* inset). For large post-pre ISIs, the number of spikes per burst has little effect on the $Ca^{2+}$ transients during the post-synaptic stimulation (*Figure 6C* inset), but the SK currents activated by a large number of spikes per burst contribute to significantly decrease the $Ca^{2+}$ transient caused by the pre-synaptic stimulus during the decay period (*Figure 6C*). By contrast, for small pre-post inter-vals, the additional spikes in the post-synaptic stimulus significantly contribute to the $Ca^{2+}$ transients (*Figure 6D*). To show that the effects of the number of spikes per post-synaptic burst are mediated by the SK current, we ran the simulation of *Figure 5J* using a partial to complete blockage of the SK currents in the biophysically detailed simulations of the layer 2/3 pyramidal cell. The paired-pulse protocol of *Figure 5M* (involving both β-adrenergic and cholinergic neuromodulation) caused an STDP in all cases, but decreasing the SK conductance shortened the post-pre LTD window and decreased the amplitude of LTD (*Figure 6E*). Similar effects were obtained with a decrease of $Ca^{2+}$-channel conductances (not shown), which is in agreement with the data of ***Nevian and Sakmann, 2006***. Our model predictions also agree with the observation that the plasticity outcome is not determined by $Ca^{2+}$ transient amplitude (***Nevian and Sakmann, 2006***), instead, our model suggests

that the total Ca²⁺ is a better predictor of the plasticity outcome: the correlation coefficient between the post-STDP synaptic conductance and the peak Ca²⁺ transient amplitude (see *Figure 5—figure supplement 1E*) was 0.53, while that between the post-STDP synaptic conductance and the mean Ca²⁺ input during the inter-stimulus interval (see *Figure 5—figure supplement 1F*) was 0.96 (*Figure 6—figure supplement 1*).

## The model predicts multimodal, protein concentration- and neuromodulation-dependent rules of plasticity

Cortical neurons express a variety of forms of LTP/LTD depending on the brain region and cell type. In computational studies, neocortical plasticity is most typically described by simple rules according to which small-amplitude Ca²⁺ inputs lead to depression of the synapse whereas large-amplitude inputs lead to potentiation. Apart from a few examples *Castellani et al., 2001*; d'*D'Alcantara et al., 2003*; *Castellani et al., 2005*; *Honda et al., 2013*, these models typically do not describe the intracellular signalling machinery leading to the resulting plasticity *Holthoff et al., 2002*; *Karmarkar et al., 2002*; *Badoual et al., 2006*; *Cornelisse et al., 2007*; *Kubota and Kitajima, 2008*; *Urakubo et al., 2008*. Unlike biochemically detailed models, the simple models cannot be used to explore whether and how the prevalence of different plasticity-related proteins gives rise to various types of LTP/LTD or their impairments, which is an important question in the study of mental disorders with deficits in cortical plasticity. Here, we analysed the biochemical underpinnings of different types of plasticity rules using our unified model of cortical plasticity in order to predict the conditions for different forms of plasticity.

In a similar fashion to section '*Ca²⁺ activates multiple pathways that regulate the post-synaptic plasticity in cortical PCs*', we simulated our model of the post-synaptic spine when stimulated with a prolonged (5 min) square-pulse influx of Ca²⁺ and neuromodulators. We randomly altered the model parameters controlling the initial concentrations of different proteins, namely, the ratio of GluR1 to all GluR subunits (i.e., $\frac{[\text{GluR1}]_{\text{total}}}{[\text{GluR1}]_{\text{total}}+[\text{GluR2}]_{\text{total}}}$, from here on referred to as GluR1 ratio), the concentration of NCX (regulating the rate of Ca²⁺ decay from the spine), and the concentrations of PKA-pathway and PKC-pathway proteins (upstream of PKA and PKC). Alterations of the initial concentration of CaMKII (the only molecule in our model that exclusively affects the CaMKII pathway) had little effect in most domains of plasticity considered here (not shown), and thus, we omitted it in this analysis. We sampled these parameters from the following intervals: GluR1 ratio from the interval from 0 to 1 (keeping the total concentration of GluR subunits fixed at 540 nM), NCX concentration from the interval from 0 to twice the original value (2 × 0.54 mM), and the PKA and PKC-pathway factors $f_{\text{PKA}}$ and $f_{\text{PKC}}$ from the interval from 0 to 2 (see Materials and methods, section '*Parameter alterations and model fitting*'). We simulated the post-synaptic spine 150,000 times using different random values for these parameters under zero, low (50 particles/ms), medium (150 particles/ms), and high (250 particles/ms) levels of Ca²⁺ input.

We classified the parameter sets based on the total synaptic conductance 15 min after the onset of the stimulation with the high Ca²⁺ flux (250 particles/ms): the relative synaptic conductance varied between 0.16 and 5.92, and thus, we grouped the parameter sets to 16 classes using a bin size of 0.36 (*Figure 7A*). We then analysed the parameter distributions and their co-variations within these classes and how the different parameters affected the shape of the LTP/LTD curve within each class. A special subset of the LTP/LTD curves were the BCM-type plasticity curves, where either 50 or 150 particles/ms Ca²⁺ injection resulted in LTD and the 250 particles/ms resulted in LTP.

Our model with the standard protein concentrations (GluR1 ratio 0.5, [NCX] = 0.54 mM, $f_{\text{PKA}} = f_{\text{PKC}} = 1.0$) produced a BCM-type curve in class 6 (*Figure 7A*, black dashed curve). Classes 11–16 exhibited the strongest LTP, with large synaptic conductance for both 150 and 250 particles/ms Ca²⁺ injection, whereas classes 1 and 2 only exhibited LTD (*Figure 7A*). Classes 3–12 exhibited BCM-type of plasticity but the majority of the LTP/LTD curves were of non-BCM type in each class (*Figure 7A*).

Three parameters — the GluR1 ratio, NCX concentration and $f_{\text{PKA}}$, differed significantly across the 16 classes (*Figure 7B–D*). Low GluR1 ratio was needed for strong LTD and medium or high GluR1 ratio for strong LTP (*Figure 7B*). However, the strongest forms of LTP (classes 11–16) were induced only when GluR1 ratio was smaller than 1 (*Figure 7B*), because a very low number of GluR2 subunits implied that the synapse has many homomeric GluR1 tetramers at a basal state, and thus

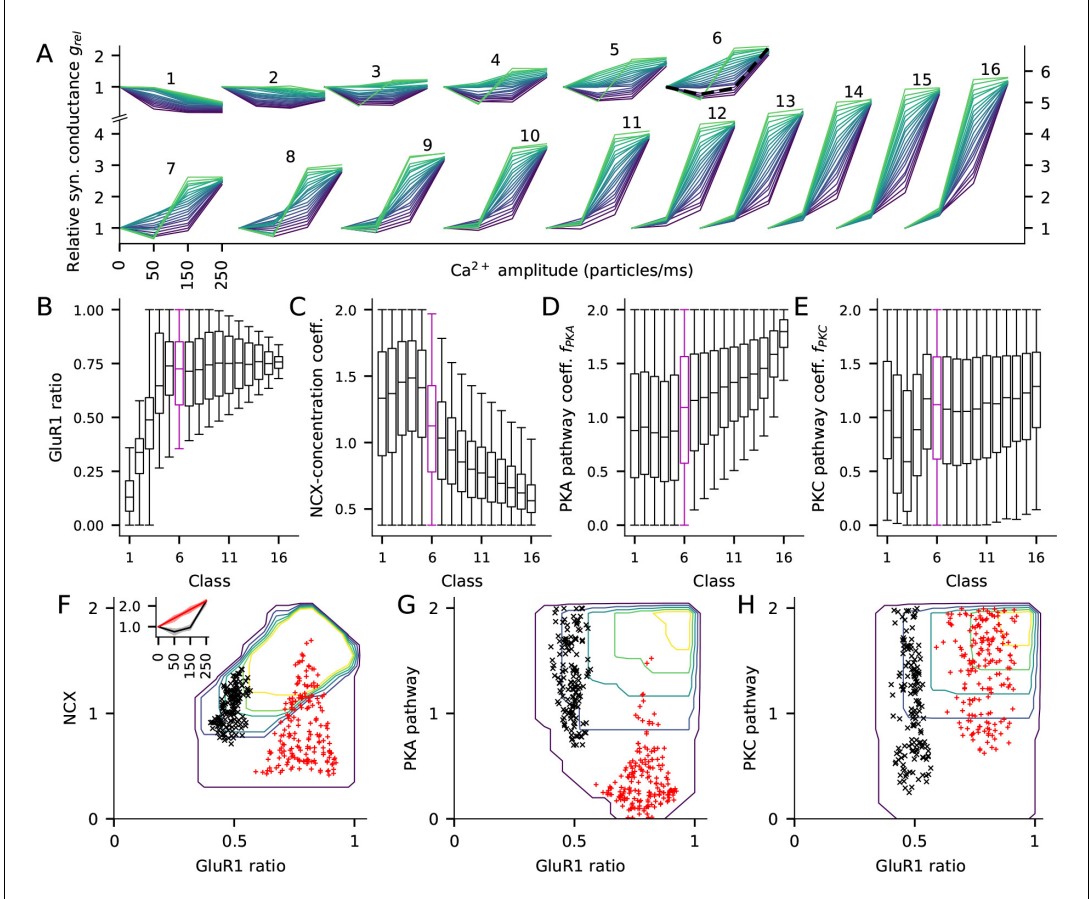

**Figure 7.** The fraction of GluR1s, number of Ca²⁺ extrusion proteins, and the concentrations of PKA and PKC-pathway proteins in the post-synaptic spine determine the type of LTP/LTD in the post-synaptic spine. (**A**) The LTP/LTD curves for all 16 classes. Four values of Ca²⁺ input amplitude were considered: 0, 50, 150, and 250 particles/ms (x-axis; repeated and overlaid for space). The y-axis shows the relative synaptic conductance, that is, total synaptic conductance 15 min after the onset of the Ca²⁺ input divided by the total synaptic conductance before the Ca²⁺ input. 20 representative parameter sets are displayed from each class, coloured from purple (lowest relative synaptic conductance response for medium Ca²⁺ input) to green (highest conductance). The black, dashed trace in class six represents the model with the default concentration parameters. (**B–E**) Distribution of model parameters, that is, GluR1 ratio (**B**), NCX-concentration coefficient (**C**), PKA pathway-concentration coefficient $f_{PKA}$ (**D**), and PKC pathway-concentration coefficient $f_{PKC}$ in the 16 classes. Class 6 (purple) highlighted for further analysis. F–H: GluR1 ratio plotted against NCX-concentration coefficient (**F**), $f_{PKA}$ (**G**), and $f_{PKC}$ (**H**) in class 6. The contours represent the distribution of parameters (N = 5837) that produced class-6 plasticity. No parameters yielding class-6 plasticity were found beyond the purple contour, and the inner contours cover the parameter space where the distribution is higher than 0%, 20%, 40%, 60% or 80% of the maximal density value. The black and red markers represent parameter sets that produced two plasticity subclasses, namely, one where the total deviance (summed absolute difference) from the BCM-type plasticity produced by the default parameter set (black, N = 145) or from a linearly increasing LTP (red, N = 183) was less than 0.2 (a.u.). Inset: The LTP/LTD plasticity curves of the two subclasses. The thick lines represent the centre of the subclasses (black: relative conductances in response to 50, 150, and 250 Ca²⁺ ions/ms: 0.76, 0.96, 2.24; red: relative conductances in response to 50, 150, and 250 Ca²⁺ ions/ms: 1.41, 1.83, 2.24).

The online version of this article includes the following figure supplement(s) for figure 7:

**Figure supplement 1.** The PKC-pathway parameter distributions differ between clusters separated by their response to low (50 particles/ms) Ca²⁺ input.

---

stimulation-induced GluR1 exocytosis and GluR2 endocytosis did not radically increase the number of homomeric GluR1 tetramers (**Equation 5**, see also **Figure 4D**). For LTD and moderate LTP (classes 1–5), any NCX concentration and PKA-pathway coefficient could be used, but very strong LTP (classes 10–16) required a small to medium NCX concentration (**Figure 7C**) and a medium to large PKA-pathway coefficient (**Figure 7D**). By contrast, PKC-pathway coefficient alone was not predictive of plasticity outcome (**Figure 7E**).

The model results for the large parameter distributions of **Figure 7B–E** imply that there are manifestly different combinations of parameters that lead to the same LTP/LTD outcome. To analyse this

intrinsic variability, we studied the distributions of the model parameters within the class of moderate LTP (class 6, 132–168% LTP for 250 particles/ms; indicated by purple boxes in *Figure 7B–E*) in more detail. Dependencies among the four parameters could be observed in 2-dimensional contour plots of the parameter distributions (*Figure 7F–H*). With large (0.6) GluR1 ratios, any NCX, PKA or PKC concentration could be used, but with smaller ($\lesssim$ 0.6) GluR1 ratios, smaller NCX concentration (*Figure 7F*) or larger PKA-pathway coefficients (*Figure 7G*) were needed to obtain class-6 type of plasticity. To illustrate how these parameters affect the shape of the plasticity curves within class 6, we plotted the parameter sets that produced a BCM-type LTP/LTD curve similar to the one produced by our default model (*Figure 7F* inset, black) or an LTP curve that was linear within this regime (*Figure 7F* inset, red). Moderate GluR1 ratios (0.40–0.57; *Figure 7F*, black) and moderate NCX concentrations (0.7–1.4 times the default value; *Figure 7F*, black) were needed for the default BCM-type plasticity, while for the linear LTP curve a larger GluR1 ratio (0.59–0.92; *Figure 7F*, red) was needed but the NCX concentration was more variable (values ranged from 0.4 to 1.7 times the default value; *Figure 7G*, red). The PKA pathway coefficients were generally larger in the default BCM-type plasticity parameter sets than in the parameter sets producing the linear LTP curve (*Figure 7G*). *Figure 7H* shows the distributions of a set of coefficients, i.e. the PKC pathway, which were not correlated with the plasticity outcome within this group.

Our previous analysis showed that PKC-pathway-mediated GluR2 endocytosis was important in lower stimulation frequency protocols (*Figure 3*) or in protocols with large separation between pre- and post-synaptic stimuli (*Figure 5*). To further analyze the contribution of PKC-pathway proteins to plasticity outcomes, we repeated the analysis of *Figure 7* by clustering the plasticity outcome based on the relative synaptic strength after a steady-state $Ca^{2+}$ input of low amplitude (50 particles/ms; *Figure 7—figure supplement 1A*). As observed with the previous clustering, the GluR1 ratio and NCX concentration differed across classes (*Figure 7—figure supplement 1B and C*). However, in this classification, the PKA-pathway coefficient was not predictive of the plasticity outcome (*Figure 7—figure supplement 1D*) whereas the PKC-pathway coefficient varied across the classes (*Figure 7—figure supplement 1E*). Separation between BCM-like plasticity and gradual LTD was also evident within class 6', and due to the same GluR1, PKA and NCX parameters as with the original classification (*Figure 7—figure supplement 1F–H*). This shows our identification of critical parameters is robust to how the classification was performed.

Taken together, our results show that alterations of the concentrations of the proteins regulating $Ca^{2+}$ efflux or PKA/PKC-pathway signalling and the numbers of GluR1 and GluR2 subunits, ranging from complete absence to moderate increase (±100%), have a large effect both on the type of plasticity (LTP or LTD) and on the sensitivity of the plasticity outcome to the amplitude of the $Ca^{2+}$ flux. These data suggest that neocortical post-synaptic spines may exhibit a vast set of plasticity rules by downregulation or relatively mild upregulation of their protein expression.

## A parametric analysis confirms the robustness of the model

We analysed the model responses to 4xHFS and LFS protocols (as in *Figure 3*) under small (±10%) changes in the parameters describing the initial concentrations and reaction rates (*Figure 8*). As expected, most parameter changes led to small deviations from the predicted magnitudes of LTP/LTD (*Figure 8*, grey bars). Alterations of the initial concentration of a number of species (10 out of 47) and reaction rates (12 out of 223) resulted in a notable (>15%) amplification or attenuation of LTD (*Figure 8A*) or LTP (*Figure 8B*). The parameters affecting the LFS-induced LTD were all related to GluR1 membrane insertion or total amount of GluR1 or GluR2 (*Figure 8A*), while the parameters affecting the 4xHFS-induced LTP were related to NCX-mediated $Ca^{2+}$ extrusion, PP1 concentration, production of cAMP by AC1, PKA buffering/deactivation, or GluR1 membrane insertion (*Figure 8B*). Importantly, none of the parameter changes completely abolished the LTP or LTD. Taken together, our model is robust to small alterations in initial concentrations and reaction rates, but parameters influencing the $Ca^{2+}$ dynamics, GluR1 activity, or the PKA-pathway signalling can have relatively large effects on the model output.

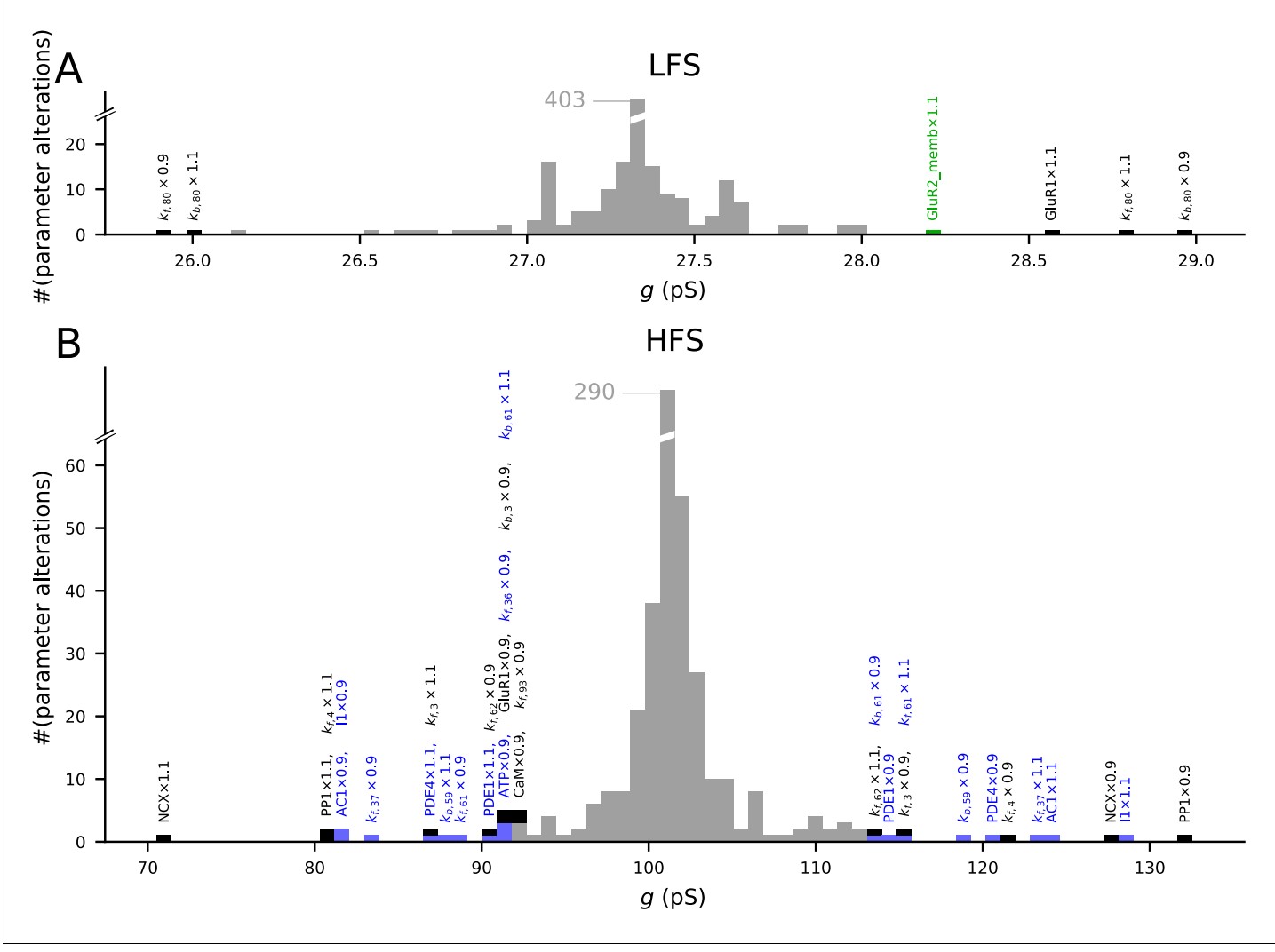

**Figure 8.** The model predictions of LTP and LTD are robust to small changes in model parameters. Values of initial concentrations (47 parameters) or reaction rates (223 parameters) were changed one at the time by −10% or +10%, and the resulting synaptic conductance 16 min after LFS (**A**) or 4xHFS (**B**) protocol was measured (NEURON RxD simulations). The initial synaptic conductance is 33.4 pS (see *Figure 3A,S*), although some parameter changes mildly affected this value (data not shown). The x-axis shows the post-LFS (**A**) or post-HFS (**B**) synaptic conductance, and the y-axis shows the number of parameter alterations. Majority of the parameter changes had small effect on plasticity (grey bars), but changes in initial concentrations of 10 species and 12 reaction rates caused >15% change in the amplitude of LTP or LTD — these changes are represented by black (multi-pathway parameters), blue (PKA-pathway-related parameters), and green (PKC-pathway-related parameters) bars. The underlying parameter changes are printed above the corresponding bar.

## The model flexibly reproduces data from various cortical LTP/LTD experiments

The richness of the intracellular signalling machinery behind LTP and LTD poses challenges for both qualitative and quantitative comparison between results from different cell types, obtained using different stimulation protocols, or even published by different laboratories *Larkman and Jack, 1995*. Computational biochemically detailed models have been proposed as an absolutely reproducible tool that is particularly suited for unifying our understanding of LTP and LTD across cell types and brain regions *Manninen et al., 2010*. Here, we show that our model for intracellular signalling in a cortical post-synaptic spine — through the use of varying concentrations of different proteins — can be flexibly tuned to reproduce data from the experimental literature of cortical LTP/LTD. This allows one to make predictions for the differences in intracellular machineries underlying each of the experiments, leading to a more complete view of the plasticity-related signalling pathways in

different cell types in the cortex and the effects of the stimulation protocol on the plasticity outcome across cortical areas.

To show the flexibility of our model, we aimed to reproduce a large amount of data on cortical plasticity across cortical areas and stimulation paradigms. We reviewed the literature of cortical

**Table 2.** List of LTP/LTD experiments in the cortex.

The first column labels the experimental data set and names the underlying study. The second column shows the considered synaptic pathway and the third column shows whether the observed LTP/LTD had a pre- or post-synaptic origin. The fourth and fifth columns show the frequency (in Hz) of stimulation and the number of pulses delivered, respectively: $10 \times 4$ means that 10 trains of 4 pulses with 10 ms interval (100 Hz) were delivered, and likewise, $25 \times 5$ means that 25 trains of 5 pulses with 10 ms interval were delivered. The sixth column tells whether the data were obtained in control conditions or under additional blockers or agonists. The seventh, eighth, ninth, and tenth columns show the relative change in synaptic strength 10, 15, and 20 min after the start of the stimulus protocol and an average SD of the relative synaptic strengths — these values were approximated from the LTP/LTD curves plotted in the underlying references. The rows correspond to experiments from a given reference that are divided to 11 different experimental data sets. Within each data set, the underlying system is assumed to be otherwise similar to the control except for the applied modifier: as an example, the chemical or genetic blockade of CaMKII activity (as performed in *Ma et al., 2008* and *Hardingham et al., 2003*) is here expected to only affect the ability of CaMKII to autophosphorylate, and the rest of the model parameters are kept fixed. The experiments printed in grey were included in the underlying study, but were excluded from the main analyses of the present work (see main text). EC – entorhinal cortex; PFC – prefrontal cortex; BC – barrel cortex; ACC – anterior cingulate cortex; VC – visual cortex; AuC – auditory cortex; CC – corpus callosum. (*): The LFS of 900 3-ms pulses at 5 Hz in data sets VC-1 and VC-2 was replaced by 180 15-ms pulses at 1 Hz to decrease computational load in the optimisation.

| Data set | Reference | Pathway | Pre/post | Freq. | $N_{pulses}$ | Experiment | 10 min | 15 min | 20 min | SD |
|---|---|---|---|---|---|---|---|---|---|---|
| EC-1 | *Ma et al., 2008* | horizontal | mostly | 100 | 100 | control | 1.3 | 1.4 | 1.3 | 0.1 |
| | | | post | | | CaMKII blocked | 1.05 | 1.02 | 0.95 | 0.07 |
| | | | | | | without post-syn. $Ca^{2+}$ | 1.05 | 1.05 | 1.1 | 0.09 |
| EC-2 | *Ma et al., 2008* | ascending | mostly | 100 | 100 | control | 1.6 | 1.6 | 1.6 | 0.11 |
| | | | post | | | PKA blocked | 1.4 | 1.4 | 1.4 | 0.13 |
| | | | | | | without post-syn. $Ca^{2+}$ | 1.3 | 1.4 | 1.4 | 0.13 |
| PFC-1 | *Sáez-Briones et al., 2015* | CC→PFC | n/a | 312 | 156 | control | 2.0 | 1.98 | 1.9 | 0.08 |
| | | | | | | without $\beta$-adrenergic ligand | 1.34 | 1.4 | 1.36 | 0.09 |
| PFC-2 | *Flores et al., 2011* | CC→PFC | n/a | 312 | 156 | control | 1.7 | 1.6 | 1.64 | 0.12 |
| | | | | | | without $\beta-1$-receptor agonist | 1.43 | 1.45 | 1.43 | 0.1 |
| BC | *Hardingham et al., 2003* | L4→L2/3 | n/a | 5 | $10 \times 4$ | control | 1.35 | 1.4 | 1.3 | 0.09 |
| | | | | | | CaMKII mutant | 1.25 | 1.2 | 1.1 | 0.09 |
| ACC | *Song et al., 2017* | L5/6 → L2/3 | post | 5 | $10 \times 4$ | control | 1.55 | 1.4 | 1.4 | 0.05 |
| | | | | | | without s845 | 1.1 | 1.05 | 1.05 | 0.07 |
| | | | | | | without s831 | 1.35 | 1.4 | 1.3 | 0.1 |
| PFC-3 | *Zhou et al., 2013* | L2/3 → L2/3 | mostly | 0.1 | 50 | control | 1.3 | 1.4 | 1.4 | 0.14 |
| | | | post | | | without $\beta-1$-receptor agonist | 1.1 | 1.2 | 1.2 | 0.13 |
| VC-1 | *Kirkwood et al., 1997* | L4 → L3 | n/a | 5 | $10 \times 4$ | (CTR, HFS) | 1.3 | 1.26 | 1.26 | 0.07 |
| (adult) | | | | | | (without CaMKII, HFS) | 1.02 | 1.02 | 1.02 | 0.02 |
| | | | | 5 | 900* | (CTR, LFS) | n/a | 0.95 | 0.95 | 0.05 |
| | | | | | | (without CaMKII, LFS) | n/a | 0.88 | 0.93 | 0.03 |
| VC-2 | *Kirkwood et al., 1997* | L4 → L3 | n/a | 5 | $10 \times 4$ | (CTR, HFS) | 1.2 | 1.18 | 1.18 | 0.05 |
| (4–5 w) | | | | | | (without CaMKII, HFS) | 1.07 | 1.09 | 1.08 | 0.03 |
| | | | | 5 | 900* | (CTR, LFS) | n/a | 0.79 | 0.82 | 0.03 |
| | | | | | | (without CaMKII, LFS) | n/a | 0.82 | 0.89 | 0.03 |
| AuC-1 | *Kotak et al., 2007* | L6 → L5 | n/a | 1 | $25 \times 5$ | LTP-expressing cells | 1.98 | 1.58 | 1.93 | 0.19 |
| AuC-2 | *Kotak et al., 2007* | L6 → L5 | n/a | 1 | $25 \times 5$ | LTD-expressing cells | 0.77 | 0.68 | 0.67 | 0.09 |

plasticity, and picked eight studies where one or more types of neurons were tested using one or more stimulation protocols and the outcome was quantified using electrophysiological measurements (*Table 2*). These studies comprised 11 data sets that described the response of a neuron population in entorhinal cortex (EC), prefrontal cortex (PFC), barrel cortex (BC), anterior cingulate cortex (ACC), visual cortex (VC), or auditory cortex (AuC) to plasticity inducing protocols (*Table 2*). For each experiment in each data set, we assigned an objective function that quantified the error between the predicted LTP/LTD outcome (measured in relative synaptic conductance) and the data (typically measured in fold change of field EPSP slope). The objective functions were averaged across different time instants (10, 15, and 20 min post-stimulus-onset). We then ran a multi-objective optimisation algorithm (see Materials and methods, section '*Parameter alterations and model fitting*') that aimed at finding the values for model parameters that minimised these objective functions. The fitted parameters included the amplitudes of pre-synaptic stimulation-associated fluxes of $Ca^{2+}$, β-adrenergic ligand and glutamate in addition to GluR1 fraction and factors for the protein concentrations of different pathways. Here, both the $Ca^{2+}$ and the neuromodulatory inputs were square-pulse injections that followed every pre-synaptic stimulation although some of the studies of *Table 2* used bath application of neuromodulatory agents; however, the temporal distribution of the neuromodulators has only a small effect in our model as shown earlier (*Figure 3—figure supplement 4*).We ran the optimiser for 20 generations. For data sets VC-1 and VC-2, we did not find parameter sets that would fulfil all four objective functions, and therefore, we re-fitted the model for these data sets excluding the CaMKII-blocked experiments (printed in grey in *Table 2*).

We found groups of parameter sets that fit within one standard deviation (SD) on average from the target values of synaptic conductance for each data set of *Table 2* (*Figure 9A–C*). There were differences in the numbers of acceptable parameter sets between the data sets due to differences in the postulated strength of the LTP/LTD, the number of experiments, and the SD of the post-stimulus synaptic conductance (*Figure 9D*). The data sets EC-1 and BC were particularly challenging to fit (<0.2% of the parameter sets tested by the optimiser gave an acceptable fit; *Figure 9D*). By contrast, the data set AuC-2 was the easiest to fit (3.9% of the parameter sets were acceptable; *Figure 9D*).

The obtained parameters reflect the pathways needed for the type of plasticity. For example, the LTP of synapses of the horizontal but not those of the ascending pathway to EC were blocked by CaMKII inhibition, while the LTP of synapses of the ascending pathway were blocked by PKA inhibition (*Ma et al., 2008*). This is reflected in the obtained parameter sets: the parameter controlling CaM and CaMKII concentrations ($f_{CaMKII}$) was significantly larger (U-test, p-value<0.001) in the horizontal-pathway (data set EC-1) synapse models, while the parameter controlling upstream PKA-pathway proteins ($f_{PKA}$) was significantly larger in the models reproducing the data from the ascending pathway (data set EC-2; *Figure 9E*). The GluR1-GluR2 ratio, in turn, was not significantly different between the two data sets (*Figure 9E*). As a contrasting example, our model predicts a large variety of parameters that reproduce the LTP and LTD of data sets AuC-1 and AuC-2 but, consistent with the results of *Figure 7*, the GluR1 ratio was significantly larger in the model parameters fitted to the data from LTP-expressing neurons than from LTD-expressing neurons (*Figure 9E*). The complete graphs of parameter value distributions in the 11 data sets and the parameter set producing the best fit (*Figure 9A*) for each data set are shown in Supplementary data, *Figure 9—figure supplements 1–11*. Taken together, our model-fitting experiment shows that the model can be fit to many types of multi-condition plasticity data — without altering the reaction rates — and that the resulting predictions of the underlying protein concentrations reflect the mechanisms proposed by the experimental studies.

The models obtained by fitting the initial concentrations to data provide an important tool for predicting the outcome of plasticity under various stimulus protocols and chemical agents. We carried out additional simulations with the obtained models using HFS protocol. The models fitted for data from EC (data sets EC-1 and EC-2, [*Ma et al., 2008*]), BC (*Hardingham et al., 2003*), ACC (*Song et al., 2017*), and LTP-expressing auditory cortical neurons (data set AuC-1, *Kotak et al., 2007*) predicted a steady increase in response to HFS, while models fitted for other cortical data predicted a mixture of LTP, LTD, and no change (*Figure 10A*). Furthermore, to obtain experimentally testable predictions for the dependency of the plasticity outcome in different cortical areas on the intracellular signalling, we simulated the models from each data set with the corresponding stimulus protocol with CaMKII, PKA, or PKC blockade. The inhibition of CaMKII impaired

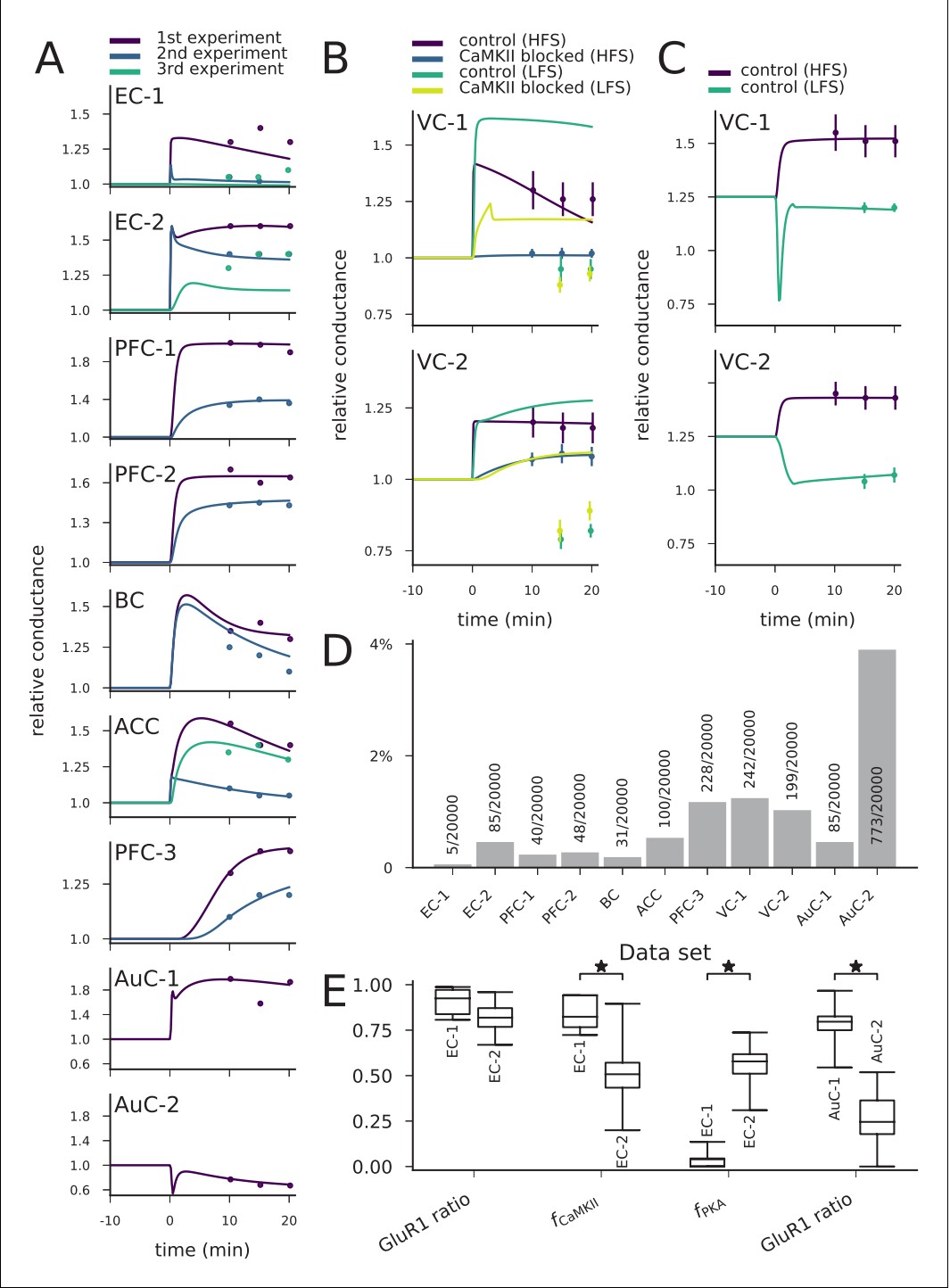

**Figure 9.** The model can be fit to LTP/LTD data from different cortical areas. (**A**) The model could be fit to LTP/LTD data from data sets EC-1 (top), EC-2, PFC-1, PFC-2, BC, ACC, PFC-3, AuC-1, and AuC-2 (bottom). The curves represent the model predictions of the best-fit parameter sets, and the dots represent the experimental data from *Table 2*. For data sets other than AuC-1 and AuC-2, several experiments with various chemical agents or genetic mutations were performed for each neuron population: these are ordered as in *Table 2* (e.g., in data set EC-1, purple (1 st experiment) corresponds to control, blue (2nd experiment) to CaMKII-blocked experiment, and green (3rd experiment) to the experiment where post-synaptic $Ca^{2+}$ was blocked). (**B**) The model could not be fit to the complete LTP/LTD data from data sets VC-1 (top) and VC-2 (bottom). The best parameter sets correctly predicted the LTP/LTD in up to two experiments (e.g., the selected parameter sets reproduce the HFS data with and without

*Figure 9 continued on next page*

*Figure 9 continued*

CaMKII inhibitor, but failed to reproduce the LFS data). (**C**) The model could be fit to the LTP/LTD data from data sets VC-1 (top) and VC-2 (bottom) when CaMKII-blocked experiments were ignored. The vertical bars in (**B**) and (**C**) represent the SD from the experimental data. (**D**) Proportion of accepted parameter sets across the 20 generations of multi-objective optimisation (20'000 parameter sets in total) in each data set. (**E**) Box plots of selected parameters in the acceptable parameter sets of data sets EC-1 and EC-2 (three left-most pairs) and AuC-1 and AuC-2 (right-most pair). Values of $f_{CaMKII}$ and $f_{PKA}$ are linearly scaled such that the values 0 and 1 correspond to 0 and double the original value of the underlying parameters, respectively (CaM and CaMKII for $f_{CaMKII}$, and R, Gs, AC1, and AC8 for $f_{PKA}$, see Materials and methods, section '*Parameter alterations and model fitting*'). The medians were significantly different in the compared data sets (U-test, p-value<0.001).

The online version of this article includes the following figure supplement(s) for figure 9:

**Figure supplement 1.** Parameters for data set EC-1.
**Figure supplement 2.** Parameters for data set EC-2.
**Figure supplement 3.** Parameters for data set PFC-1.
**Figure supplement 4.** Parameters for data set PFC-2.
**Figure supplement 5.** Parameters for data set BC.
**Figure supplement 6.** Parameters for data set ACC.
**Figure supplement 7.** Parameters for data set PFC-3.
**Figure supplement 8.** Parameters for data set VC-1.
**Figure supplement 9.** Parameters for data set VC-2.
**Figure supplement 10.** Parameters for data set AC-1.
**Figure supplement 11.** Parameters for data set AC-2.

the LTP in data set EC-1 (horizontal pathway) but had little or no effect on the plasticity in other cortical areas (*Figure 10B*). The lack of effect of CaMKII blockade on the ascending pathway of EC — an experiment which was not included in the fitting of the model (*Table 2*) — validates the underlying models in this aspect since similar results were observed in *Ma et al., 2008*. Moreover, the similarities in the predicted effect of CaMKII-, PKA-, and PKC-pathway blockades between the two models of CC→PFC synapses (PFC-1 and PFC-2; *Figure 10B–D*) serve as an additional validation of these models. The inhibition of PKA impaired LTP in all cortical areas, except for LTP in the horizontal pathway of the EC (EC-1; *Figure 10C*). Our models also predicted that LTP of the CC→PFC pathway and LFS-induced LTP in VC can be effectively weakened or even be transformed to a mild LTD by PKA blockade (*Figure 10C*). Finally, our models predicted that PKC inhibition transformed all forms of LTD (LFS-induced LTD in VC-1 and VC-2; LTD in AuC-2) into LTP and impaired certain forms of LTP (LTP in EC-2; LTP in AuC-1) (*Figure 10D*). Taken together, our results suggest that almost all forms of post-synaptic plasticity in the cortex are likely to be PKA-dependent, and that many types of cortical plasticity are also influenced by CaMKII and PKC activity. Our results highlight the need for additional chemical or genetic manipulations to be done when experimenting on cortical plasticity in order to correctly reveal the intracellular signalling cascades in the post-synaptic spine.

## Discussion

We built a single-compartment model describing the major post-synaptic signalling pathways leading to LTP and LTD in the cortex. We showed that our model reproduced conventional types of LTP and LTD, where an HFS-induced increase in GluR1 can increase the synaptic conductance (LTP) and an LFS-induced endocytosis of GluR2 can decrease it (LTD; *Figure 3*) and reproduced STDP data from visual cortical layer 2/3 pyramidal cells (*Figure 5*). Our model explains how different forms of plasticity depend on the concentrations of PKA- and PKC-pathway proteins (*Figure 7*). We also showed that our model can be fit to explain the pathway dependencies of various types of neocortical LTP/LTD data published in the literature by altering the magnitude of $Ca^{2+}$ and ligand fluxes and the concentrations of post-synaptic proteins regulating the $Ca^{2+}$ efflux and PKA- and PKC-pathway dynamics (*Figure 9*). Our fitted models provide a powerful tool for testing hypotheses on the effects of chemical or genetic manipulations on the LTP and LTD in different cortical regions (*Figure 10*).

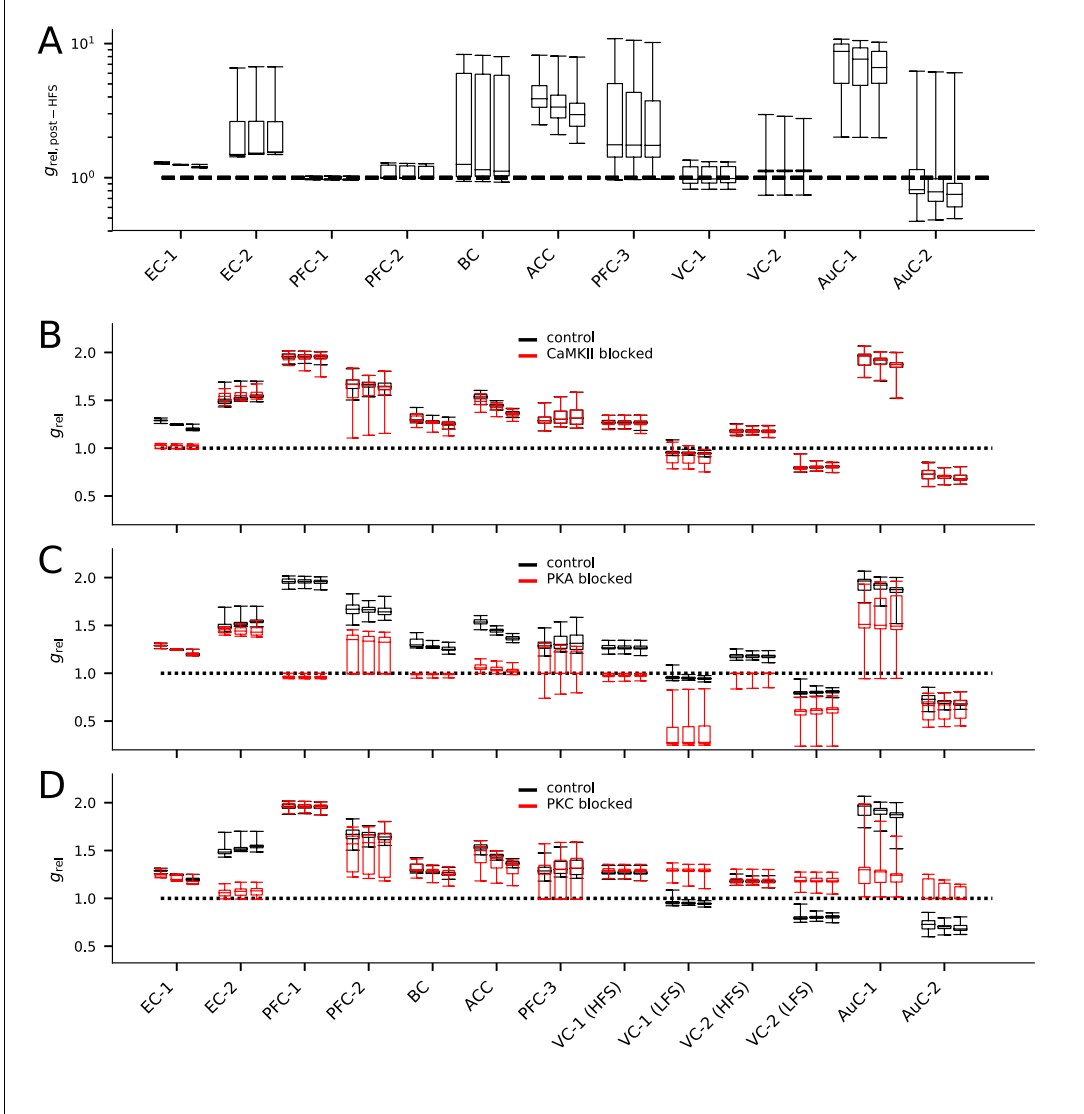

**Figure 10.** The models describing plasticity in different cortical areas predict diverse responses to modified stimulation protocol and stimulation under chemical blockers. (**A**) The predicted responses of the 20 best models in each data set to HFS (100 pulses at 100 Hz) stimulation. (**B–D**) The predicted responses of the 20 best models in each data set to the applied stimulation protocol (see *Table 2*) when CaMKII (**B**), PKA (**C**), or PKC (**D**) activity was blocked (red) or under control condition (black).

## Role of GluR2 in synaptic plasticity in the neocortex

GluR2 subunits are highly expressed in neocortical neurons (*Kondo et al., 1997*), and their endocytosis mediates (or, at minimum, is correlated with) synaptic depression in many cortical regions such as ACC (*Toyoda et al., 2007*), VC (*Heynen et al., 2003*), and PFC (*Van den Oever et al., 2008*). Previous intracellular signalling-based models of neocortical LTP/LTD exist, but they do not take into account the contributions of GluR2 subunit. For example, in three previous models (*D'Alcantara et al., 2003*; *Castellani et al., 2005*; *Honda et al., 2013*), S831-phosphorylation-mediated LTP was described in a fashion similar to our model (although more approximations were made), but the phosphorylation site S845 was assumed to be basally phosphorylated and LTD was caused by modest $Ca^{2+}$ inputs that led to PP1 or PP2B-mediated dephosphorylation of S845. Although there is support for this order of events (*Lee and Kirkwood, 2011*; *Diering et al., 2016*), newer findings have confirmed the low degrees of phosphorylation of both S831 and S845 at a basal state in cortical cells, especially in synaptic spines (*Diering et al., 2016*). A recent phenomenological model has also shed light on the dependency of cortical STDP on the pairing interval and location of

the synapse (*Ebner et al., 2019*), but their mechanisms are not specific to any particular AMPAR or NMDAR subunit. To analyse the contributions of GluR2 subunits to neocortical LTP/LTD, we included in our model the signalling pathways leading to both phosphorylation-mediated exocytosis of GluR1 and endocytosis of GluR2. Our model could thus be used to study not only PKA-mediated LTP or PKC-mediated LTD but also their co-effects and co-dependencies.

## Implications of the study

The modelling results of the present work give rise to experimentally testable predictions. For example, our STDP model, when stimulated without β-adrenergic ligands, suggests that at near-zero pairing-intervals the magnitude of the depression may be decreased or even switched to mild LTP (*Figure 5J*). In many experimental studies (including *Seol et al., 2007*), the type and magnitude of plasticity in this regime of STDP is not reported. We also predict that a mild LTP (24–60% LTP; class 4 in *Figure 7*) can be obtained through many differently weighted interactions of PKA and PKC pathways and Ca$^{2+}$ extrusion strengths (*Figure 7F–H*, *Figure 9—figure supplement 1* and *2*, *Figure 9—figure supplement 5–7*).Importantly, this is the regime of a wealth of experimental LTP data (*Tsumoto, 1990*; *Table 2*), which is consistent with the great diversity of LTP mechanisms observed in the neocortex (*Feldman, 2009*). Based on our simulated data (*Figure 10*), we suggest that in order to correctly characterise the mechanisms behind LTP of especially this magnitude, both experiments that activate the PKA pathway and experiments that block or activate the PKC pathway should be carried out. It is also important to know whether and to what degree GluR1 and GluR2 subunits are present in the synapse, since the balance of GluR1 and GluR2 subunits seems to be a determinant parameter permitting certain types of plasticity while prohibiting others (*Figures 7B* and *9E*, and *Figure 3—figure supplement 1*).

A key challenge in the study of synaptic plasticity is the diversity of LTP/LTD observed across the cell types in the brain (*Granger and Nicoll, 2014*). Differences in the transcriptome have been proposed as one of the sources for this variability (*Lisachev and Shtark, 2018*). We believe our model can be used to explain some of the discrepancies in the experimental data in this regard and expand the understanding of possible molecular contributors to LTP/LTD. For example, it is known that activation of PKA pathway by dopamine or noradrenaline in PFC pyramidal neurons increases the synaptic conductance through GluR1 membrane insertion (*Sun et al., 2005*; *Xu et al., 2010*). Our model is in agreement with this (*Figure 3*), but it also proposes that the LTP can be impaired by over- or underexpression of many involved proteins, such as AC1, I-1 (inhibitor of PP1), PDE4, PDE1, GluR1, and CaM, and even alterations in ATP concentration (see *Figure 8B*). Small differences in the concentrations of a number of such contributing proteins are likely to cause significant alterations to LTP observed in different brain areas and cell types.

Due to the inclusion of three major LTP/LTD pathways in the neocortex, our model provides a more accurate means than earlier models for exploring how the Ca$^{2+}$ dynamics in the spine affects the plasticity outcome in many stimulation protocols, STDP in particular. Our model suggests that the plasticity outcome of the STDP protocol is strongly correlated with the total amount of Ca$^{2+}$ entering the post-synaptic spine, and less so with the peak Ca$^{2+}$ flux (*Figure 6—figure supplement 1*).The total amount of Ca$^{2+}$ influx could thus provide a better biomarker for plasticity than the previously considered amplitude and duration of the Ca$^{2+}$ transient (*Evans and Blackwell, 2015*).

## Validity of the results and limitations of the study

Our model of total synaptic conductance of the post-synaptic spine is based upon a number of assumptions. First, the prediction of a large increase of conductance that follows the replacement of GluR2 subunits at the membrane by GluR1 subunits (e.g., *Figure 3*) is based upon the findings on differences in single-channel conductances of different types of AMPAR tetramers in hippocampal neurons (*Oh and Derkach, 2005*). Following *Oh and Derkach, 2005*, we assumed that CaMKII-phosphorylation of S831 only increases the conductance of GluR1 homomers and not that of GluR1/GluR2 heteromers, although also heteromers have been observed to increase their conductance in the presence of transmembrane AMPAR regulatory proteins (*Kristensen et al., 2011*). Second, we assumed a random tetramerization procedure in which each of the four subunits in the tetramer may be either GluR1 or GluR2 subunit. Traditionally, AMPARs were thought to assemble as dimers of like dimers, that is, that first GluR1s and GluR2s assemble into homomeric R1-R1 and R2-R2 dimers and

R1-R2 heterodimers and that these three types of dimers only assemble into tetramers with a dimer of its own kind (*Gan et al., 2015*). However, recent findings of heterotetramers with only one GluR1 subunit (*Zhao et al., 2019*) challenge this model. To show that our results were not dependent on the details of this process, we reproduced our results using the alternative (dimer of like dimers) tetramer formation rule. Using a slightly modified GluR1-GluR2 balance (35:65), this model reproduced HFS-induced LTP and LFS-induced LTD (*Figure 3—figure supplement 2A*) as well as the neuromodulator-gated STDP (*Figure 3—figure supplement 2A*).In summary, our model predictions were not dependent on the assumptions on the tetramer formation rule.

Our model reproduced the qualitative results of STDP of layer 2/3 pyramidal cells in visual cortex being gated by neuromodulators, but there were quantitative differences. When acetylcholine was present, our model predicted a prominent decrease in GluR2 membrane-expression regardless the pairing interval (*Figure 5I*), which caused a notable LTD for very large pairing intervals (*Figure 5J*), whereas the experimental data showed attenuation of the depression for large inter-stimulus intervals (*Seol et al., 2007*). This discrepancy is likely caused by processes allowing slower time-scale (>50 ms) interaction between the pre- and post-synaptic stimulus that are either not included (e.g., $Ca^{2+}$-induced $Ca^{2+}$ release or cAMP-dependence of HCN channels) or not adequately strong in the multi-compartmental model. For example, the contributions of voltage-gated $Ca^{2+}$ channels and SK channels to the neuron electrophysiology may be large (*Mäki-Marttunen et al., 2017*; *Mäki-Marttunen et al., 2018*) — to this end, we showed here that the SK currents are amplified by the subsequent pulses stimulating the post-synaptic neuron and that this is one factor increasing the LTD for large ISIs (*Figure 6*). Note that this prediction of lowered synaptic strength for large absolute ISIs is not to be considered a basal synaptic state under spontaneous activity since the amplitude of the LTD is significantly decreased both by the removal of cholinergic neuromodulation (*Figure 6J*) and a decrease of stimulating frequency (data not shown). On the other hand, mechanisms lacking from the biochemical model (e.g., voltage-dependence of the $Ca^{2+}$-extrusion rate of NCX *Weber et al., 2002*) could also impede our results in this matter. Some aspects of cellular physiology could therefore be better represented if we incorporated both biochemical signalling and multicompartmental Hodgkin-Huxley-type modelling into the simulations, as done in modelling studies of persistent neuron firing (*Neymotin et al., 2016*) and astrocyte electrophysiology (*Savtchenko et al., 2018*).

The quality of the model fitting to experimental data in *Figure 9* is restricted by the fact that not all of the LTP/LTD data in *Table 2* were confirmed to have a post-synaptic origin. This may be the key source of discrepancy in the fitting of the model to the CaMKII-blocked data from *Kirkwood et al., 1997* (*Figure 9B*), since CaMKII activation at the pre-synaptic spine may lead to EPSC potentiation through an increase in neurotransmitter release (*Ninan and Arancio, 2004*). This scenario is supported by *Seol et al., 2007* where S831-deficient mice were observed to show normal post-synaptic LTP in the VC.

## Outlook

Our results on interactions of the different pathways in post-synaptic spines including both GluR1 and GluR2 subunits provide valuable insights on the contributions of protein expression on the plasticity of the synapse. Previously, synaptic plasticity outcomes in the cortex have been conjectured to depend on the type of the post-synaptic cell type, in addition to the timing and frequency of the applied stimuli and dendritic filtering properties (*Bi and Poo, 1998*; *Sjöström et al., 2001*). Our model provides a way to analyse exactly which aspects of PKA-, PKC- and CaMKII-pathway signalling underlie these cell-type-dependent differences in synaptic plasticity. Combining our biochemically detailed model with biophysically detailed models from different cortical areas will provide models with better predictive power in the future. Moreover, our model can be used for initial testing of hypotheses concerning dysfunctions (including chemical and genetic manipulations) of many intracellular signalling proteins and their role in impairments of cortical synaptic plasticity. By altering the initial concentrations or reaction rates of various species according to disease-associated functional genetics data, the model can be used to provide insights into the disease mechanisms of mental disorders that express both genetic disposition of post-synaptic signalling pathways and plasticity-related phenotypes, such as schizophrenia (*Devor et al., 2017*).

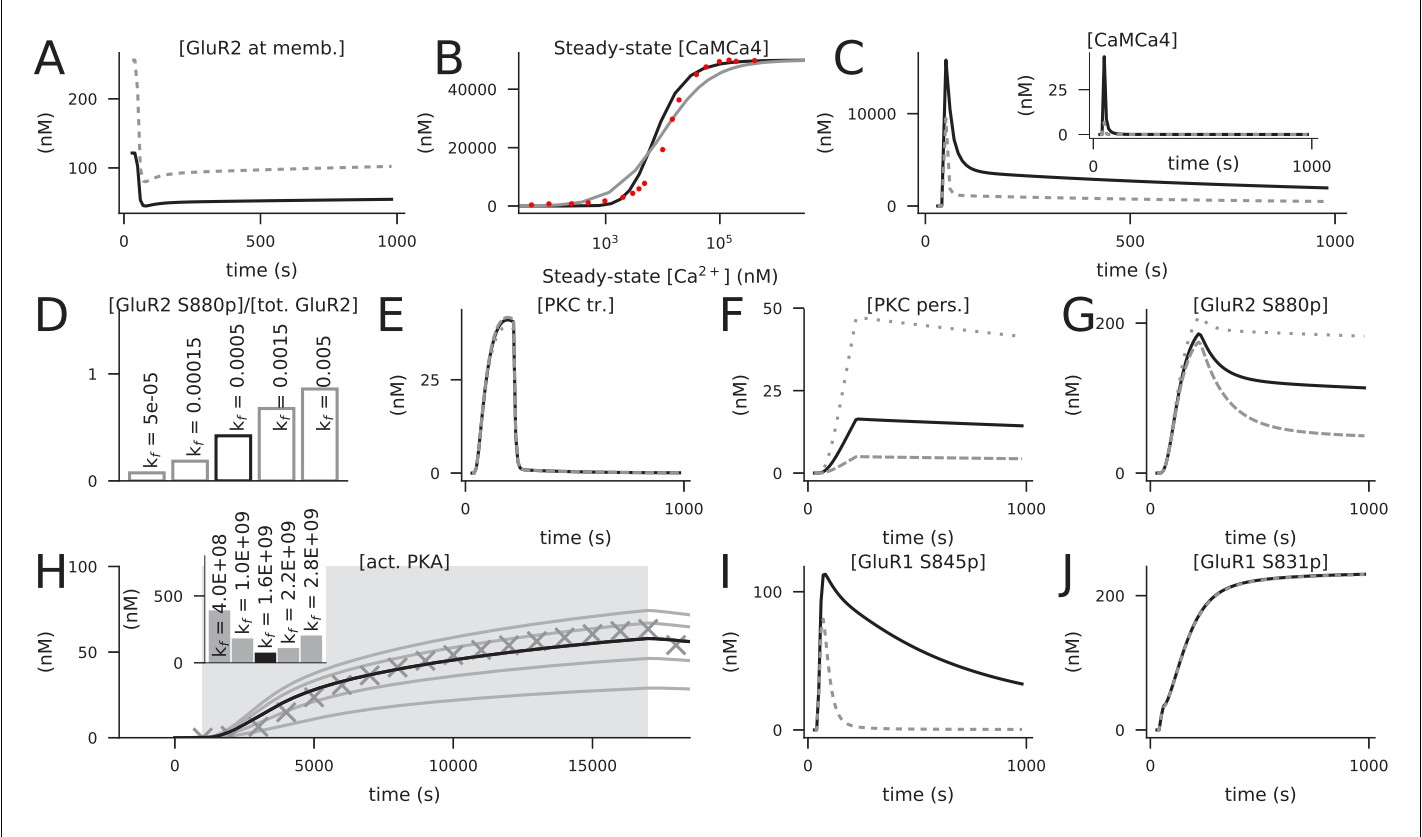

**Figure 11.** Calibration of the model. Black curves represent the final model, while grey lines represent predictions of models where previous model components or tentative parameter values were used. (**A**) Concentration of membrane-inserted GluR2 in 4xHFS when the forward rate of the membrane insertion of non-phosphorylated GluR2 was 0.0055 1/ms *Gallimore et al., 2018* (grey) or 0.00025 1/ms (black). The rate 0.00025 1/ms caused a resting-state concentration of 121 nM for the membrane-bound GluR2 subunits, which is 45% of the total GluR2 concentration (270 nM). (**B**) Steady-state concentration of activated (bound by four $Ca^{2+}$ ions) CaM in response to a prolonged $Ca^{2+}$ input amplitude when the two-step (grey) or three-step (black) activation of CaM by $Ca^{2+}$ was used. The x-axis shows the corresponding steady-state concentration of free $Ca^{2+}$. Here, the initial concentrations of molecular species were as in *Li et al., 2020*, namely, 50 μM for CaM, Ng, PP2B, and CaMKII and 0 μM for all other species. Red dots show experimental data from *Hoffman et al., 2014*. (**C**) Concentration time course of non-protein-bound activated CaM (inset) or total activated CaM (main figure) in response to 4xHFS when the two-step (grey) or three-step (black) activation of CaM by $Ca^{2+}$ was used. (**D**) Percentage of S880-phosphorylated GluR2 15 min after LFS when different forward rates of the activation of persistent PKC ($k_f$ between 0.00005 and 0.005 1/(nM ms)) were used. The value $k_f$ = 0.0005 1/(nM ms) gave a percentage of 47%, in close agreement with *Ashby et al., 2004*. (**E–G**) The dynamics of transiently active PKC (**E**) were not strongly influenced by the forward rate of the activation of persistent PKC (reaction 140), but those of persistently active PKC (**F**) and S880-phosphorylated GluR2 (**G**) were significantly affected. Black curves show the data corresponding to $k_f$ = 0.0005 1/(nM ms), while the grey lines show the data corresponding to $k_f$ = 0.00015 1/(nM ms) (dashed) and $k_f$ = 0.0015 1/(nM ms) (dotted). (**H**) Predicted responses of an isolated PKA activation model (reactions 59 and 93) to a 16 s cAMP input (dim grey background) when different values of the forward rate of PKA binding with four cAMP molecules were used. The curves show the concentration of the catalytic PKA subunit when different forward rates of PKA–cAMP binding (from bottom to top: 0.4 $\times 10^9$, 1.0$\times 10^9$, 1.6 $\times 10^9$, 2.2$\times 10^9$, and 2.8 $\times 10^9$ 1/(nM$^4$ms)) were used. The markers show the corresponding data when the two-step PKA–cAMP binding model of *Jędrzejewska-Szmek et al., 2017* was used. Inset: summed absolute differences between the tentative data (curves) and simulated data from the previous model (markers). The model with the forward rate of $k_f$ = 1.6 $\times$ $10^9$ 1/(nM$^4$ms) gave the closest correspondence to the model of *Jędrzejewska-Szmek et al., 2017*. (**I**) Concentration of S845-phosphorylated GluR1 in response to 4xHFS when the single-step (reaction 59, black) or two-step (from *Jędrzejewska-Szmek et al., 2017*, grey) PKA–cAMP binding was used. (**J**) Concentration of S831-phosphorylated GluR1 in response to 4xHFS when PKC did (black) or did not (grey, overlaid) phosphorylated S831 in GluR1s.

The online version of this article includes the following figure supplement(s) for figure 11:

**Figure supplement 1.** 1 hr of simulation without inputs is sufficient to obtain a steady state.

**Figure supplement 2.** The model STDP model is robust to changes in AMPA conductance but sensitive to changes in NMDA condutance in the multicompartmental layer 2/3 pyramidal cell model.

**Table 3.** List of model reactions.

(A) The reaction-rate units are in 1/ms, 1/(nMms), 1/(nM$^2$ms), 1/(nM$^3$ms), or 1/(nM$^4$ms), depending on the number of reactants. Reactions are grouped by similar modes of action and identical forward and backward rates. The denominators **X**, **Y**, and **Z** represent groups of species detailed below. †: backward reaction rate proportional to [PKAc], not to [PKAc]$^2$. (B) Groups of species as used in panel (A).

**(A)**

| ID | Reaction | Forw. Rate | Backw. Rate | ID | Reaction | Forw. Rate | Backw. Rate |
|----|----------|------------|-------------|----|----------|------------|-------------|
| 1 | Ca + PMCA ⇌ PMCACa | 5e-05 | 0.007 | 71 | GluR1$\mathbf{X}_{22}$ + $\mathbf{Y}_{22}$ ⇌ GluR1$\mathbf{Z}_{22}$ | 2.78e-08 | 0.002 |
| 2 | PMCACa ⇌ PMCA + CaOut | 0.0035 | 0.0 | 72 | GluR1$\mathbf{X}_{23}$ ⇌ GluR1 $\mathbf{Y}_{23}$ + $\mathbf{Z}_{23}$ | 0.0005 | 0 |
| 3 | Ca + NCX ⇌ NCXCa | 1.68e-05 | 0.0112 | 73 | GluR1$\mathbf{X}_{24}$ + PKAc ⇌ GluR1$\mathbf{Z}_{24}$ | 4e-06 | 0.024 |
| 4 | NCXCa ⇌ NCX + CaOut | 0.0056 | 0.0 | 74 | GluR1$\mathbf{X}_{25}$ + PP1 ⇌ GluR1$\mathbf{Z}_{25}$ | 8.7e-07 | 0.00068 |
| 5 | CaOut + Leak ⇌ CaOutLeak | 1.5e-06 | 0.0011 | 75 | GluR1$\mathbf{X}_{26}$ ⇌ GluR1 $\mathbf{Y}_{26}$ + PP1 | 0.00017 | 0 |
| 6 | CaOutLeak ⇌ Ca + Leak | 0.0011 | 0.0 | 76 | GluR1$\mathbf{X}_{27}$ + PP1 ⇌ GluR1$\mathbf{Z}_{27}$ | 8.75e-07 | 0.0014 |
| 7 | Ca + Calbin ⇌ CalbinC | 2.8e-05 | 0.0196 | 77 | GluR1$\mathbf{X}_{28}$ ⇌ GluR1 $\mathbf{Y}_{28}$ + PP1 | 0.00035 | 0 |
| 8 | L ⇌ LOut | 0.0005 | 2e-09 | 78 | GluR1$\mathbf{X}_{29}$ + PP2BCaMCa4 ⇌ GluR1$\mathbf{Z}_{29}$ | 2.01e-06 | 0.008 |
| 9 | L + R ⇌ LR | 5.555e-06 | 0.005 | 79 | GluR1$\mathbf{X}_{30}$ ⇌ GluR1$\mathbf{Y}_{30}$ + PP2BCaMCa4 | 0.002 | 0 |
| 10 | LR + Gs ⇌ LRGs | 6e-07 | 1e-06 | 80 | GluR1$\mathbf{X}_{31}$ ⇌ GluR1_memb$\mathbf{X}_{31}$ | 2e-07 | 8e-07 |
| 11 | Gs + R ⇌ GsR | 4e-08 | 3e-07 | 81 | GluR1_S845$\mathbf{X}_{32}$ | | |
| 12 | GsR + L ⇌ LRGs | 2.5e-06 | 0.0005 | | ⇌ GluR1_memb_S845$\mathbf{X}_{32}$ | 3.28e-05 | 8e-06 |
| 13 | LRGs ⇌ LRGsbg + GsaGTP | 0.02 | 0.0 | 82 | PDE1 + CaMCa4 ⇌ PDE1CaMCa4 | 0.0001 | 0.001 |
| 14 | LRGsbg ⇌ LR + Gsbg | 0.08 | 0.0 | 83 | PDE1CaMCa4 + cAMP ⇌ PDE1CaMCa4cAMP | 4.6e-06 | 0.044 |
| 15 | $\mathbf{X}_1$ + PKAc ⇌ PKAc$\mathbf{X}_1$ | 8e-07 | 0.00448 | 84 | PDE1CaMCa4cAMP ⇌ PDE1CaMCa4 + AMP | 0.011 | 0.0 |
| 16 | PKAc$\mathbf{X}_2$ ⇌ p$\mathbf{X}_2$ + PKAc | 0.001 | 0.0 | 85 | AMP ⇌ ATP | 0.001 | 0.0 |
| 17 | ppLR + PKAc ⇌ PKAcppLR | 1.712e-05 | 0.00448 | 86 | PDE4 + cAMP ⇌ PDE4cAMP | 2.166e-05 | 0.0034656 |
| 18 | pppLR + PKAc ⇌ PKAcpppLR | 0.001712 | 0.00448 | 87 | PDE4cAMP ⇌ PDE4 + AMP | 0.017233 | 0.0 |
| 19 | ppppLR + Gi ⇌ ppppLRGi | 0.00015 | 0.00025 | 88 | $\mathbf{X}_{33}$ + $\mathbf{Y}_{33}$ ⇌ PKAc $\mathbf{Z}_{33}$ | 2.5e-07 | 8e-05 |
| 20 | ppppLRGi ⇌ ppppLRGibg + GiaGTP | 0.000125 | 0.0 | 89 | PKAc$\mathbf{X}_{34}$ ⇌ pPDE4$\mathbf{Y}_{34}$ + PKAc | 2e-05 | 0.0 |
| 21 | pppp$\mathbf{X}_3$ ⇌ pppp$\mathbf{Y}_3$ + Gibg | 0.001 | 0.0 | 90 | pPDE4 ⇌ PDE4 | 2.5e-06 | 0.0 |
| 22 | p $\mathbf{X}_4$ ⇌ $\mathbf{X}_4$ | 2.5e-06 | 0.0 | 91 | pPDE4 + cAMP ⇌ pPDE4cAMP | 0.000433175 | 0.069308 |
| 23 | pp$\mathbf{X}_5$ ⇌ p$\mathbf{X}_5$ | 2.5e-06 | 0.0 | 92 | pPDE4cAMP ⇌ pPDE4 + AMP | 0.3446674 | 0.0 |
| 24 | R + PKAc ⇌ PKAcR | 4e-08 | 0.00448 | 93 | PKAcAMP4 ⇌ PKAr + 2*PKAc | 0.00024 | 2.55e-05 |
| 25 | pR + PKAc ⇌ PKAcpR | 4e-07 | 0.00448 | 94 | Ca + fixedbuffer ⇌ fixedbufferCa | 0.0004 | 20.0 |
| 26 | ppR + PKAc ⇌ PKAcppR | 4e-06 | 0.00448 | 95 | Glu ⇌ GluOut | 0.0005 | 2e-10 |
| 27 | pppR + PKAc ⇌ PKAcpppR | 0.0004 | 0.00448 | 96 | Ca + PLC ⇌ PLCCa | 4e-07 | 0.001 |
| 28 | ppppR + Gi ⇌ ppppRGi | 7.5e-05 | 0.000125 | 97 | GqaGTP + PLC ⇌ PLCGqaGTP | 7e-07 | 0.0007 |
| 29 | ppppRGi ⇌ ppppRGibg + GiaGTP | 6.25e-05 | 0.0 | 98 | Ca + PLCGqaGTP ⇌ PLCCaGqaGTP | 8e-05 | 0.04 |
| 30 | GsaGTP ⇌ GsaGDP | 0.01 | 0.0 | 99 | GqaGTP + PLCCa ⇌ PLCCaGqaGTP | 0.0001 | 0.01 |
| 31 | GsaGDP + Gsbg ⇌ Gs | 0.1 | 0.0 | 100 | PLCCa + Pip2 ⇌ PLCCaPip2 | 3e-08 | 0.01 |
| 32 | GiaGTP ⇌ GiaGDP | 0.000125 | 0.0 | 101 | PLCCaPip2 ⇌ PLCCaDAG + Ip3 | 0.0003 | 0.0 |
| 33 | GiaGDP + Gibg ⇌ Gi | 0.00125 | 0.0 | 102 | PLCCaDAG ⇌ PLCCa + DAG | 0.2 | 0.0 |
| 34 | GsaGTP + AC1 ⇌ AC1GsaGTP | 3.85e-05 | 0.01 | 103 | PLCCaGqaGTP + Pip2 ⇌ PLCCaGqaGTPPip2 | 1.5e-05 | 0.075 |
| 35 | AC1 $\mathbf{X}_6$ + CaMCa4 ⇌ AC1 $\mathbf{Z}_6$ | 6e-06 | 0.0009 | 104 | PLCCaGqaGTPPip2 ⇌ PLCCaGqaGTPDAG + Ip3 | 0.25 | 0.0 |
| 36 | $\mathbf{X}_7$ + ATP ⇌ $\mathbf{Z}_7$ | 1e-05 | 2.273 | 105 | PLCCaGqaGTPDAG ⇌ PLCCaGqaGTP + DAG | 1.0 | 0.0 |
| 37 | AC1GsaGTPCaMCa4ATP | | | 106 | Ip3degrad + PIkinase ⇌ Ip3degPIk | 2e-06 | 0.001 |
| | ⇌ cAMP + AC1GsaGTPCaMCa4 | 0.02842 | 0.0 | 107 | Ip3degPIk ⇌ PIkinase + Pip2 | 0.001 | 0.0 |
| 38 | $\mathbf{X}_8$ + $\mathbf{Y}_8$ ⇌ AC1Gsa$\mathbf{Z}_8$ | 6.25e-05 | 0.01 | 108 | PLC$\mathbf{X}_{35}$ ⇌ PLC$\mathbf{Y}_{35}$ + GqaGDP | 0.012 | 0.0 |
| 39 | $\mathbf{X}_9$ ⇌ cAMP + $\mathbf{Z}_9$ | 0.002842 | 0.0 | 109 | GqaGTP ⇌ GqaGDP | 0.001 | 0.0 |

*Table 3 continued on next page*

*Table 3 continued*

**(A)**

| ID | Reaction | Forw. Rate | Backw. Rate | ID | Reaction | Forw. Rate | Backw. Rate |
|---|---|---|---|---|---|---|---|
| 40 | AC1GiaGTPCaMCa4ATP $\rightleftharpoons$ cAMP + AC1GiaGTPCaMCa4 | 0.0005684 | 0.0 | 110 | GqaGDP $\rightleftharpoons$ Gqabg | 0.01 | 0.0 |
| 41 | AC1CaMCa4ATP $\rightleftharpoons$ cAMP + AC1CaMCa4 | 0.005684 | 0.0 | 111 | Ca + DGL $\rightleftharpoons$ CaDGL | 0.000125 | 0.05 |
| 42 | AC8 + CaMCa4 $\rightleftharpoons$ AC8CaMCa4 | 1.25e-06 | 0.001 | 112 | DAG + CaDGL $\rightleftharpoons$ DAGCaDGL | 5e-07 | 0.001 |
| 43 | CaM + 2*Ca $\rightleftharpoons$ CaMCa2 | 1.7e-08 | 0.035 | 113 | DAGCaDGL $\rightleftharpoons$ CaDGL + 2AG | 0.00025 | 0.0 |
| 44 | $X_{10}$ + Ca $\rightleftharpoons$ $Z_{10}$ | 1.4e-05 | 0.228 | 114 | Ip3 $\rightleftharpoons$ Ip3degrad | 0.01 | 0.0 |
| 45 | $X_{11}$ + Ca $\rightleftharpoons$ $Z_{11}$ | 2.6e-05 | 0.064 | 115 | 2AG $\rightleftharpoons$ 2AGdegrad | 0.005 | 0.0 |
| 46 | CaM + Ng $\rightleftharpoons$ NgCaM | 2.8e-05 | 0.036 | 116 | DAG + DAGK $\rightleftharpoons$ DAGKdag | 7e-08 | 0.0008 |
| 47 | CaM + PP2B $\rightleftharpoons$ PP2BCaM | 4.6e-06 | 1.2e-06 | 117 | DAGKdag $\rightleftharpoons$ DAGK + PA | 0.0002 | 0.0 |
| 48 | CaMCa$X_{12}$ + PP2B $\rightleftharpoons$ PP2B$Z_{12}$ | 4.6e-05 | 1.2e-06 | 118 | Ca + PKC $\rightleftharpoons$ PKCCa | 1.33e-05 | 0.05 |
| 49 | PP2BCaM + 2*Ca $\rightleftharpoons$ PP2BCaMCa2 | 1.7e-07 | 0.35 | 119 | PKCCa + DAG $\rightleftharpoons$ PKCt | 1.5e-08 | 0.00015 |
| 50 | CaMCa4 + CK $\rightleftharpoons$ CKCaMCa4 | 1e-05 | 0.003 | 120 | Glu + MGluR $\rightleftharpoons$ MGluR_Glu | 1.68e-08 | 0.0001 |
| 51 | 2*CKCaMCa4 $\rightleftharpoons$ Complex | 1e-07 | 0.01 | 121 | MGluR_Glu $\rightleftharpoons$ MGluR_Glu_desens | 6.25e-05 | 1e-06 |
| 52 | CKpCaMCa4 + CKCaMCa4 $\rightleftharpoons$ pComplex | 1e-07 | 0.01 | 122 | Gqabg + MGluR_Glu $\rightleftharpoons$ MGluR_Gqabg_Glu | 9e-06 | 0.00136 |
| 53 | CK$X_{13}$ + Complex $\rightleftharpoons$ CK$X_{13}$ + pComplex | 1e-07 | 0.0 | 123 | MGluR_Gqabg_Glu $\rightleftharpoons$ GqaGTP + MGluR_Glu | 0.0015 | 0.0 |
| 54 | 2*Complex $\rightleftharpoons$ Complex + pComplex | 1e-05 | 0.0 | 124 | GluR2$X_{36}$ + PKC$Y_{36}$ $\rightleftharpoons$ GluR2$Z_{36}$ | 4e-07 | 0.0008 |
| 55 | Complex + pComplex $\rightleftharpoons$ 2*pComplex | 3e-05 | 0.0 | 125 | GluR2$X_{37}$ $\rightleftharpoons$ GluR2$Y_{37}$ + PKC$Z_{37}$ | 0.0047 | 0 |
| 56 | CKpCaMCa4 $\rightleftharpoons$ CaMCa4 + CKp | 8e-07 | 1e-05 | 126 | GluR2$X_{38}$ + PP2A $\rightleftharpoons$ GluR2$Z_{38}$ | 5e-07 | 0.005 |
| 57 | CKp$X_{14}$ + PP1 $\rightleftharpoons$ CKp$Z_{14}$ | 4e-09 | 0.00034 | 127 | GluR2$X_{39}$ $\rightleftharpoons$ GluR2$Y_{39}$ + PP2A | 0.00015 | 0 |
| 58 | CKp$X_{15}$ $\rightleftharpoons$ PP1 + CK$Z_{15}$ | 8.6e-05 | 0.0 | 128 | GluR2$X_{40}$ $\rightleftharpoons$ GluR2_memb$X_{40}$ | 0.00024545 | 0.0003 |
| 59 | PKA + 4*cAMP $\rightleftharpoons$ PKAcAMP4 | 1.6e-15 | 6e-05 | 129 | GluR2_S880$X_{41}$ $\rightleftharpoons$ GluR2_memb_S880$X_{41}$ | 0.0055 | 0.07 |
| 60 | Epac1 + cAMP $\rightleftharpoons$ Epac1cAMP | 3.1e-08 | 6.51e-05 | 130 | ACh + M1R $\rightleftharpoons$ AChM1R | 9.5e-08 | 0.0025 |
| 61 | I1 + PKAc $\rightleftharpoons$ I1PKAc | 1.4e-06 | 0.0056 | 131 | Gqabg + AChM1R $\rightleftharpoons$ AChM1RGq | 2.4e-05 | 0.00042 |
| 62 | I1PKAc $\rightleftharpoons$ Ip35 + PKAc | 0.0014 | 0.0 | 132 | Gqabg + M1R $\rightleftharpoons$ M1RGq | 5.76e-07 | 0.00042 |
| 63 | Ip35 + PP1 $\rightleftharpoons$ Ip35PP1 | 1e-06 | 1.1e-06 | 133 | ACh + M1RGq $\rightleftharpoons$ AChM1RGq | 3.96e-06 | 0.0025 |
| 64 | Ip35$X_{16}$ + PP2BCaMCa4 $\rightleftharpoons$ Ip35PP2B$Z_{16}$ | 9.625e-05 | 0.33 | 134 | AChM1RGq $\rightleftharpoons$ GqaGTP + AChM1R | 0.0005 | 0.0 |
| 65 | Ip35PP2B$X_{17}$ $\rightleftharpoons$ I1 + PP2B$X_{17}$ | 0.055 | 0.0 | 135 | ACh $\rightleftharpoons$ | 0.006 | 0 |
| 66 | PP1PP2BCaMCa4 $\rightleftharpoons$ PP1 + PP2BCaMCa4 | 0.0015 | 0.0 | 136 | Ca + PLA2 $\rightleftharpoons$ CaPLA2 | 6e-07 | 0.003 |
| 67 | GluR1$X_{18}$ + PKAc $\rightleftharpoons$ GluR1$Z_{18}$ | 4.02e-06 | 0.024 | 137 | CaPLA2 + Pip2 $\rightleftharpoons$ CaPLA2Pip2 | 2.2e-05 | 0.444 |
| 68 | GluR1$X_{19}$ $\rightleftharpoons$ GluR1$Y_{19}$ + PKAc | 0.006 | 0 | 138 | CaPLA2Pip2 $\rightleftharpoons$ CaPLA2 + AA | 0.111 | 0.0 |
| 69 | GluR1$X_{20}$ + CK$Y_{20}$ $\rightleftharpoons$ GluR1$Z_{20}$ | 2.224e-08 | 0.0016 | 139 | AA $\rightleftharpoons$ Pip2 | 0.001 | 0.0 |
| 70 | GluR1$X_{21}$ $\rightleftharpoons$ GluR1$Y_{21}$ + CK$Z_{21}$ | 0.0004 | 0 | 140 | PKCt + AA $\rightleftharpoons$ PKCp | 5e-09 | 1.76e-07 |

**(B)**

| | |
|---|---|
| $X_1 \in$ {LR, pLR} | $(X_{23}, Y_{23}, Z_{23}) \in$ { (_CKpCam, _S831, CKpCaMCa4), (_PKCt, |
| $X_2 \in$ {LR, pLR, ppLR, pppLR, R, pR, ppR, pppR} | _S831, PKCt), (_PKCp, _S831, PKCp), (_S845_CKpCam, _S845_S831, |
| $(X_3, Y_3) \in$ { (LRGibg, LR), (RGibg, R) } | CKpCaMCa4), (_S845_PKCt, _S845_S831, PKCt), (_S845_PKCp, |
| $X_4 \in$ {LR, R, pR} | _S845_S831, PKCp), (_memb_CKpCam, _memb_S831, CKpCaMCa4), |
| $X_5 \in$ {LR, pLR, ppLR, pR, ppR} | (_memb_PKCt, _memb_S831, PKCt), (_memb_PKCp, _memb_S831, PKCp), |
| $(X_6, Z_6) \in$ { (GsaGTP, GsaGTPCaMCa4), (GsaGTPGiaGTP, GsaGTPGiaGTPCaMCa4), ({}, CaMCa4) } | (_memb_S845_CKpCam, _memb_S845_S831, CKpCaMCa4), (_memb_S845_PKCt, _memb_S845_S831, PKCt), (_memb_S845_PKCp, |

*Table 3 continued on next page*

*Table 3 continued*

**(B)**

| | |
|---|---|
| $(\mathbf{X}_7, \mathbf{Z}_7) \in \{$ (AC1GsaGTPCaMCa4, AC1GsaGTPCaMCa4ATP), (AC1GsaGTPGiaGTPCaMCa4, AC1GsGiCaMCa4ATP), (AC1GiaGTPCaMCa4, AC1GiaGTPCaMCa4ATP), (AC1CaMCa4, AC1CaMCa4ATP), (AC8CaMCa4, AC8CaMCa4ATP) $\}$ | _memb_S845_S831, PKCp) $\}$ |
| $(\mathbf{X}_8, \mathbf{Y}_8, \mathbf{Z}_8) \in \{$ (GiaGTP, AC1GsaGTP, GTPGiaGTP), (GiaGTP, AC1CaMCa4, GTPCaMCa4), (AC1GiaGTP, GsaGTP, GTPGiaGTP) $\}$ | $(\mathbf{X}_{24}, \mathbf{Z}_{24}) \in \{$ (_S831, _S831_PKAc), (_memb_S831, _memb_S831_PKAc) $\}$ |
| $(\mathbf{X}_9, \mathbf{Z}_9) \in \{$ (AC1GsGiCaMCa4ATP, AC1GsaGTPGiaGTPCaMCa4), (AC8CaMCa4ATP, AC8CaMCa4) $\}$ | $(\mathbf{X}_{25}, \mathbf{Z}_{25}) \in \{$ (_S845, _S845_PP1), (_memb_S845, _memb_S845_PP1) $\}$ |
| $(\mathbf{X}_{10}, \mathbf{Z}_{10}) \in \{$ (CaMCa2, CaMCa3), (PP2BCaMCa2, PP2BCaMCa3) $\}$ | $(\mathbf{X}_{26}, \mathbf{Y}_{26}) \in \{$ (_S845_PP1, {}), (_memb_S845_PP1, _memb) $\}$ |
| $(\mathbf{X}_{11}, \mathbf{Z}_{11}) \in \{$ (CaMCa3, CaMCa4), (PP2BCaMCa3, PP2BCaMCa4) $\}$ | $(\mathbf{X}_{27}, \mathbf{Z}_{27}) \in \{$ (_S845_S831, _S845_S831_PP1), (_S831, _S831_PP1), (_memb_S845_S831, _memb_S845_S831_PP1), (_memb_S831, _memb_S831_PP1) $\}$ |
| $(\mathbf{X}_{12}, \mathbf{Z}_{12}) \in \{$ (2, CaMCa2), (4, CaMCa4) $\}$ | $(\mathbf{X}_{28}, \mathbf{Y}_{28}) \in \{$ (_S845_S831_PP1, _S845), (_S845_S831_PP1, _S831), (_S831_PP1, {}), (_memb_S845_S831_PP1, _memb_S845), (_memb_S845_S831_PP1, _memb_S831), (_memb_S831_PP1, _memb) $\}$ |
| $\mathbf{X}_{13} \in \{$pCaMCa4, CaMCa4$\}$ | |
| $(\mathbf{X}_{14}, \mathbf{Z}_{14}) \in \{$ ({}, PP1), (CaMCa4, CaMCa4PP1) $\}$ | $(\mathbf{X}_{29}, \mathbf{Z}_{29}) \in \{$ (_S845, _S845_PP2B), (_S845_S831, _S845_S831_PP2B), (_memb_S845, _memb_S845_PP2B), (_memb_S845_S831, _memb_S845_S831_PP2B) $\}$ |
| $(\mathbf{X}_{15}, \mathbf{Z}_{15}) \in \{$ (PP1, {}), (CaMCa4PP1, CaMCa4) $\}$ | |
| $(\mathbf{X}_{16}, \mathbf{Z}_{16}) \in \{$ ({}, CaMCa4), (PP1, P2BCaMCa4) $\}$ | $(\mathbf{X}_{30}, \mathbf{Y}_{30}) \in \{$ (_S845_PP2B, {}), (_S845_S831_PP2B, _S831), (_memb_S845_PP2B, _memb), (_memb_S845_S831_PP2B, _memb_S831) $\}$ |
| $\mathbf{X}_{17} \in \{$CaMCa4, P2BCaMCa4$\}$ | |
| $(\mathbf{X}_{18}, \mathbf{Z}_{18}) \in \{$ ({}, _PKAc), (_memb, _memb_PKAc) $\}$ | $\mathbf{X}_{31} \in \{\{\}$, _PKAc, _CKCam, _CKpCam, _CKp, _PKCt, _PKCp, _S831, _S831_PKAc, _S831_PP1$\}$ |
| $(\mathbf{X}_{19}, \mathbf{Y}_{19}) \in \{$ (_PKAc, _S845), (_S831_PKAc, _S845_S831), (_memb_PKAc, _memb_S845), (_memb_S831_PKAc, _memb_S845_S831) $\}$ | $\mathbf{X}_{32} \in \{\{\}$, _CKCam, _CKpCam, _CKp, _PKCt, _PKCp, _S831, _PP1, _S831_PP1, _PP2B, _S831_PP2B$\}$ |
| $(\mathbf{X}_{20}, \mathbf{Y}_{20}, \mathbf{Z}_{20}) \in \{$ ({}, CaMCa4, _CKCam), ({}, p, _CKp), (_S845, CaMCa4, _S845_CKCam), (_S845, p, _S845_CKp), (_memb, CaMCa4, _memb_CKCam), (_memb, p, _memb_CKp), (_memb_S845, CaMCa4, _memb_S845_CKCam), (_memb_S845, p, _memb_S845_CKp) $\}$ | $(\mathbf{X}_{33}, \mathbf{Y}_{33}, \mathbf{Z}_{33}) \in \{$ (PKAc, PDE4, PDE4), (PDE4cAMP, PKAc, _PDE4_cAMP) $\}$ |
| | $(\mathbf{X}_{34}, \mathbf{Y}_{34}) \in \{$ (PDE4, {}), (_PDE4_cAMP, cAMP) $\}$ |
| $(\mathbf{X}_{21}, \mathbf{Y}_{21}, \mathbf{Z}_{21}) \in \{$ (_CKCam, _S831, CaMCa4), (_CKp, _S831, p), (_S845_CKCam, _S845_S831, CaMCa4), (_S845_CKp, _S845_S831, p), (_memb_CKCam, _memb_S831, CaMCa4), (_memb_CKp, _memb_S831, p), (_memb_S845_CKCam, _memb_S845_S831, CaMCa4), (_memb_S845_CKp, _memb_S845_S831, p) $\}$ | $(\mathbf{X}_{35}, \mathbf{Y}_{35}) \in \{$ (GqaGTP, {}), (CaGqaGTP, Ca) $\}$ |
| | $(\mathbf{X}_{36}, \mathbf{Y}_{36}, \mathbf{Z}_{36}) \in \{$ ({}, t, _PKCt), ({}, p, _PKCp), (_memb, t, _memb_PKCt), (_memb, p, _memb_PKCp) $\}$ |
| $(\mathbf{X}_{22}, \mathbf{Y}_{22}, \mathbf{Z}_{22}) \in \{$ ({}, CKpCaMCa4, _CKpCam), ({}, PKCt, _PKCt), ({}, PKCp, _PKCp), (_S845, CKpCaMCa4, _S845_CKpCam), (_S845, PKCt, _S845_PKCt), (_S845, PKCp, _S845_PKCp), (_memb, CKpCaMCa4, _memb_CKpCam), (_memb, PKCt, _memb_PKCt), (_memb, PKCp, _memb_PKCp), (_memb_S845, CKpCaMCa4, _memb_S845_CKpCam), (_memb_S845, PKCt, _memb_S845_PKCt), (_memb_S845, PKCp, _memb_S845_PKCp) $\}$ | $(\mathbf{X}_{37}, \mathbf{Y}_{37}, \mathbf{Z}_{37}) \in \{$ (_PKCt, _S880, t), (_PKCp, _S880, p), (_memb_PKCt, _memb_S880, t), (_memb_PKCp, _memb_S880, p) $\}$ |
| | $(\mathbf{X}_{38}, \mathbf{Z}_{38}) \in \{$ (_S880, _S880_PP2A), (_memb_S880, _memb_S880_PP2A) $\}$ |
| | $(\mathbf{X}_{39}, \mathbf{Y}_{39}) \in \{$ (_S880_PP2A, {}), (_memb_S880_PP2A, _memb) $\}$ |
| | $\mathbf{X}_{40} \in \{\{\}$, _PKCt, _PKCp$\}$ |
| | $\mathbf{X}_{41} \in \{\{\}$, _PP2A$\}$ |

**Table 4.** List of initial concentrations of molecular species.

All non-mentioned species have an initial concentration of 0 nM.

| Species | | Conc. (nM) | Species | | Conc. (nM) | Species | | Conc. (nM) |
|---|---|---|---|---|---|---|---|---|
| CaOut | extracell. Ca$^{2+}$ | 1900000 | AMP | adenosine monophosphate | 980 | Pip2 | phosphatidylinositol 4,5-bisphosphate | 24000 |
| Leak | leak channels | 2000 | Ng | neurogranin | 20000 | PIkinase | phosphatidylinositol kinase | 290 |
| Calbin | calbindin | 150000 | CaM | calmodulin | 60000 | Ip3degPIk | Ip3-bound PI kinase | 400 |
| CalbinC | Ca$^{2+}$-bound calbindin | 15000 | PP2B | protein phosphatase 2B | 2300 | PKC | protein kinase C | 15000 |
| LOut | extracell. β-adr. ligand | 2500000 | CK | CaMKII | 23000 | DAG | diacylglycerol | 90 |
| Epac1 | Epac1 | 500 | PKA | protein kinase A | 6400 | DAGK | DAG kinase | 300 |
| PMCA | Ca$^{2+}$ pump | 22000 | I1 | inhibitor-1 | 2200 | DGL | DAG lipase | 1600 |
| NCX | Ca$^{2+}$ exchanger | 540000 | PP1 | protein phosphatase 1 | 1600 | CaDGL | Ca$^{2+}$-bound DAG lipase | 250 |
| L | β-adrenergic ligand | 10 | GluR1 | AMPAR subunit type 1 | 180 | DAGCaDGL | Ca$^{2+}$-and DAG-bound DAG lipase | 90 |
| R | β-adrenergic receptor | 1600 | GluR1_memb | membrane-inserted GluR1 | 90 | Ip3degrad | degraded Ip3 | 600 |
| Gs | S-type G-protein | 13000 | PDE4 | phosphodiesterase type 4 | 670 | GluR2 | AMPAR subunit type 2 | 14 |
| Gi | I-type G-protein | 2600 | fixedbuffer | immobile buffer | 500000 | GluR2_memb | membrane-inserted GluR2 | 256 |
| AC1 | adenylyl cyclase type 1 | 430.0 | mGluR | metab. glutamate receptor | 800 | PP2A | protein phosphatase 2A | 500 |
| ATP | adenosine triphosphate | 2000000 | GluOut | extracell. glutamate | 1000000 | M1R | acetylcholine receptor M1 | 450 |
| AC8 | adenylyl cyclase type 8 | 370 | Gqabg | Q-type G-protein | 1400 | PLA2 | phospholipase A2 | 1000 |
| PDE1 | phosphodiesterase type 1 | 12000 | PLC | phospholipase C | 250 | | | |

## Materials and methods

### Construction and calibration of the biochemically detailed model of post-synaptic plasticity in the cortex

We created a model of pathways leading from Ca$^{2+}$ inputs and activation of β-adrenergic receptors, metabotropic glutamate receptors, and muscarinic acetylcholine receptors to the phosphorylation and insertion of AMPARs into the membrane. We started by using the model of *Jędrzejewska-Szmek et al., 2017* for GluR1 phosphorylation at sites S831 and S845, which are phosphorylated by PKA and CaMKII, respectively, as a basis for our unified model. We added the mGluR- and M1 receptor-dependent pathways leading to PKC activation from *Kim et al., 2013* and *Blackwell et al., 2019*, respectively, and adopted the PKC-dependent endocytosis of GluR2 and reinsertion to the membrane from *Gallimore et al., 2018* as these pathways are critical for neocortical plasticity (*Seol et al., 2007*). As we included molecular species from different models and as we omitted certain molecular species that affected the dynamics of the underlying species but were not imperative for the pathways we wanted to describe, calibration of the model reactions was necessary. Following *Hayer and Bhalla, 2005*, we allowed the insertion and removal of GluR1 subunits to and from the membrane that depended on their state of S845 phosphorylation. We also allowed spontaneous membrane insertion of non-S845-phosphorylated GluR1; we chose the rate of this reaction so that on average one fifth of the (non-S845-phosphorylated) GluR1s were membrane-expressed in steady state, as suggested by experimental data (*Oh et al., 2006*). We adjusted the forward rate of GluR2 insertion to the membrane to decrease the proportion of membrane-inserted vs. internalised GluR2s (*Figure 11A*), following experimental data according to which 45% of GluR2s were membrane-

inserted at resting conditions (*Ashby et al., 2004*). We also adopted the three-step CaM activation of *Gallimore et al., 2018* instead of the two-step activation of *Jędrzejewska-Szmek et al., 2017* where the reaction rates of CaM binding two $Ca^{2+}$ ions were linearly dependent on the number of $Ca^{2+}$ ions. The response curve for CaM activation by $Ca^{2+}$ was steeper in this model (*Figure 11B*), which was in better accordance with recent experimental data (*Hoffman et al., 2014*). As a result, our model predicted a more prominent activation of CaM in response to a large influx of $Ca^{2+}$ but milder activation in respose to small $Ca^{2+}$ influx than the model of *Jędrzejewska-Szmek et al., 2017* (*Figure 11C*).

To allow long-term activation of PKC, we adopted a persistently activated form of PKC, mediated by arachidonic acid (AA), from *Kotaleski et al., 2002*. We calibrated the rates of this reaction as follows. The backward rate was chosen so that approximately 90% of PKC would be active after 10 min, inspired by experimental data of *Shirai et al., 1998*. The forward rate was chosen so that LFS with effective PLC activation led to approximately 50% of the GluR2s being phosphorylated (*Figure 11D*), following experimental data (*Ahmadian et al., 2004*). The implications of these adjustments on the dynamics of transiently activated PKC, persistently activated PKC and GluR2 S880 phosphorylation are illustrated in *Figures 11E, F and G*, respectively.

We adopted the simplified, mass-action law-based PKA activation model (reaction 59; *Table 3*) from *Williamson et al., 2009* (where it was called model 'C') instead of the 2-stage, linearised cAMP-binding of PKA in *Jędrzejewska-Szmek et al., 2017* and *Blackwell et al., 2019*. We fitted the forward rate to data simulated with the original model (*Figure 11H*) to produce a longer-lasting S845 phosphorylation (typically, >10 min duration of S845 phosphorylation was observed *Seol et al., 2007*; *Xue et al., 2014*) in the 4xHFS protocol (*Figure 11I*). To account for the experimental observation the PKC phosphorylates GluR1 at site S831 *Roche et al., 1996*, we added this reaction using the rates identical to those of GluR1-S831 phosphorylation by phosphorylated CaMKII (see reactions 71–72). However, the presence of this reaction did not have a large effect on the S831 phosphorylation of GluR1 under standard conditions (*Figure 11J*). Finally, we introduced an immobile $Ca^{2+}$ buffer with a $Ca^{2+}$ binding rate of 0.0004 1/(nM ms), a release rate 20.0 1/ms, and an initial concentration of 500 µM (these values are within the range of experimental observations and values used in models *Matthews and Dietrich, 2015*). The model reactions are described in *Table 3* and the initial concentrations are listed in *Table 4*.

Throughout the work, we simulated the signalling pathways in a single compartment representing a dendritic spine of size 0.5 µm$^3$. In reality, some of the molecular species are prevalently present in the cytosol, some attached to the membrane, some in the extracellular medium in an immediate vicinity to the membrane, and others outside the cell further away from the synaptic cleft (free in the extracellular medium or sequestered to other cells). As commonly done in the field, we solved this problem by introducing species that represent a molecular species confined in a particular location: reactions 1–6 describe the extrusion of $Ca^{2+}$ from the cytosol into the extracellular medium, reactions 8, 95, and 135 describe the escape of ligands from the vicinity of the synapse, and reactions 80–81 and 128–129 for the translocation of the AMPARs to/from the membrane (*Table 3*). All stimulations start after 4040 s of simulation without inputs, which is sufficient for attaining a steady state for all species (*Figure 11—figure supplement 1*).

## Statistical model for numbers of AMPAR tetramers at the membrane and the total synaptic conductance

AMPARs have different conductances depending on their subunit composition and phosphorylation state (*Oh and Derkach, 2005*), but it is challenging to take this into account in models that include a large number of receptor subunits. In our model, AMPAR subunits GluR1 and GluR2 can be in one of 21 or five states, respectively, when counting all the different phosphorylation states and bonds with other molecules (*Table 3*), which leads to 28$^4$ possible types of tetramers. This makes it virtually impossible to model the dynamics of AMPAR tetramer assembly using the mass-action law-based approach where the concentration of each type of species is monitored (*Michalski and Loew, 2012*). To avoid this problem, we used a statistical model that estimated the numbers and types of different types of AMPAR tetramers given the numbers of GluR1 and GluR2 subunits located at the membrane.

We assumed that the composition of AMPAR tetramers is random such that there is no preference of one type of subunit being more likely to bind with any other type of subunit. Thus, the

probability of a tetramer being a GluR1 homomer without any S831-phosphorylated subunits is approximately

$$p_{\text{GluR1 homomer, non-phos.}} = \frac{\binom{N_1 - N_{1,\text{phos.}}}{4}}{\binom{N_1 + N_2}{4}} \approx \frac{(N_1 - N_{1,\text{phos.}})^4}{(N_1 + N_2)^4} \tag{1}$$

where $N_1$ and $N_2$ are the numbers of GluR1 and GluR2 subunits bound to the membrane, respectively, and $N_{1,\text{phos.}}$ is the number of S831-phosphorylated GluR1 subunits at the membrane (note that the $N_{1,\text{phos.}}$ subunits are included in all GluR1 subunits, that is, $N_{1,\text{phos.}} \leq N_1$). Accordingly, the probabilities of a tetramer being a GluR1 homomer with at least one S831-phosphorylated subunit, a GluR2 homomer, or a heteromer, are approximately:

$$rlp_{\text{GluR1 homomer, phos.}} = \frac{N_1^4 - (N_1 - N_{1,\text{phos.}})^4}{(N_1 + N_2)^4} \tag{2}$$

$$p_{\text{GluR2 homomer}} = \frac{N_2^4}{(N_1 + N_2)^4} \tag{3}$$

$$p_{\text{heteromer}} = 1 - \frac{N_1^4}{(N_1 + N_2)^4} - \frac{N_2^4}{(N_1 + N_2)^4}. \tag{4}$$

The number of membrane-bound tetramers that the $N_1$ GluR1 subunits and $N_2$ GluR2 subunits at the membrane can form is $\frac{N_1 + N_2}{4}$. Here, we ignore the unpaired subunits by estimating that $\frac{N_1 + N_2}{4} \approx \lfloor \frac{N_1 + N_2}{4} \rfloor$ — we also disregard the states of the non-membrane-bound subunits as they are not assumed to contribute to the synaptic conductance. This gives us approximate values for expected numbers of different types of tetramers on the membrane:

$$N_{\text{GluR1 homomer, non-phos.}} = \frac{N_1 + N_2}{4} p_{\text{GluR1 homomer, non-phos.}}$$

$$N_{\text{GluR1 homomer, phos.}} = \frac{N_1 + N_2}{4} p_{\text{GluR1 homomer, phos.}}$$

$$N_{\text{GluR2 homomer}} = \frac{N_1 + N_2}{4} p_{\text{GluR2 homomer}}$$

$$N_{\text{heteromer}} = \frac{N_1 + N_2}{4} p_{\text{heteromer}}$$

These estimates allow us to determine the total maximal synaptic conductance as the sum of the numbers of these tetramers multiplied with the corresponding single-channel conductances:

$$\begin{aligned} g_{\text{syn}} = 12.4\text{pS} \times N_{\text{GluR1 homomer, non-phos.}} &+ 18.9\text{pS} \times N_{\text{GluR1 homomer, phos.}} \\ &+ 2.2\text{pS} \times N_{\text{GluR2 homomer}} + 2.5\text{pS} \times N_{\text{heteromer}}. \end{aligned} \tag{5}$$

The single-channel conductance values 12.4 pS, 18.9 pS, 2.2 pS, and 2.5 pS are taken from experimental data (*Oh and Derkach, 2005*).

## Modelling the Ca$^{2+}$ inputs and neuromodulatory inputs

We modelled the neurotransmission to the post-synaptic spine as fluxes of Ca$^{2+}$ ions, β-adrenergic ligand, glutamate, and acetylcholine (labelled as Ca, L, Glu, and ACh, respectively, in *Table 3*). We used various stimulation paradigms: In sections '*Ca$^{2+}$ activates multiple pathways that regulate the post-synaptic plasticity in cortical PCs*' and '*The model predicts multimodal, protein concentration- and neuromodulation-dependent rules of plasticity*', long-lasting, single pulses of input species were applied. In sections '*High-frequency stimulation (HFS) causes LTP and low-frequency stimulation (LFS) causes LTD in GluR1-GluR2-balanced synapses*' and '*A parametric analysis confirms the robustness of the model*', we used the following repeated stimulus protocols: HFS — 100 pulses of Ca$^{2+}$ (3 ms), repeated at 100 Hz; 4xHFS — 4 trains of HFS, separated by 3 s of quiescence; LFS — 900 pulses of Ca$^{2+}$ (3 ms), repeated at 5 Hz. Unless otherwise stated, each Ca$^{2+}$ pulse was accompanied by a 3 ms pulse of β-adrenergic ligand, glutamate, and acetylcholine. The activation of cholinergic and noradrenergic terminals by electrical stimulation is supported by experimental data in, e.g., slices of mouse prefrontal cortex (*Mundorf et al., 2001*). In section '*The model flexibly reproduces data from*

*various cortical LTP/LTD experiments*', the same approach was used, but the frequencies and numbers of repetitions of the inputs were taken from the experiments (see *Table 2*).

In section '*Paired pre- and post-synaptic stimulation induces PKA- and PKC-dependent spike-timing-dependent plasticity (STDP) in GluR1-GluR2-balanced synapses*' (and *Figure 3—figure supplement 3* of section '*High-frequency stimulation (HFS) causes LTP and low-frequency stimulation (LFS) causes LTD in GluR1-GluR2-balanced synapses*'), we used a multicompartmental model of a layer 2/3 pyramidal cell (*Markram et al., 2015*) (L23_PC_cADpyr229_1) to determine the amplitudes and time courses of the $Ca^{2+}$ inputs conducted by NMDARs when different stimulus patterns were applied. This model included the fast $Na^+$ current $I_{Na,t}$, Shaw-related $K^+$ current $I_{Kv3.1}$, muscarinic $K^+$ current $I_m$, and hyperpolarization-activated cyclic nucleotide-gated current $I_{HCN}$ in the apical dendrite. The axo-somatic region contained all these (except for $I_m$) as well as the low and high-voltage activated $Ca^{2+}$ currents $I_{CaLVA}$ and $I_{CaLVA}$, the small-conductance $Ca^{2+}$-dependent $K^+$ current $I_{SK}$, the transient $K^+$ current $I_{K,t}$, and the persistent $Na^+$ and $K^+$ currents $I_{Na,p}$ and $I_{K,p}$ in the axo-somatic region (*Markram et al., 2015*). We placed 10 post-synaptic spines, consisting of a 0.5 µm long and 0.1 µm thick neck and a 0.4 µm long and 0.4 µm thick head, to the proximal apical dendrite (250–300 µm from the soma). For an analysis of the effects of synapse location on $Ca^{2+}$ flux and plasticity in cortical pyramidal cells, see *Ebner et al., 2019*. Each spine was equipped with the AMPA–NMDA synapse model of *Hay and Segev, 2015* using the NMDA gating mechanism of *Spruston et al., 1995* and an adjustment in the pre-synaptic resource update (*Mäki-Marttunen et al., 2019*). The ten synapses were synchronously stimulated but the glutamate release was probabilistic, the release events at different synapses being independent of each other as in *Hay and Segev, 2015*). We set the maximal AMPAR conductance to 0.001 µS, a value typically used in computational neuron models. We set the maximal conductance of NMDARs to 0.0032 µS to compensate for the lack of the slow component in the model of the NMDA current (*Markram et al., 2015*) — a value significantly smaller or larger than this abolished the LTP or LTD, respectively, in our STDP model, while the AMPAR conductance was a less crucial parameter (*Figure 11—figure supplement 2*).We estimated the numbers of $Ca^{2+}$ ions entering into the post-synaptic spine across time, and used these numbers as the input to the biochemical model. In section '*Paired pre- and post-synaptic stimulation induces PKA- and PKC-dependent spike-timing-dependent plasticity (STDP) in GluR1-GluR2-balanced synapses*', following the experiments of *Seol et al., 2007*, we used extracellular $[Mg^{2+}]$ of 1.0 mM and a 1 Hz pre-synaptic stimulation, where each pulse was paired with a burst of post-synaptic stimulus currents (*Seol et al., 2007*). For *Figure 3—figure supplement 3* of section '*High-frequency stimulation (HFS) causes LTP and low-frequency stimulation (LFS) causes LTD in GluR1-GluR2-balanced synapses*', following the experiments of *Heusler et al., 2000* we used $[Mg^{2+}]$ of 1.3 mM and solely pre-synaptic stimulation of one of the two stimulus protocols: 6xHFSt — 10 bursts of 4 pulses (at 100 Hz), repeated every 100 ms, and the whole train repeated 6 times every 10 s; LFS-1Hz — 1800 pulses delivered at a frequency of 1 Hz.

The effects of LTP/LTD on the size of $Ca^{2+}$ inputs were not considered in this work.

## Parameter alterations and model fitting

In sections '*The model predicts multimodal, protein concentration- and neuromodulation-dependent rules of plasticity*' and '*The model flexibly reproduces data from various cortical LTP/LTD experiments*', we altered the initial concentrations of many proteins to explore the parameter space or to perform model fitting. We chose to fit protein concentrations instead of reaction rates, since the reaction rates can be considered to be the same across cell types while the protein expression is known to be cell-type and age-dependent. This is analogous to fitting maximal conductances that correlate with ion-channel densities in Hodgkin-Huxley-type models instead of the ion-channel activation and inactivation curve parameters as is usually done in the fitting of biophysically detailed neuron models. The concentrations of upstream PKA-pathway proteins R (β-adrenergic receptor), Gs, AC1, and AC8 were varied in proportion using a factor parameter $f_{PKA} \in [0, 2]$, and, likewise, the concentrations of upstream PKC-pathway proteins mGluR, M1, Gq, and PLC using a factor parameter $f_{PKC} \in [0, 2]$. Furthermore, in section '*The model flexibly reproduces data from various cortical LTP/LTD experiments*', CaM and CaMKII were altered in proportion by a factor $f_{CaMKII} \in [0, 2]$, phosphatases PP1 and PP2B by a factor $f_{PP} \in [0, 2]$, and phosphodiesterases PDE1 and PDE4 by a factor $f_{PDE} \in [0, 2]$. In both sections, the rapidity of $Ca^{2+}$ extrusion was varied by altering the concentration of NCX, and in section '*The model flexibly reproduces data from various cortical LTP/LTD*

*experiments*', the concentrations of PKA and PKC were varied in addition to the upstream proteins — these concentrations were varied within the interval from 0 to double the original value. For the multi-objective optimisation in section '*The model flexibly reproduces data from various cortical LTP/LTD experiments*', we used the Python implementation (published by the authors of *Bahl et al., 2012*) of the non-dominated sorting genetic algorithm II (NSGA-II) (*Deb et al., 2002*) with population size 1000. To restrict to physiologically realistic $Ca^{2+}$ dynamics, we disregarded the data where free $Ca^{2+}$ concentrations rose above 2 μM for one or more levels of $Ca^{2+}$ input in section '*The model predicts multimodal, protein concentration- and neuromodulation-dependent rules of plasticity*'. In a similar manner, in section '*The model flexibly reproduces data from various cortical LTP/LTD experiments*', we introduced an objective function that penalised parameter sets that produced $Ca^{2+}$ transients larger than 2 μM.

### Simulation software and code accessibility

For deterministic simulations of intracellular signalling, we used the NEURON simulator with the reaction-diffusion (RxD) extension (*McDougal et al., 2013*). For stochastic simulations, we used NeuroRD software (https://github.com/neurord). In both types of simulations, we used adaptive time-step integration methods. The NEURON simulator was also used for simulating the multicompartmental model of layer 2/3 pyramidal cell in section '*Paired pre- and post-synaptic stimulation induces PKA- and PKC-dependent spike-timing-dependent plasticity (STDP) in GluR1-GluR2-balanced synapses*'. The full model along with the fitting and data-analysis algorithms (Python scripts) that were used in this study are publicly available in ModelDB at http://modeldb.yale.edu/260971.

## Acknowledgements

UNINETT Sigma2 resources (project NN9529K) were used for simulations. Funding: Research Council of Norway (248828), European Union Horizon 2020 Research and Innovation Programme under Grant Agreement No. 785907 [Human Brain Project (HBP) SGA2].

## Additional information

### Funding

| Funder | Grant reference number | Author |
| --- | --- | --- |
| Research Council of Norway | 248828 | Tuomo Mäki-Marttunen<br>Andrew G Edwards<br>Gaute T Einevoll |
| Horizon 2020 - Research and Innovation Framework Programme | 785907 | Gaute T Einevoll |

The funders had no role in study design, data collection and interpretation, or the decision to submit the work for publication.

### Author contributions

Tuomo Mäki-Marttunen, Conceptualization, Software, Validation, Investigation, Methodology, Writing - original draft, Project administration, Writing - review and editing; Nicolangelo Iannella, Andrew G Edwards, Gaute T Einevoll, Formal analysis, Methodology; Kim T Blackwell, Conceptualization, Software, Supervision, Methodology

### Author ORCIDs

Tuomo Mäki-Marttunen https://orcid.org/0000-0002-7082-2507
Gaute T Einevoll http://orcid.org/0000-0002-5425-5012
Kim T Blackwell http://orcid.org/0000-0003-4711-2344

### Decision letter and Author response

Decision letter https://doi.org/10.7554/eLife.55714.sa1

Author response https://doi.org/10.7554/eLife.55714.sa2

## Additional files

### Data availability

All data generated or analysed during this study are included in the manuscript and supporting files. Simulation scripts can be found at http://modeldb.yale.edu/260971.

The following dataset was generated:

| Author(s) | Year | Dataset title | Dataset URL | Database and Identifier |
|---|---|---|---|---|
| Mäki-Marttunen T, Iannella N, Edwards AG, Einevoll GT, Blackwell KT | 2020 | Biochemically detailed model of LTP and LTD in a cortical spine (Maki-Marttunen et al 2020) | http://modeldb.yale.edu/260971 | ModelDB, 260971 |

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
