## [Decision Letter]

**Acceptance summary:**

This is a very ambitious project, that models many of the different molecular interactions that can affect synaptic plasticity. The inclusion of neuromodulators is important, novel and can account for recent data about the role of neuromodulators. The statistical model of AMPA receptors is interesting novel and practical. It is approximation that should be validated experimentally. The authors take seriously the role of data. Due to the models high dimensionality and complexity it is hard to currently validate many assumptions. The significance of this paper is not necessarily the specific assumptions being made, rather it is that many pathway previously ignored are now included, and are shown to significantly contribute.

**Decision letter after peer review:**

Thank you for submitting your article "A unified computational model for cortical post-synaptic plasticity" for consideration by *eLife*. Your article has been reviewed by two peer reviewers, including Harel Z Shouval as the Reviewing Editor and Reviewer #1, and the evaluation has been overseen by Michael Frank as the Senior Editor.

The reviewers have discussed the reviews with one another and the Reviewing Editor has drafted this decision to help you prepare a revised submission.

As the editors have judged that your manuscript is of interest, but as described below that additional experiments are required before it is published, we would like to draw your attention to changes in our revision policy that we have made in response to COVID-19 (https://elifesciences.org/articles/57162). First, because many researchers have temporarily lost access to the labs, we will give authors as much time as they need to submit revised manuscripts. We are also offering, if you choose, to post the manuscript to bioRxiv (if it is not already there) along with this decision letter and a formal designation that the manuscript is 'in revision at *eLife*'. Please let us know if you would like to pursue this option. (If your work is more suitable for medRxiv, you will need to post the preprint yourself, as the mechanisms for us to do so are still in development.)

Summary:

This is a very ambitious project, that models many of the different molecular interactions that can affect synaptic plasticity. The inclusion of neuromodulators is important and relatively novel. The statistical model of AMPA receptors is interesting, novel and practical. The authors take seriously the role of data. It is quite a heroic effort, but on the other hand the model is hard to verify, it is hard to extract more general principles, and the predictions of the model are limited. This limits the overall impact of the work.

The model is very complex, and almost none of the reaction coefficients are known. There are so many parameter combinations that can yield the same results. What do we actually learn from this? This is an essential part of this model, not much can be done to address this.

Spine model should be used in all simulations of plasticity protocol. It is not clear and how the magnitude of the calcium currents through the NMDAR, and of efflux through diffusion and pumps are calibrated.

It is not clear if all initial conditions are at the fixed points.

The dynamics of the neuromodulators are odd and sometimes unclear. They should be justified, explained better or corrected if these dynamics cannot be justified. Does the application of neuromodulator to the bath on its own change the efficacies?

Essential revisions:

1) All simulations of plasticity protocols should use the spine model, and it should be explained how the NMDA currents in the spine model are calibrated.

2) All simulations should start from a fixed point of the model.

3) The neuromodulator dynamics should be better explained and justified. Does it make sense to have square neuromodulator pulses? What are the neuromodulator dynamics in the STDP experiments? Especially must take into account that in the Seol et al., 2007 paper neuromodulators are bath applied. Does bath application of neuromodulators, alone, without stimulation change the efficacies?

Reviewer #1:

This is a very ambitious project, almost heroic. The attempt is to carry out true quantitative modeling of many of the signal transduction pathways involved in LTP/LTD in neocortex. The work here is very detailed and extensive, and there are many interesting and novel components to this work. However, I am not yet convinced that this goal is feasible. The model is very complex, there is a huge number of unknown parameters and I am not sure that I understand all of the methods and therefore do not know if they are appropriate.

The good:

1) The inclusion of neuromodulators is important and relatively novel.

2) The statistical model of AMPA receptors is interesting novel and practical.

3) The authors take seriously the role of data.

The bad:

1) The model is very complex, and almost none of the reaction coefficients are known. There are so many parameter combinations that can yield the same results. What do we actually learn from this?

2) All simulation of plasticity including LFS and HFS should be carried out on the basis if the spine model in Neuron. It is not clear to me though how that was calibrated.

3) The neuromodulator dynamics, which are pulses for every Ca pulse, are problematic, and not justified. It is unclear to me what the NM dynamics are in the subsection “Paired-pulse stimulus protocol induces PKA- and PKC-dependent spike-timing-dependent plasticity (STDP) in GluR1-GluR2-balanced synapses”

Detailed points – Major:

The good:

4) The inclusion of neuromodulators and their effects on the PKC and PKA pathways is important. Most previous models have ignored these effects, but results in adult neocortical slices have proven that these pathways are essential for LTP and LTD.

Most previous models have ignored the role of the different subunits in the AMPA receptor heteromers. Modeling of these receptors directly as independent species in a mass action approach is nearly impossible due to the huge number of possible combinations. The statistical model proposed is a novel idea that has significant practical advantages. There are assumptions that go into this, which the authors acknowledge. Nevertheless, it is a novel and sensible approach. I think a separate work could simply be based on this model and the statistical testing of its validity.

These authors take seriously the role of data and test the model (with different parameters) against many different experimental results.

The major problems:

5) This is a very complex model with 140 reactions and 47 types of elementary molecules which could be in many states. A tiny fraction of the reaction coefficients have been measured, even if they have it might not be relevant for in vivo conditions. The determination of these coefficients in the paper does not seem to depend on measured coefficients, but instead seems to depend on these coefficients being appropriate for matching experimental results. As we know, and as is shown for example in Figure 7, there is a very large space of parameter combinations that can produce very similar experimental results. What do we actually learn from all of this then? This problem is actually made clear in Figure 7, what are the authors trying to clarify in this very complex figure?

6) The real complete model is the one that uses the Calcium transients from a Neuron simulation, subsection “Paired-pulse stimulus protocol induces PKA- and PKC-dependent spike-timing-dependent plasticity (STDP) in GluR1-GluR2-balanced synapses” and Figure 6. Only here there is a possibility that realistic calcium transients are used, and in principle this should have been used from all comparisons to data. Not only STDP protocols have real synapses, HFS and LFS stimuli also use real synapses. However, it is not clear to me how this synapse model was calibrated. Peak Ca influx rate here is much larger than for the other cases, more than an order of magnitude larger than in Figure 3 and much larger than in Figures 4, 5 as well. How was this calibrated? Was this based on the number of NMDAR and this influx through each? If so, what are these numbers? An alternative way of calibrating the spine model is to use estimates of calcium influx from Ca imaging, for example from Sabatini and Svoboda, 2002. A third option is to use whatever works, however then even this elementary component of the model is not based on biophysical realism. Several details and references are given in the subsection “Modelling the Ca^2+^ inputs and neuromodulatory inputs”, but this is still not clear to me. It would also be useful to see what the spine voltages are and how they affect the Mg block. Why is there more calcium influx at -30 ms vs. only presynaptic stimulation? Is the -30 ms measured from the first or last spike in the 4-spike train? What is it about a spike that occurred 30 ms prior to the presynaptic stimulus that affects calcium influx?

7) Neuromodulator dynamics. It is important that neuromodulators have been included here. However, the assumptions about their dynamics do not make sense to me. In several sections neuromodulators pulses are assumed to follow the Ca pulses? What is the logic here, that stimulation of axons also causes neuromodulator release? This clearly does not seem to match experiments like Seol et al., 2007, where neuromodulators are bath applied and should just be at a constant level. Is there any evidence that in other slice experiments neuromodulators are indeed release at every pulse? What about the culture experiments? It is also not clear to me what is done with neuromodulators in the subsection “Paired-pulse stimulus protocol induces PKA- and PKC-dependent spike-timing-dependent plasticity (STDP) in GluR1-GluR2-balanced synapses”, where Ca transients were taken from the spine model. Here it is clear to me that constant Neuromodulator levels should be used as in the experiment.

8) Are initial conditions steady at steady state for each parameter combination? It is not clear to me if all plasticity simulations are started at the steady levels of the system for the given parameter set? Are they?

Reviewer #2:

1) The authors develop a complicated model of the biochemical pathways underlying LTP and LTD. It is quite a heroic effort, but on the other hand the model is hard to verify, it is hard to extract more general principles, and the predictions of the model are limited. This limits the overall impact of the work.

2) Some of the design decisions are hard to follow.

For instance, why was the CaM activation made steeper?

3) I also did not understand how the model fitting was done by changing initial concentrations (Materials and methods). Changing the reaction rates would be a more conventional way. I wonder how these concentrations develop in the absence of stimulation. Do they stay the same, or do they have to be clamped to certain values? Yet in the subsection “The model flexibly reproduces data from various cortical LTP/LTD experiments” other variables are changed to fit the data (what are 'factors for the protein concentrations'?).

4) The model's complexity make it difficult to understand it's properties. For instance, does CaMKII act as a switch, and is the expression essentially binary (O'Connor and Wang)? Does it fit the observations of Nevian and Sakmann?

5) The STDP curves look odd, with no below baseline LTD for short negative intervals.

6) Does the last sentence of the subsection “High-frequency stimulation causes LTP and low-frequency stimulation causes LTD in GluR1-GluR2-balanced synapses” really imply a causal relation, so that GuR2 endocytosis *leads* to potentiation or depression? If so, the mechanism was not clear to me.

7) The y-axis labels on the plots are odd. In Figure 2 they put the quantity as the plot label, and the units as axis label. The authors do it correctly on the x-axis.

In other figures other conventions are followed.

---

## [Author Response]

Essential revisions:1) All simulations of plasticity protocols should use the spine model, and it should be explained how the NMDA currents in the spine model are calibrated.

Our STDP simulations do indeed use the electrical model to determine calcium input, and now we have added simulations of HFS and LFS using calcium inputs from the multicompartmental neuron model. We have explained the calibration of the model in more detail, and we have added a figure that justifies the chosen parameter values.

2) All simulations should start from a fixed point of the model.

We have confirmed that the simulations start from a steady state, and added a figure (Figure 11—figure supplement 1) to show this.

3) The neuromodulator dynamics should be better explained and justified. Does it make sense to have square neuromodulator pulses? What are the Neuromodulator dynamics in the STDP experiments? Especially must take into account that in the Seol et al., 2007 paper neuromodulators are bath applied. Does bath application of neuromodulators, alone, without stimulation change the efficacies?

We have added justification to the dynamics of the neuromodulators, as explained in more detail below. Furthermore, we have now simulated the same bath-application protocol as used in Seol et al., 2007. We also showed that the type of dynamics (pulse-associated or bath-applied) of the neuromodulators does not make a large difference to the results of the model. The bath application of neuromodulators alone did not cause plasticity, as we now mention in the text.

Reviewer #1:[…] 1) The model is very complex, and almost none of the reaction coefficients are known. There are so many parameter combinations that can yield the same results. What do we actually learn from this?

We thank the reviewer for this comment. We agree with the reviewer that the model is complex when measured in numbers of reactions or equations. Nevertheless, all dynamics of the modeled molecular species follow the law of mass action, which gives the model an intrinsic modular property: if future modelers want to simplify parts of it or (as we believe to be more likely) extend it to take into account even more details, they will only have to replace a set of targeted species and their interactions with the rest of the model species without refitting the whole model. This is an important difference compared to simpler models with few parameters and non-mass-action-law type of interactions that have been particularly fitted to describe the plasticity-related phenomena.

It would be ideal to have all reaction rates explicitly fitted to data obtained from carefully constrained experiments, but such data are scarcely published. Nonetheless, rate constants for many of the reactions were constrained by data. We used off-the-shelf models of the biochemistry of PKA, PKC and CaMKII pathways where the reaction rates and initial concentrations were usually fitted directly to enzyme kinetic or FRET imaging data.

What we can learn from such a complex model is the interplay of different molecular species in the induction of post-synaptic plasticity in the neocortex. This is an important research topic in polygenic mental disorders, schizophrenia in particular, where subtle changes (due to common gene variants) in plasticity-related proteins can lead to pathological cellular and network physiology. Our model can provide biochemically detailed model predictions that are urgently needed in discovering mechanisms for these disorders. A step toward this direction is Figure 7, which, although it shows that various combinations of model parameters can result in similar plasticity curve, also pinpoints the most sensitive molecular species in different types of plasticity. Many of these sensitive species are indeed in the groups of proteins encoded by risk genes of schizophrenia (see Figure 1 in Devor et al., 2017). Like any model predictions, such biochemistry-based predictions on disease pathophysiology may be biased due to under-constrained model parameters (e.g., rate constants) and thus should be tested experimentally – until that happens, they can serve as useful and potentially ground-breaking proof-of-principle mechanisms.

2) All simulation of plasticity including LFS and HFS should be carried out on the basis if the spine model in Neuron. It is not clear to me though how that was calibrated.

Using our biophysically detailed NEURON simulations, we have now calculated the Ca^2+^ transients resulting from HFS and LFS stimulation protocols. In particular, we showed that our model reproduced the LTP caused by a theta-HFS protocol (6xHFSt) and the LTD caused by a 1-Hz LFS protocol (LFS^-1^Hz) as described in Heusler et al., 2000. We have extended the description of the model calibration for this. We now write:

“In the above analyses, we used brief square-pulse fluxes of Ca^2+^to the synapse model, which is a simple representation of inputs during synaptic plasticity induction protocols. […] In accordance with Figure 3 and experimental data from somatosensory cortex [Heusler et al., 2000], our model predicted that 6xHFSt induced LTP whereas LFS-1Hz induced LTD (Figure 3—figure supplement 3).”

As both reviewers noted, the biochemical model is in itself rather complex, and we try to avoid bringing extra complexity to the manuscript when not absolutely necessary. Moreover, when we talk about the proposed “unified” model that can provide explanations for a range of forms of plasticity in the neocortex, we mainly refer to the biochemical network of the synapse, not the combination of the biochemical network and the multicompartmental biophysical model (the latter of which is known to vary between pyramidal cell types and cortical areas). For these reasons, we consider some of the simulation experiments added in response to the reviewer’s comments supplementary, and thus included these results in the supplementary material. We also believe it helps the reader that we start from generic HFS-induced LTP and LFS-induced LTD rather than a specific LTP and LTD in the somatosensory cortex.

3) The Neuromodulator dynamics, which are pulses for every Ca pulse, are problematic, and not justified. It is unclear to me what the NM dynamics are in the subsection “Paired-pulse stimulus protocol induces PKA- and PKC-dependent spike-timing-dependent plasticity (STDP) in GluR1-GluR2-balanced”.

Little is known about the dynamics of neuromodulatory inputs during different stimulation protocols. We know that norepinephrinic projections span all layers of cortex (Simpson KL, Waterhouse BD, Lin RC. Anat Rec A Discov Mol Cell Evol Biol. 2006;288(2):166‐173), and that cholinergic projections, depending on the source, may span either all layers or only deep layers of the cortex (Obermayer J, Verhoog MB, Luchicchi A, Mansvelder HD. Front Neural Circuits. 2017;11:100). Thus, it is reasonable to hypothesize that HFS and LFS, when applied to the cortex, also cause rhythmic neurotransmitter release. There indeed exists evidence that noradrenaline and acetylcholine are released in response to electrical stimulation in cortical slices (see e.g. Mundorf et al., 2001; Vizi ES, Zsilla G, Caron MG, Kiss JP. J Neurosci. 2004;24(36):7888‐7894). Thus, our model of short pulses that accompany the Ca^2+^ pulses is justified experimentally. Nevertheless, the amplitude of this release may vary between cortical areas and slices prepared in different ways, as some experiments (such as Seol et al., 2007) indicate that bath application of neuromodulators (or corresponding agents) is required for activation of PKA or PKC pathways – this is why we allowed the amplitudes of the stimulus-associated neuromodulatory fluxes to vary in our model fitting to experimental data (Figure 9).

To justify these choices, we now write:

“Unless otherwise stated, each Ca^2+^ pulse was accompanied by a 3-ms pulse of β-adrenergic ligand, glutamate, and acetylcholine. The activation of cholinergic and noradrenergic terminals by electrical stimulation is supported by experimental data in, e.g., slices of mouse prefrontal cortex [Mundorf et al., 2001].”

To show that our modelling results are not dependent on the type of neuromodulator fluxes (pulse-associated or bath-applied), we have performed the HFS and LFS simulation experiments using the same amounts of neuromodulators but with different temporal distributions. We now write:

“The HFS-induced LTP and LFS-induced LTD of Figure 3 could also be reproduced with alternative durations of neuromodulator inputs, including 10-minute bath applications (Figure 3—figure supplement 4). These results indicate that our model can reproduce HFS-induced LTP and LFS-induced LTD also when using realistic NMDAR-conducted Ca^2+^ transients and that these forms of plasticity are robust to the temporal profile of the neuromodulatory inputs.”

We have now clarified the neuromodulator dynamics in the subsection “Paired pre- and post-synaptic stimulation induces PKA- and PKC-dependent spike-timing-dependent plasticity (STDP) in GluR1-GluR2-balanced synapses”. We now write:

“When added as bath application, the β-adrenergic and cholinergic ligands were simulated by prolonged injections of 50 particles/s for 10 min, starting 8 min before the STDP protocol – these neuromodulators alone (without the electric stimulation-mediated Ca^2+^ inputs) did not cause synaptic plasticity.”

4) The inclusion of neuromodulators and their effects on the PKC and PKA pathways is important. Most previous models have ignored these effects, but results in adult neocortical slices have proven that these pathways are essential for LTP and LTD.Most previous models have ignored the role of the different subunits in the AMPA receptor heteromers. Modeling of these receptors directly as independent species in a mass action approach is nearly impossible due to the huge number of possible combinations. The statistical model proposed is a novel idea that has significant practical advantages. There are assumptions that go into this, which the authors acknowledge. Nevertheless, it is a novel and sensible approach. I think a separate work could simply be based on this model and the statistical testing of its validity.

We thank the reviewer for these comments. We agree that the validity of the statistical model for AMPAR conductance could be tested using detailed models of AMPAR assembly and transport, and that it should be a separate paper.

5) This is a very complex model with 140 reactions and 47 types of elementary molecules which could be in many states. A tiny fraction of the reaction coefficients have been measured, even if they have it might not be relevant for in vivo conditions. The determination of these coefficients in the paper does not seem to depend on measured coefficients, but instead seems to depend on these coefficients being appropriate for matching experimental results. As we know, and as is shown for example in Figure 7, there is a very large space of parameter combinations that can produce very similar experimental results. What do we actually learn from all of this then? This problem is actually made clear in Figure 7, what are the authors trying to clarify in this very complex figure?

We have now simplified Figure 7 to transmit the main idea – that, despite the large number of model parameters (initial concentrations of proteins), the outcome of plasticity in the model is largely determined by 3-4 groups of species, namely, the GluR1 vs. GluR2 ratio, the main Ca^2+^ extrusion proteins, and the concentrations of PKA and PKC pathway-related proteins. Understanding the contributions of different proteins to synaptic plasticity is important since there is a large diversity of reported types of plasticity in the cortex. Previously, we illustrated three classes of LTP/LTD curves in panels F-L, in part because not all experiments demonstrate both LTP and LTD; now, we have mainly focused on a single class, namely, the one (class 6) that produced the same LTP outcome as the model with the default parameters for a large Ca^2+^ input. We have also removed panels I-L as they are not imperative for the main message of our manuscript. Instead, we added a supplementary figure (Figure 7—figure supplement 1) where the classification is based on plasticity outcome for small Ca^2+^ inputs, which better highlights the contribution of the PKC pathway.

We now write:

“The model results for the large parameter distributions of Figure 7B–E imply that there are manifestly different combinations of parameters that lead to the same LTP/LTD outcome. […] This shows our identification of critical parameters is robust to how the classification was performed.”

To highlight the model with the default parameters instead of any arbitrary parameter set, we write:

“Our model with the standard protein concentrations (GluR1 ratio 0.5, [NCX]=0.54 mM, f_PKA_ = f_PKC_ = 1.0) produced a BCM-type curve in class 6 (Figure 7A, black dashed curve).”

To further clarify the take-home message of this section, we now write:

“These data suggest that neocortical post-synaptic spines may exhibit a vast set of plasticity rules by downregulation or relatively mild upregulation of their protein expression.”

This conclusion is important, as the last Results section shows the implications of this hypothesis in synapse models fitted to real data – namely, we show that the particular types of post-synaptic plasticity can indeed be obtained by concentration changes of a similar magnitude as shown here.

6) The real complete model is the one that uses the Calcium transients from a Neuron simulation subsection “Paired pre- and post-synaptic stimulation induces PKA- and PKC-dependent spike-timing-dependent plasticity (STDP) in GluR1-GluR2-balanced synapses” and Figure 6. Only here there is a possibility that realistic calcium transients are used, and in principle this should have been used from all comparisons to data. Not only STDP protocols have real synapses, HFS and LFS stimuli also use real synapses. However, it is not clear to me how this synapse model was calibrated. Peak Ca influx rate here is much larger than for the other cases, more than an order of magnitude larger than in Figure 3 and much larger than in Figures 4, 5 as well. How was this calibrated? Was this based on the number of NMDAR and this influx through each? If so, what are these numbers? An alternative way of calibrating the spine model is to use estimates of calcium influx from Ca imaging, for example from Sabatini and Svoboda, 2002. A third option is to use whatever works, however then even this elementary component of the model is not based on biophysical realism. Several details and references are given in the subsection “Modelling the Ca^2+^ inputs and neuromodulatory inputs”, but this is still not clear to me. It would also be useful to see what the spine voltages are and how they affect the Mg block. Why is there more calcium influx at -30 ms vs. only presynaptic stimulation? Is the -30 ms measured from the first or last spike in the 4-spike train? What is it about a spike that occurred 30 ms prior to the presynaptic stimulus that affects calcium influx?

We agree with the reviewer that it is important that our model is responsive to realistic levels of Ca^2+^ inputs. To complement the simple input protocols of Figure 3 (previously Figure 4), we have now implemented the HFS and LFS protocols using real synapses in addition to the previously shown STDP protocol. We now write:

“In the above analyses, we used brief square-pulse fluxes of Ca^2+^ to the synapse model, which is a simple representation of inputs during synaptic plasticity induction protocols. […] In accordance with Figure 3 and experimental data from somatosensory cortex [Heusler et al., 2000], our model predicted that 6xHFSt induced LTP whereas LFS^-1^Hz induced LTD (Figure 3—figure supplement 3).”

To introduce this extension of the study, we added details into the Materials and methods, where we now write:

“For Figure 3—figure supplement 3 of the subsection “High-frequency stimulation (HFS) causes LTP and low-frequency stimulation (LFS) causes LTD in GluR1-GluR2-balanced synapses”, following the experiments of [Heusler et al., 2000] we used [Mg^2+^] of 1.3 mM and solely pre-synaptic stimulation of one of the two stimulus protocols: 6xHFSt – 10 bursts of 4 pulses (at 100 Hz), repeated every 100 ms, and the whole train repeated 6 times every 10 sec; LFS-1Hz – 1800 pulses delivered at a frequency of 1 Hz.”

In the same section, we have now also added details on the calibration of the biophysical model. We write:

“In the subsection “Paired pre- and post-synaptic stimulation induces PKA- and PKC-dependent spike-timing-dependent plasticity (STDP) in GluR1-GluR2-balanced synapses” (and Figure 3—figure supplement 3 of the subsection “High-frequency stimulation (HFS) causes LTP and low-frequency stimulation (LFS) causes LTD in GluR1-GluR2-balanced synapses”), we used a multicompartmental model of a layer 2/3 pyramidal cell [Markram et al., 2015] (L23 PC cADpyr229 1) to determine the amplitudes and time courses of the Ca^2+^ inputs conducted by NMDA receptors (NMDARs) when different stimulus patterns were applied. […] We set the maximal conductance of NMDARs to 0.0032 μS to compensate for the lack of the slow component in the model of the NMDA current [Markram et al., 2015] – a value significantly smaller or larger than this abolished the LTP or LTD, respectively, in our STDP model, while the AMPAR conductance was a less crucial parameter (Figure 11—figure supplement 2).”

The calcium in Figure 2 (earlier Figure 3) was injected as a constant influx for long duration, and therefore, the amplitudes of the intracellular calcium responses were much smaller than in HFS and LFS protocols (where only 3-ms pulses of calcium were given). In Figure 3 (earlier Figure 4), the rates of the square-pulse calcium inputs in HFS and LFS experiments were indeed in line with the observations from one of the only publications of calcium in response to 100 Hz input (Regehr WG, Tank DW. J Neurosci 1992;12(11):4202-23). Note that the calcium response to 100 Hz is not 100 times the response to a single PSP, to avoid producing unrealistic peak responses. The calcium inputs of Figure 5 (earlier Figure 6) are of the same order of magnitude as in Figures 3-4. We have now plotted the Ca^2+^ concentration in the spine (Figure 5E-G, Figure 6C-D) instead of the Ca^2+^ influx to make the comparison with experimental data easier. The Ca^2+^ influx and its dependence on the ISI in the STDP protocol are now illustrated in Figure 5—figure supplement 1.

We have also added panels Figure 5B-D showing the membrane potential time courses at the spine heads and the corresponding [Mg^2+^]-block factors. The inter-stimulus interval was indeed determined from the onset of the pre-synaptic stimulus until the onset of the first post-synaptic stimulus. We have now changed it to the interval from the onset of the pre-synaptic stimulus until the onset of the *last* post-synaptic stimulus, to use the same convention as experimentalists. This shifts the STDP curve 30 ms to the right. In the legend of Figure 5, we now write:

“Membrane potential at the dendritic spine when the pre-synaptic stimulation onset is 50 ms after (B), at the same time as (C), or 50 ms prior to (D) the onset of the last somatic stimulus. Inset (red): Mg^2+^-gate variable as a function of time, ranging from -80 ms to 140 ms in a similar manner as the data in the main panel.”

7) Neuromodulator dynamics. It is important that Neuromodulators have been included here. However, the assumptions about their dynamics do not make sense to me. In several sections Neuromodulators pulses are assumed to follow the Ca pulses? What is the logic here, that stimulation of axons also causes neuromodulator release? This clearly does not seem to match experiments like Seol et al., 2007, where neuromodulators are bath applied and should just be at a constant level. Is there any evidence that in other slice experiments neuromodulators are indeed release at every pulse? What about the culture experiments? It is also not clear to me what is done with neuromodulators in the subsection “Paired-pulse stimulus protocol induces PKA- and PKC-dependent spike-timing-dependent plasticity (STDP) in GluR1-GluR2-balanced synapses” where Ca transients were taken from the spine model. Here it is clear to me that constant Neuromodulator levels should be used as in the experiment.

We agree that in the subsection “Paired-pulse stimulus protocol induces PKA- and PKC-dependent spike-timing-dependent plasticity (STDP) in GluR1-GluR2-balanced synapses” it makes sense to model the same bath application of neuromodulatory agonists as done in Seol et al. We reran the simulations for Figure 5 using the same protocol of neuromodulation (10 min bath application, starting 8 min before the onset of the pairing) and updated Figure 5 with those new results (which indeed are the same as before). We have also shown that the temporal dynamics of the neuromodulation do not play a crucial role in determining the outcome of the plasticity (see our response to point #2). As it is not known what the temporal dynamics are like in vivo, we keep the pulse-like neuromodulation previously implemented in Figure 4 (now Figure 3).

We now write:

“When added as bath application, the β-adrenergic and cholinergic ligands were simulated by prolonged injections of 50 particles/s for 10 min, starting 8 min before the STDP protocol”.

To clarify that M1 receptors were activated by cholinergic ligands and that the induced LTD was dependent on the pairing interval, we rephrased the sentence as follows:

“When β-adrenergic neurotransmission was blocked but M1 receptors were activated by cholinergic ligands, the model predicted a prominent (up to 60%) decrease in GluR2 membrane expression, with little effect on GluR1 membrane expression (Figure 5K–L, purple).”

On the question on neuromodulatory inputs in slices, please see our response to point #2. We are not aware of evidence of norepinephrine or cholinergic neurotransmission in cultured cortical neurons. However, the data we previously mentioned as being from cultured neurons from auditory cortex (Kotak et al., 2007) were in fact from slices, we apologize for this error in the text. Thus, all experimental data that we reproduce in the present work were from cortical slices, except from Flores et al., 2011, which recorded field responses in prefrontal cortex in vivo.

8) Are initial conditions steady at steady state for each parameter combination? It is not clear to me if all plasticity simulations are started at the steady levels of the system for the given parameter set? Are they?

We always simulated the model for more than 1 hour before stimulation onset. This is now clarified, where we write:

“All stimulations start after 4040 sec of simulation without inputs, which is sufficient for attaining a steady state for all species (Figure 11—figure supplement 1).”

The added supplementary figure (Figure 11—figure supplement 1) shows that after one hour the concentrations change by much less than 1 molecule/sec.

Reviewer #2:1) The authors develop a complicated model of the biochemical pathways underlying LTP and LTD. It is quite a heroic effort, but on the other hand the model is hard to verify, it is hard to extract more general principles, and the predictions of the model are limited. This limits the overall impact of the work.

We thank the reviewer for this comment. We would like to point out that by mapping model entities to biological entities, the biochemical network model is straightforward to verify, because experimentalists can measure plasticity outcome while inhibiting specific molecules to test some of our predictions. A major goal of the proposed work is to make sense of the vast and confusing literature on synaptic plasticity, in which a diversity of induction protocols in a diversity of brain regions yields sometimes contradictory plasticity outcomes. Thus, a general principle extracted from our modeling study is that a relatively small number of key molecules can explain some of this diversity (as analyzed in Figure 7 and Figure 9). Another general principle is that the balance of GluR1/GluR2 subunits is important for determining the type of plasticity (see also our response to point #6 below). On the scarcity of model predictions: we have now performed additional simulations using our combination of biophysically detailed simulations of layer 2/3 pyramidal cells with the biochemically detailed post-synaptic spine model. Our results show that the shape of the STDP curve is radically affected by the number of post-synaptic stimuli per burst (Figure 6), fewer stimuli resulting in both weaker LTP and weaker LTD in an SK-channel dependent manner. This may explain why large number of stimuli per post-synaptic burst are often used in experimental settings. Our results also suggest that STDP is more correlated with the total Ca^2+^ entering the post-synaptic spine than with the peak Ca^2+^ flux (Figure 6—figure supplement 1).

In the Results section, we now added a paragraph to present and discuss the abovementioned predictions:

“The combination of our biochemically detailed model with the biophysically detailed model of layer 2/3 pyramidal cell model provides a compelling means of hypothesis testing for cortical STDP in this cell type. […] Our model predictions also agree with the observation that the plasticity outcome is not determined by Ca^2+^ transient amplitude [Nevian and Sakmann, 2006], instead, our model suggests that the total Ca^2+^ is a better predictor of the plasticity outcome: the correlation coefficient between the post-STDP synaptic conductance and the peak Ca^2+^ transient amplitude (see Figure 5—figure supplement 1E) was 0.53, while that between the post-STDP synaptic conductance and the mean Ca^2+^ input during the inter-stimulus interval (see Figure 5—figure supplement 1F) was 0.96 (Figure 6—figure supplement 1).”

We have also rewritten parts of the Discussion to better highlight the above-mentioned take-home messages. We now write:

“We also showed that our model can be fit to explain the pathway dependencies of various types of neocortical LTP/LTD data published in the literature by altering the magnitude of Ca^2+^ and ligand fluxes and the concentrations of post-synaptic proteins regulating the Ca^2+^ efflux and PKA- and PKC-pathway dynamics (Figure 9).”

Furthermore, we write:

“It is also important to know whether and to what degree GluR1 and GluR2 subunits are present in the synapse, since the balance of GluR1 and GluR2 subunits seems to be a determinant parameter permitting certain types of plasticity while prohibiting others (Figures 7B, 9E, and Figure 3—figure supplement 1).”

We added a Discussion paragraph on the total amount of Ca^2+^ vs. peak Ca^2+^ flux in determining the outcome of plasticity:

“Due to the inclusion of three major LTP/LTD pathways in the neocortex, our model provides a more accurate means than earlier models for exploring how the Ca^2+^ dynamics in the spine affects the plasticity outcome in many stimulation protocols, STDP in particular. […] The total amount of Ca^2+^ influx could thus provide a better biomarker for plasticity than the previously considered amplitude and duration of the Ca^2+^ transient [Evans and Blackwell, 2015].”

To argue for the overall impact of the present work, we would like to point out that our study is the first one (as far as we are aware) to model the contributions of both GluR1 and GluR2 to synaptic plasticity in the neocortex. This is significant since both subunits are strongly expressed in the neocortex. Our statistical plasticity rule based on the numbers GluR1 and GluR2 at the membrane is an important step in this prediction pipeline. To further verify our model according to higher-level plasticity data, we have now implemented the HFS and LFS using Ca^2+^ transients from a biophysically detailed model (Figure 3—figure supplement 3) as previously done for the STDP experiment. On the lack of general principles, we have now simplified Figure 7 to better highlight the most central model parameters affecting the plasticity outcomes in our model.

2) Some of the design decisions are hard to follow.For instance, why was the CaM activation made steeper?

A steeper Ca^2+^ activation curve has been observed in experiments. We are now showing the calcium-sensitivity of our CaM model and that of the old CaM model, overlaid with experimental data of Hoffman et al., 2014, in Figure 11B. We believe that this property of steeper CaM activation may be important for bidirectional LTP/LTD so that small amounts of calcium activate PKC but not PKA pathway, while larger calcium inputs also activate the PKA pathway – however, analyzing this particular question is out of the scope of the present work. Note that the data used to adjust the reactions was independent of the synaptic plasticity experiments we were trying to emulate. We have now addressed this:

“The response curve for CaM activation by Ca^2+^ was steeper in this model (Figure 11B), which was in better accordance with recent experimental data [Hoffman et al., 2014]”.

In Figure 11B, we replaced the amplitude of the Ca^2+^ flux in a 4xHFS protocol on the x-axis by the steady-state Ca^2+^ concentration. This is explained in the legend of Figure 11:

“Steady-state concentration of activated (bound by four Ca^2+^ ions) CaM in response to a prolonged Ca^2+^ input amplitude when the two-step (grey) or three-step (black) activation of CaM by Ca^2+^ was used. […] Red dots show experimental data from [Hoffman et al., 2014].”

3) I also did not understand how the model fitting was done by changing initial concentrations (Materials and methods). Changing the reaction rates would be a more conventional way. I wonder how these concentrations develop in the absence of stimulation. Do they stay the same, or do they have to be clamped to certain values?Yet in the subsection “The model flexibly reproduces data from various cortical LTP/LTD experiments” other variables are changed to fit the data (what are 'factors for the protein concentrations'?).

For the model construction (Materials and methods), we changed reaction rates because experimental measurements of protein concentration are less common. On the other hand, for fitting data from various cortical regions our reasoning is that the reaction rates should be relatively constant across cells of the same type, and even across cell types. However, we know that the numbers of proteins vary dramatically depending on the metabolic state of the cell, and there are also cell-type-specific and age-dependent differences in the levels of expression of different proteins. We therefore kept the rate constants fixed and varied the concentrations of different species in the subsection “The model flexibly reproduces data from various cortical LTP/LTD experiments” to show that our model can reproduce a number of experimental data sets with the same reaction rates. This could be compared to fitting of Hodgkin-Huxley-type models, where usually the same activation and inactivation parameters are assumed for an ion-channel species in many cell types, but the maximal conductance term, relating to the number of ion channels per membrane area, is varied.

We now write:

“We chose to fit protein concentrations instead of reaction rates, since the reaction rates can be considered to be the same across cell types while the protein expression is known to be cell-type and age-dependent. This is analogous to fitting maximal conductances that correlate with ion-channel densities in Hodgkin-Huxley-type models instead of the ion-channel activation and inactivation curve parameters as is usually done in the fitting of biophysically detailed neuron models.”

The concentrations of the species are in a steady state when we start the stimulation, this is now shown in Figure 11—figure supplement 1.

By factors for protein concentrations we mean parameters that control the number of a handful of proteins. We have now clarified this:

“The concentrations of upstream PKA-pathway proteins R (β-adrenergic receptor), Gs, AC1, and AC8 were varied in proportion using a factor parameter f_PKA_ ∈ [0,2], and, likewise, the concentrations of upstream PKC-pathway proteins mGluR, M1, Gq, and PLC using a factor parameter f_PKC_ ∈ [0,2].”

4) The model's complexity make it difficult to understand it's properties. For instance, does CaMKII act as a switch, and is the expression essentially binary (O'Connor and Wang)? Does it fit the observations of Nevian and Sakmann?

Our CaMKII model is not bistable, but once activated, CaMKII slowly becomes dephosphorylated (see reactions 52 and 56). The rate of this deactivation is dependent on the rate of CaMCa4 unbinding from phosphorylated CaMKII (reaction 56, forward reaction rate). These rate constants for CaMKII were tuned in a previous model to match the data of (De Koninck P and Schulman H, Science 1998 Jan 9;279(5348):227-30). However, due to the autophosphorylation of CaMKII, CaMKII activation is very steep (Figure 2I).

Our STDP model agrees with the data from Nevian and Sakmann, 2006, on the effects of decreased Ca^2+^ channel conductivity on LTD, namely, that a blockade decreases the amplitude of LTD. Our model suggests that these effects are due to decreased SK currents for post-pre intervals. Our model also agrees with their observation that the peak of the Ca^2+^ transient is a poor predictor of the plasticity outcome. As for the molecular dependence of the STDP, Nevian and Sakmann, 2006, showed that the STDP they observed was dependent on mGluRs – our predictions agree that it is PLC-pathway-dependent, but in our model and in Seol et al., 2007, the activation of the PLC pathway was driven by cholinergic receptors instead of mGluRs.

We now write:

“The paired-pulse protocol of Figure 5M (involving both β-adrenergic and cholinergic neuromodulation) caused an STDP in all cases, but decreasing the SK conductance shortened the post-pre LTD window and decreased the amplitude of LTD (Figure 5E). […] Our model predictions also agree with the observation that the plasticity outcome is not determined by Ca^2+^ transient amplitude [Nevian and Sakmann, 2006], instead, our model suggests that the total Ca^2+^ is a better predictor of the plasticity outcome: the correlation coefficient between the post-STDP synaptic conductance and the peak Ca^2+^ transient amplitude (see Figure 5—figure supplement 1E) was 0.53, while that between the post-STDP synaptic conductance and the mean Ca^2+^ input during the inter-stimulus interval (see Figure 5—figure supplement 1F) was 0.96 (Figure 6—figure supplement 1).”

5) The STDP curves look odd, with no below baseline LTD for short negative intervals.

In the original report, we measured the ISI from the onset of the pre-synaptic stimulus to the onset of the first post-synaptic stimulus. We have now changed the measurement of the ISI to be consistent with experimental conventions – it is now the interval from the pre-synaptic stimulus onset to the onset of the *last* post-synaptic stimulus (i.e., fourth somatic pulse). This means that our STDP curve is now shifted 30 ms to the right when using the four somatic stimulus pulses, separated by 10 ms. This has the consequence of now exhibiting below baseline LTD for short negative intervals. In the legend of Figure 5 we now write:

“Membrane potential at the dendritic spine when the pre-synaptic stimulation onset is 50 ms after (B), at the same time as (C), or 50 ms prior to (D) the onset of the last somatic stimulus.”

6) Does the last sentence of the subsection “High-frequency stimulation causes LTP and low-frequency stimulation causes LTD in GluR1-GluR2-balanced synapses” really imply a causal relation, so that GluR2 endocytosis leads to potentiation or depression? If so, the mechanism was not clear to me.

Yes. Our model uses the data of Oh et al., 2005, which shows that homomeric GluR1 receptors have much larger conductance than heteromeric GluR1-GluR2 tetramers (and that the conductance of homomeric GluR1 receptors, unlike that of heteromeric GluR1-GluR2 tetramers, is further increased through S831 phosphorylation). As decreasing GluR2 subunits from the spine membrane through S880 phosphorylation can significantly increase the proportion of homomeric GluR1 tetramers, our model predicts that this can lead to increase in the total conductance although the total number of AMPAR tetramers is decreased. We have now illustrated this more clearly in panels D and H of Figure 4 (earlier Figure 5) by adding a third pair of bars showing the contribution of homomeric GluR1s to the total synaptic conductance. In the legend of Figure 4, we write:

“D, H: The fraction of membrane-inserted GluR1 over all membrane-inserted GluR subunits (left), the probability of an AMPAR tetramer being homomeric GluR1 (middle), and the relative contribution of homomeric GluR1 subunits to the total conductance (i.e., summed conductance of homomeric GluR1 tetramers divided by the summed conductance of all tetramers; right).”

7) The y-axis labels on the plots are odd. In Figure 2 they put the quantity as the plot label, and the units as axis label. The authors do it correctly on the x-axis.In other figures other conventions are followed.

We have now written the labels so that units are always in parentheses. The quantity is written, as before, either before the units on the axis label or in the main panel.